# Bayesian Extensive-Rank Matrix Factorization with Rotational Invariant Priors

**Farzad Pourkamali** & **Nicolas Macris**
School of Computer and Communication Sciences,
Ecole Polytechnique Fédérale de Lausanne
{farzad.pourkamali,nicolas.macris}@epfl.ch

## Abstract

We consider a statistical model for matrix factorization in a regime where the rank of the two hidden matrix factors grows linearly with their dimension and their product is corrupted by additive noise. Despite various approaches, statistical and algorithmic limits of such problems have remained elusive. We study a Bayesian setting with the assumptions that (a) one of the matrix factors is symmetric, (b) both factors as well as the additive noise have rotational invariant priors, (c) the priors are known to the statistician. We derive analytical formulas for *Rotation Invariant Estimators* to reconstruct the two matrix factors, and conjecture that these are optimal in the large-dimension limit, in the sense that they minimize the average mean-square-error. We provide numerical checks which confirm the optimality conjecture when confronted to *Oracle Estimators* which are optimal by definition, but involve the ground-truth. Our derivation relies on a combination of tools, namely random matrix theory transforms, spherical integral formulas, and the replica method from statistical mechanics.

## 1 Introduction

Matrix factorization (MF) is the problem of reconstructing two matrices $\boldsymbol{X}$ and $\boldsymbol{Y}$ from the noisy observations of their product. Applications in signal processing and machine learning abound, such as for example dimensionality reduction [1, 2], sparse coding [3–5], representation learning [6], robust principal components analysis [7, 8], blind source separation [9], or matrix completion [10, 11].

In this work we approach the problem from a Bayesian perspective and assume that an observation or data matrix $\boldsymbol{S} = \sqrt{\kappa}\boldsymbol{XY} + \boldsymbol{W}$ is given to a statistician who knows the prior distributions of $\boldsymbol{X}$ and $\boldsymbol{Y}$ as well as the prior of the additive noise matrix $\boldsymbol{W}$ and the signal-to-noise ratio $\kappa > 0$. The task of the statistician is to construct estimators $\boldsymbol{\Xi}_X(\cdot), \boldsymbol{\Xi}_Y(\cdot)$ for the matrix factors $\boldsymbol{X}, \boldsymbol{Y}$, that ideally, minimize the average mean-square-error (MSE) $\mathbb{E}\|\boldsymbol{X} - \boldsymbol{\Xi}_X(\boldsymbol{S})\|_{\mathrm{F}}^2$ and $\mathbb{E}\|\boldsymbol{Y} - \boldsymbol{\Xi}_Y(\boldsymbol{S})\|_{\mathrm{F}}^2$ ($\|.\|_{\mathrm{F}}$ the Frobenius norm and $\mathbb{E}$ the expectation w.r.t $\boldsymbol{X}, \boldsymbol{Y}, \boldsymbol{W}$). We consider priors which are rotation invariant for all three matrices $\boldsymbol{X}, \boldsymbol{Y}, \boldsymbol{W}$ and for $\boldsymbol{X}$ we furthermore impose that it is square and symmetric. These matrix ensembles are defined precisely in section 2.1, but the reader can keep in mind the examples of Wigner or Wishart matrices for $\boldsymbol{X}$, and general Gaussian $\boldsymbol{Y}$ and $\boldsymbol{W}$ with i.i.d elements. We look at the asymptotic regime where all matrix dimensions and ranks tend to infinity at the same speed. We remark that the usual "rotation ambiguity" occuring in MF is not present because we impose that at least one of the two matrix factors is symmetric. We also remark that MF is different (and more difficult) than matrix denoising which would consist in constructing an estimator $\boldsymbol{\Xi}_{XY}(\boldsymbol{S})$ for the signal as a whole by minimizing $\mathbb{E}\|\boldsymbol{XY} - \boldsymbol{\Xi}_{XY}(\boldsymbol{S})\|_{\mathrm{F}}^2$.

The rotation invariance of the model implies that the estimators minimizing the MSE belong to the class of rotation invariant estimators (RIE). RIEs are matrix estimators which have the same singular vectors (or eigenvectors) as the observation or data matrix. These estimators have been

proposed for matrix *denoising* problems (see references [12–16] for covariance estimation, [17] for cross-covariance estimation, and [18], [19, 20] for extensions to rectangular matrices). For the present MF model, we derive optimal estimators (minimizing the MSE) that belong to the RIE class and can be computed explicitly in the large dimensional limit from the observation matrix and the knowledge of the priors. We propose:

1. an explicit RIE to estimate $X$, which requires the knowledge of the priors of *both* $X, Y$ and of the noise $W$. Moreover, under the assumption that $X$ is positive-semi-definite, a *sub-optimal* RIE can be derived which *does not* require any prior on $X$.

2. an explicit RIE to estimate $Y$, which requires the knowledge of the priors of the noise $W$ and $X$ *only* (the prior of $Y$ is not required).

3. combined with the singular value decomposition (SVD) of the observation matrix, our explicit RIEs provide a spectral algorithm to reconstruct both factors $X$ and $Y$.

The derivation of the proposed estimators relies on the replica method from statistical mechanics combined with techniques from random matrix theory and finite-rank spherical integrals [21, 22]. Although the replica method is not rigorous and involves concentration assumptions, the derivation is entirely analytical and suggests that the estimators are optimal in the limit of large dimensions. This is corroborated by numerical calculations comparing our explicit RIEs with Oracle Estimators which are optimal by definition and involve the ground-truth matrices.

## 1.1 Related literature and discussion

When the matrices $X$ and $Y$ are assumed to have *low-rank* compared to their dimension, the mathematical theory of MF has enjoyed much progress under various settings (Bayesian, spectral, algorithmic) and fundamental information theoretical and algorithmic limits have been rigorously derived. The behavior of eigenvalues/singular values and eigenvector/singular vectors of finite-rank perturbations of a Gaussian matrix is studied in [23–25] which leads to spectral estimators when the noise matrix is Gaussian distributed. For the case of factorized prior, and Gaussian noise, closed form expressions have been established for the asymptotic Bayes-optimal estimation error [26–30], and iterative algorithms based on approximate message passing has been proposed [31, 32]. The low-rank matrix denoising problem has been addressed in various other settings, such as structured noise matrix [33, 34], mismatched estimation problem [35–38], and estimation in the regime with diverging aspect-ratio of matrices [39].

In extensive-rank regimes, when the rank grows like the matrix dimensions, despite various attempts there is no solid theory of MF. One approach is based on Approximate Message Passing (AMP) methods developed in [40–42]. Despite acceptable performance in practical settings [43], as pointed out in [44] the AMP algorithms developed in these works are (theoretically) sub-optimal. Other approaches rooted in statistical physics have been considered in [44–47] but have not led to explicit reconstructions of matrix factors or algorithms. A practical probabilistic approach to MF problem is based on variational Bayesian approximations [48–50], in which one tries to approximate the posterior distribution with proper distribution. In [51] it is shown that under Gaussian priors, the solution to the MF problem is a reweighted SVD of the observation matrix. We point out here that these estimators can be seen as a RIE and therefore there seems to be a rather close relation between the RIE studied here and the variational Bayesian approach. This also suggests that adapting RIEs to real data is an interesting direction for future research. Finally, let us also mention optimization approaches where one constructs estimators by following a gradient flow (or gradient descent) trajectory of a training loss of the type $\|S - XY\|_F^2$ + reg. term (see [52], [53] for analysis in rotation invariant models). Benchmarking these various other algorithmic approaches against our explicit RIEs (conjectured to be optimal) is outside the scope of this work and is left for future work.

Constraints such as sparsity or non-negativity of the matrix entries which have important applications [54] are not covered by our theory. Despite this drawback, we believe that the proposed estimators are important both for theoretical and practical purposes. Even in non-rotation invariant problems our explicit RIEs may serve as sub-optimal estimators, and as we show in an example they can be used as a "warmed-up" spectral initialization for more efficient algorithms (see for example [31, 55] for related ideas in other contexts). The methodology developed here may open up the way to further analysis in inference and learning problems perhaps also in the context of neural networks where extensive rank weight matrices must be estimated.

## 1.2 Organization and notations

In section 2, we introduce the precise MF model, general class of RIEs, and the Oracle estimators. In section 3, we present the explicit RIEs (and algorithm) to estimate $\boldsymbol{X}$ and $\boldsymbol{Y}$. We provide the numerical examples and calculations in section 4. In section 5, we sketch the derivation of RIE for $\boldsymbol{X}$, while the one for $\boldsymbol{Y}$ is similar and deferred to the appendices.

The following notations are used throughout. For a vector $\boldsymbol{\gamma} \in \mathbb{R}^N$ we denote by $\boldsymbol{\Gamma} \in \mathbb{R}^{N \times M}$ a matrix constructed as $\boldsymbol{\Gamma} = \left[ \begin{array}{c|c} \boldsymbol{\Gamma}_N & \boldsymbol{0}_{N \times (M-N)} \end{array} \right]$ with $\boldsymbol{\Gamma}_N \in \mathbb{R}^{N \times N}$ a diagonal matrix with diagonal $\boldsymbol{\gamma}$. The same notations will also be used for the vector $\boldsymbol{\sigma}$ and the corresponding matrix $\boldsymbol{\Sigma}$ and . For a sequence of non-symmetric matrices $\boldsymbol{A}$ of growing size, we denote the limiting empirical singular value distribution (ESD) by $\mu_A$, and the limiting empirical eigenvalue distribution of $\boldsymbol{A}\boldsymbol{A}^\intercal$ by $\rho_A$. For a sequence of symmetric matrices $\boldsymbol{B}$ of growing size, we denote the limiting empirical eigenvalue distribution by $\rho_B$, and the limiting eigenvalue distribution of $\boldsymbol{B}^2$ by $\rho_{B^2}$.

# 2 Matrix factorization model and rotation invariant estimators

## 2.1 Matrix factorization model

Let $\boldsymbol{X} = \boldsymbol{X}^\intercal \in \mathbb{R}^{N \times N}$ a symmetric matrix distributed according to a rotationally invariant prior $P_X(\boldsymbol{X})$, i.e., for any orthogonal matrix $\boldsymbol{O} \in \mathbb{R}^{N \times N}$ we have $P_X(\boldsymbol{O}\boldsymbol{X}\boldsymbol{O}^\intercal) = P_X(\boldsymbol{X})$. Let also $\boldsymbol{Y} \in \mathbb{R}^{N \times M}$ be distributed according to a bi-rotationally invariant prior $P_Y(\boldsymbol{Y})$, i.e. for any orthogonal matrices $\boldsymbol{U} \in \mathbb{R}^{N \times N}, \boldsymbol{V} \in \mathbb{R}^{M \times M}$ we have $P_Y(\boldsymbol{U}\boldsymbol{Y}\boldsymbol{V}^\intercal) = P_Y(\boldsymbol{Y})$. We observe the data matrix $\boldsymbol{S} \in \mathbb{R}^{N \times M}$,

$$\boldsymbol{S} = \sqrt{\kappa}\boldsymbol{X}\boldsymbol{Y} + \boldsymbol{W} \tag{1}$$

where $\boldsymbol{W} \in \mathbb{R}^{N \times M}$ is also bi-rotationally invariant distributed, and $\kappa \in \mathbb{R}_+$ is proportional to the signal-to-noise-ratio (SNR). The goal is to recover *both factors* $\boldsymbol{X}$ and $\boldsymbol{Y}$ from the data matrix $\boldsymbol{S}$. For definiteness, we consider the regime $M \geq N$ with aspect ratio $N/M \to \alpha \in (0, 1]$ as $N \to \infty$. The case of $\alpha > 1$ can be analyzed in the same manner and is presented in appendix F. Furthermore, we assume that the entries of $\boldsymbol{X}, \boldsymbol{Y}$ and $\boldsymbol{W}$ are of the order $O(1/\sqrt{N})$. This scaling is such that the eigenvalues of $\boldsymbol{X}$ and singular values of $\boldsymbol{Y}, \boldsymbol{W}$ and $\boldsymbol{S}$ are of the order $O(1)$ as $N \to \infty$.

**Assumption 1.** *The empirical eigenvalue distribution of $\boldsymbol{X}$ converge weakly to measure $\rho_X$, and the ESD of $\boldsymbol{Y}, \boldsymbol{W}$ converge weakly to measures $\mu_Y, \mu_W$ with bounded support on the real line. Moreover, these measures are known to the statistician. He can deduce (in principle) these measures from the priors on $\boldsymbol{X}, \boldsymbol{Y}, \boldsymbol{W}$.*

**Remark 1.** *In a general formulation of matrix factorization the hidden matrices have dimensions $\boldsymbol{X} \in \mathbb{R}^{N \times H}, \boldsymbol{Y} \in \mathbb{R}^{H \times M}$, and in the Bayesian framework with bi-rotational invariant priors for both factors, the optimal estimators are trivially the zero matrix. Indeed, from bi-rotational invariance we have $P_X(-\boldsymbol{X}) = P_X(\boldsymbol{X})$, $P_Y(-\boldsymbol{Y}) = P_Y(\boldsymbol{Y})$, which implies that the Bayesian estimate is zero. Here, by imposing that $\boldsymbol{X} \in \mathbb{R}^{N \times N}$ is symmetric and $P_X(\boldsymbol{O}\boldsymbol{X}\boldsymbol{O}^\intercal) = P_X(\boldsymbol{X})$, we can break this symmetry and find non-trivial estimators. This is due to the fact that the map $\boldsymbol{X} \to -\boldsymbol{X}$ cannot be realized as a (real) orthogonal transformation, so $P_X(-\boldsymbol{X}) = P_X(\boldsymbol{X})$ does not hold in general (various examples are given in section 4 and appendices). Of course, if the prior is even, e.g. Wigner ensemble, again the Bayesian posterior estimate is trivially zero for both factors. As we will see our RIEs are consistent with these observations.*

## 2.2 Rotation invariant estimators

To recover matrices $\boldsymbol{X}, \boldsymbol{Y}$ from $\boldsymbol{S}$, we consider two denoising problems. One is recovering $\boldsymbol{X}$ by treating both $\boldsymbol{Y}, \boldsymbol{W}$ as "noise" matrices, and the other is estimating $\boldsymbol{Y}$ by treating $\boldsymbol{X}, \boldsymbol{W}$ as "noise". As will become clear the procedure is not iterative, and the two denoising problems are solved independently and simultaneously. In the following, for each of these two problems, we introduce two rotation invariant classes of estimators and discuss their optimum *Oracle* estimators. We then provide an explicit construction and algorithm for RIEs which we conjecture have the optimum performance of Oracle estimators in the large $N$ limit.

### 2.2.1 RIE class for $X$

Consider the SVD of $S = U_S \Gamma V_S^\mathsf{T}$, where $U_S \in \mathbb{R}^{N \times N}$, $V_S \in \mathbb{R}^{M \times M}$ are orthogonal, and $\Gamma \in \mathbb{R}^{N \times M}$ is a diagonal matrix with singular values of $S$ on its diagonal, $(\gamma_i)_{1 \leq i \leq N}$. A rotational invaraint estimator for $X$ is denoted $\Xi_X(S)$, and is constructed as:

$$\Xi_X(S) = U_S \operatorname{diag}(\xi_{x1}, \ldots, \xi_{xN}) U_S^\mathsf{T} \tag{2}$$

where $\xi_{x1}, \ldots, \xi_{xN}$ are the eigenvalues of the estimator.

First, we derive an *Oracle estimator* by minimizing the squared error $\frac{1}{N} \| X - \Xi_X(S) \|_\mathrm{F}^2$ for a given instance, over the RIE class or equivalently over the choice of the eigenvalues $(\xi_{xi})_{1 \leq i \leq N}$. Let the eigen-decomposition of $X$ be $X = \sum_{i=1}^N \lambda_i x_i x_i^\mathsf{T}$ with $x_i \in \mathbb{R}^N$ eigenvectors of $X$. The error can be expanded as:

$$\frac{1}{N} \| X - \Xi_X(S) \|_\mathrm{F}^2 = \frac{1}{N} \sum_{i=1}^N \lambda_i^2 + \frac{1}{N} \sum_{i=1}^N \xi_{xi}^2 - \frac{2}{N} \sum_{i=1}^N \xi_{xi} \sum_{j=1}^N \lambda_j \left( u_i^\mathsf{T} x_j \right)^2$$

where $u_i$'s are columns of $U_S$. Minimizing over $\xi_{xi}$'s, we find the optimum among the RIE class:

$$\Xi_X^*(S) = \sum_{i=1}^N \xi_{xi}^* u_i u_i^\mathsf{T}, \quad \xi_{xi}^* = \sum_{j=1}^N \lambda_j \left( u_i^\mathsf{T} x_j \right)^2 = u_i^\mathsf{T} X u_i \tag{3}$$

Expression (3) defines the Oracle estimator which requires the knowledge of signal matrix $X$. Surprisingly, in the large $N$ limit, the optimal eigenvalues $(\xi_{xi}^*)_{1 \leq i \leq N}$ can be computed from the observation matrix and knowledge of the measures $\rho_X, \mu_Y, \mu_W$. In the next section, we show that this leads to an *explicitly computable* (or algorithmic) RIE, which we conjecture to be optimal as $N \to \infty$, in the sense that its performance matches the one of the Oracle estimator.

Now we remark that the Oracle estimator is not only optimal within the rotation invariant class but is also Bayesian optimal. From the Bayesian estimation point of view, one wishes to minimize the average mean squared error (MSE) $\mathrm{MSE}_{\hat{X}} \equiv \frac{1}{N} \mathbb{E} \| X - \hat{X}(S) \|_\mathrm{F}^2$, where the expectation is over $X, Y, W$, and $\hat{X}(S)$ is an estimator of $X$. The MSE is minimized for $\hat{X}^*(S) = \mathbb{E}[X | S]$ which is the posterior mean. Therefore, the posterior mean estimator has the minimum MSE (MMSE) among all possible estimators, in particular $\mathrm{MSE}_{\hat{X}^*} \leq \mathrm{MSE}_{\Xi_X^*}$ for any $N$. In appendix A.1, we show that, for rotational invariant priors, the posterior mean estimator is inside the RIE class. Thus, since $\Xi_X^*(S)$ is optimum among the RIE class $\mathrm{MSE}_{\Xi_X^*} \leq \mathrm{MSE}_{\hat{X}^*}$. Therefore, we conclude that the Oracle estimator (3) is Bayesian optimal in the sense that $\mathrm{MSE}_{\Xi_X^*} = \mathrm{MSE}_{\hat{X}^*} = \mathrm{MMSE}$.

### 2.2.2 RIE class for $Y$

Estimators for $Y$ from the rotation invariant class are denoted $\Xi_Y(S)$, and are constructed as:

$$\Xi_Y(S) = U_S \left[ \operatorname{diag}(\xi_{y_1}, \ldots, \xi_{y_N}) \mid 0_{N \times (M-N)} \right] V_S^\mathsf{T} \tag{4}$$

where $\xi_{y_1}, \ldots, \xi_{y_N}$ are the singular values of the estimator.

Let the SVD of $Y$ be $Y = \sum_{i=1}^N \sigma_i y_i^{(l)} y_i^{(r)\mathsf{T}}$ with $y_i^{(l)} \in \mathbb{R}^N$, $y_i^{(r)} \in \mathbb{R}^M$ the left and right singular vectors of $Y$. To derive an *Oracle estimator*, we proceed as above. Expanding the error, we have:

$$\frac{1}{N} \| Y - \Xi_Y(S) \|_\mathrm{F}^2 = \frac{1}{N} \sum_{i=1}^N \sigma_i^2 + \frac{1}{N} \sum_{i=1}^N \xi_{y_i}^2 - \frac{2}{N} \sum_{i=1}^N \xi_{y_i} \sum_{j=1}^N \sigma_j \left( u_i^\mathsf{T} y_j^{(l)} \right) \left( v_i^\mathsf{T} y_j^{(r)} \right)$$

where $v_i$'s are columns of $V_S$. Minimizing over $\xi_{y_i}$'s, we find the optimum among the RIE class:

$$\Xi_Y^*(S) = \sum_{i=1}^N \xi_{y_i}^* u_i v_i^\mathsf{T}, \quad \xi_{y_i}^* = \sum_{j=1}^N \sigma_j \left( u_i^\mathsf{T} y_j^{(l)} \right) \left( v_i^\mathsf{T} y_j^{(r)} \right) = u_i^\mathsf{T} Y v_i \tag{5}$$

Expression (5) defines the Oracle estimator which requires the knowledge of signal matrix $Y$. Like for the case of $X$, in the large $N$ limit we can derive the optimal singular values $(\xi_{y_i}^*)_{1 \leq i \leq N}$ in terms

of the singular values of observation matrix and knowledge of the measures $\rho_X, \mu_W$. This leads to an *explicitly computable* (or algorithmic) RIE, which is conjectured to be optimal as $N \to \infty$, in the sense that it has the same performance as the Oracle estimator. Note that unlike the estimator for $X$, we do not need the knowledge of $\mu_Y$.

In appendix A.2, we show that for bi-rotationally invariant priors the posterior mean estimator $\hat{Y}^*(S) = \mathbb{E}[Y|S]$ belongs to the RIE class, which (by similar arguments to the case of $X$) implies that the Oracle estimator (5) is Bayesian optimal.

## 3 Algorithmic RIEs for the matrix factors

In this section, we present our explicit RIEs for $X, Y$ and the corresponding algorithm. We conjecture that their performance matches the one of Oracles estimators in the large $N$ limit and they are therefore Bayesian optimal in this limit. Let us first give a brief reminder on useful transforms in random matrix theory.

### 3.1 Preliminaries on transforms in random matrix theory

For a probability density function $\rho(x)$ on $\mathbb{R}$, the *Stieltjes* (or *Cauchy*) transform is defined as

$$\mathcal{G}_\rho(z) = \int_{\mathbb{R}} \frac{1}{z-x} \rho(x)\, dx \quad \text{for } z \in \mathbb{C}\backslash\mathrm{supp}(\rho)$$

By Plemelj formulae we have for $x \in \mathbb{R}$,

$$\lim_{\epsilon \to 0^+} \mathcal{G}_\rho(x - \mathrm{i}\epsilon) = \pi \mathsf{H}[\rho](x) + \pi \mathrm{i}\rho(x) \tag{6}$$

with $\mathsf{H}[\rho](x) = \text{p.v.} \frac{1}{\pi} \int_{\mathbb{R}} \frac{\rho(t)}{x-t} dt$ the *Hilbert* transform of $\rho$ (here p.v. stands for "principal value"). Denoting the inverse of $\mathcal{G}_\rho(z)$ by $\mathcal{G}_\rho^{-1}(z)$, the *R-transform* of $\rho$ is defined as [56]:

$$\mathcal{R}_\rho(z) = \mathcal{G}_\rho^{-1}(z) - \frac{1}{z}$$

For a probability density function $\mu$ with support contained in $[-K, K]$ with $K > 0$, we define a generating function of (even) moments $\mathcal{M}_\mu : [0, K^{-2}] \to \mathbb{R}_+$ as $\mathcal{M}_\mu(z) = \int \frac{1}{1-t^2z} \mu(t)\, dt - 1$. For $\alpha \in (0, 1]$, define $T^{(\alpha)}(z) = (\alpha z + 1)(z + 1)$, and $\mathcal{H}_\mu^{(\alpha)}(z) = zT^{(\alpha)}(\mathcal{M}_\mu(z))$. The *rectangular R-transform* with aspect ratio $\alpha$ is defined as [57]:

$$\mathcal{C}_\mu^{(\alpha)}(z) = T^{(\alpha)^{-1}}\left(\frac{z}{\mathcal{H}_\mu^{(\alpha)^{-1}}(z)}\right)$$

### 3.2 Explicit RIE for $X$

The RIE for $X$ is constructed as $\widehat{\Xi_X^*}(S) = \sum_{i=1}^N \widehat{\xi}_{xi}^* u_i u_i^\mathsf{T}$ with eigenvalues $\left(\widehat{\xi}_{xi}^*\right)_{1 \le i \le N}$:

$$\widehat{\xi}_{xi}^* = \frac{1}{2\kappa\pi\bar{\mu}_S(\gamma_i)} \,\mathrm{Im} \lim_{z \to \gamma_i - \mathrm{i}0^+} \left\{ \frac{1}{\zeta_3}\left[ \mathcal{G}_{\rho_X}\left(\sqrt{\frac{z-\zeta_1}{\kappa\zeta_3}}\right) + \mathcal{G}_{\rho_X}\left(-\sqrt{\frac{z-\zeta_1}{\kappa\zeta_3}}\right) \right] \right\} \tag{7}$$

where $\gamma_i$ is the $i$-th singular value of $S$, $\bar{\mu}_S$ is the symmetrized limiting ESD of $S$, and

$$\zeta_1 = \frac{1}{\mathcal{G}_{\bar{\mu}_S}(z)} \mathcal{C}_{\mu_W}^{(\alpha)}\left( \mathcal{G}_{\bar{\mu}_S}(z)\left[\alpha\mathcal{G}_{\bar{\mu}_S}(z) + \frac{1-\alpha}{z}\right] \right) \tag{8}$$

and $\zeta_3$ satisfies [1]:

$$(z - \zeta_1)\mathcal{G}_{\bar{\mu}_S}(z) - 1 = \mathcal{C}_{\mu_Y}^{(\alpha)}\left( \frac{1}{\zeta_3}\left[\alpha\mathcal{G}_{\bar{\mu}_S}(z) + \frac{1-\alpha}{z}\right]\left[(z-\zeta_1)\mathcal{G}_{\bar{\mu}_S}(z) - 1\right] \right) \tag{9}$$

**Remark 2.** *If $\rho_X$ is a symmetric measure, $\rho_X(x) = \rho_X(-x)$, then $\mathcal{G}_{\rho_X}(-z) = -\mathcal{G}_{\rho_X}(z)$. This implies that the optimal eigenvalues $\left(\widehat{\xi}_{xi}^*\right)_{1 \le i \le N}$ in (7) are all zero, and $\widehat{\Xi_X^*}(S) = 0$, see figure 4.*

---

[1]$\zeta_1, \zeta_3$ are the only parameters which appear in the final estimator. However, in derivation of the RIE, we have defined other parameters which do not appear in the final estimator and we omit them here.

### 3.2.1 An estimator for $X^2$

It is interesting to note that we can construct a RIE for $X^2$ as $\widehat{\Xi^*_{X^2}}(S) = \sum_{i=1}^N \widehat{\xi^*_{x^2\,i}} u_i u_i^\mathsf{T}$ with eigenvalues $\left(\widehat{\xi^*_{x^2\,i}}\right)_{1 \le i \le N}$:

$$\widehat{\xi^*_{x^2\,i}} = \frac{1}{\kappa} \frac{1}{\pi \bar{\mu}_S(\gamma_i)} \operatorname{Im} \lim_{z \to \gamma_i - \mathrm{i}0^+} \frac{z - \zeta_1}{\zeta_3} \mathcal{G}_{\bar{\mu}_S}(z) - \frac{1}{\zeta_3} \tag{10}$$

with $\zeta_1, \zeta_3$ as in (8), (9). Note that, $\zeta_1, \zeta_3$ can be evaluated using the observation matrix and the knowledge of $\mu_Y, \mu_W$, and therefore this time the statistician *does not need to know the prior of $X$*. Furthermore, assuming that $X$ is positive semi-definite (PSD), we can construct a sub-optimal RIE for $X$ by using $\sqrt{\widehat{\xi^*_{x^2\,i}}}$ for the eigenvalues of the estimator.

### 3.2.2 Case of Gaussian $Y, W$

If $Y, W$ have i.i.d. Gaussian entries with variance $1/N$, then $\mathcal{C}^{(\alpha)}_{\mu_Y}(z) = \mathcal{C}^{(\alpha)}_{\mu_W}(z) = z/\alpha$. Consequently, $\zeta_1, \zeta_3$ can easily be computed to be $\zeta_1 = \zeta_3 = \mathcal{G}_{\bar{\mu}_S}(z) + (1-\alpha)/(\alpha z)$, thus the estimator (7) can be evaluated from the observation matrix. In particular, the estimator (10) simplifies to:

$$\widehat{\xi^*_{x^2\,i}} = \frac{1}{\kappa}\left[ -1 + \frac{1}{\alpha\left(\pi^2 \bar{\mu}_S(\gamma_i)^2 + \left(\pi \mathsf{H}[\bar{\mu}_S](\gamma_i) + \frac{1-\alpha}{\alpha\gamma_i}\right)^2\right)} \right] \tag{11}$$

## 3.3 Explicit RIE for $Y$

Our explicit RIE for $Y$ is constructed as $\widehat{\Xi^*_Y}(S) = \sum_{i=1}^N \widehat{\xi^*_{y\,i}} u_i v_i^\mathsf{T}$ with singular values $\left(\widehat{\xi^*_{y\,i}}\right)_{1 \le i \le N}$:

$$\widehat{\xi^*_{y\,i}} = \frac{1}{\sqrt{\kappa}} \frac{1}{\pi \bar{\mu}_S(\gamma_i)} \operatorname{Im} \lim_{z \to \gamma_i - \mathrm{i}0^+} q_4 \tag{12}$$

where $\gamma_i$ is the $i$-th singular value of $S$, and $q_4$ is the solution to the following system of equations [2]:

$$\begin{cases} \beta_1 = \frac{\mathcal{C}^{(\alpha)}_{\mu_W}(q_1 q_2)}{q_1} + \frac{1}{2}\sqrt{\frac{q_3}{q_1}}\left(\mathcal{R}_{\rho_X}\left(q_4 + \sqrt{q_1 q_3}\right) - \mathcal{R}_{\rho_X}\left(q_4 - \sqrt{q_1 q_3}\right)\right) \\ \beta_4 = \frac{1}{2}\left(\mathcal{R}_{\rho_X}\left(q_4 + \sqrt{q_1 q_3}\right) + \mathcal{R}_{\rho_X}\left(q_4 - \sqrt{q_1 q_3}\right)\right) \\ q_1 = \mathcal{G}_{\bar{\mu}_S}(z), \quad q_2 = \alpha \mathcal{G}_{\bar{\mu}_S}(z) + (1-\alpha)\frac{1}{z} \\ q_3 = \frac{(z-\beta_1)^2}{\beta_4^2}\mathcal{G}_{\bar{\mu}_S}(z) - \frac{z-\beta_1}{\beta_4^2}, \quad q_4 = \frac{z-\beta_1}{\beta_4}\mathcal{G}_{\bar{\mu}_S}(z) - \frac{1}{\beta_4} \end{cases} \tag{13}$$

Similarly to the estimator derived for $X$, if $\rho_X$ is a symmetric measure then the optimal singular values for the estimator of $Y$ are all zero, see remark 5.

If $X$ is a shifted Wigner matrix, i.e. $X = F + cI$ with $F = F^\mathsf{T} \in \mathbb{R}^{N \times N}$ having i.i.d. Gaussian entries with variance $1/N$ and $c \ne 0$ a real number, then $\mathcal{R}_{\rho_X}(z) = z + c$. Moreover, if $W$ is Gaussian matrix with variance $1/N$, then the set of equations (13) simplifies to a great extent, and we can compute $q_4$ analytically in terms of $\mathcal{G}_{\bar{\mu}_S}(z)$, see appendix D.4.

## 3.4 Algorithmic nature of the RIEs

The explicit RIEs (7) and (12) proposed in this section, provide spectral algorithms to estimate the matrix factors from the data matrix (and the priors). An essential ingredient that must be extracted from the data matrix is $\mathcal{G}_{\bar{\mu}_S}(z)$. This quantity can be approximated from the observation matrix using Cauchy kernel method introduced in [58](see section 19.5.2), from which $\bar{\mu}_S(.)$ can be approximated using (6). Therefore, given an observation matrix $S$, the spectral algorithm proceeds as follows:

1. Compute the SVD of $S$.
2. Approximate $\mathcal{G}_{\bar{\mu}_S}(z)$ from the singular values of $S$.
3. Construct the RIEs for $X, Y$ as proposed in paragraphs 3.2, 3.3.

---

[2]Like the case for $X$, we omit some of the parameters which do not appear in the final estimator.

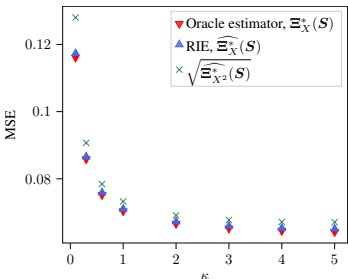

Figure 1: MSE of estimating $X$. MSE is normalized by the norm of the signal, $\|X\|_{\mathrm{F}}^2$. $X$ is a Wishart matrix with aspect ratio $1/4$, $X = \mathbf{H}\mathbf{H}^{\mathsf{T}}$ with $\mathbf{H} \in \mathbb{R}^{N \times 4N}$ having i.i.d. Gaussian entries of variance $1/N$. Both $Y$ and $W$ are $N \times M$ matrices with i.i.d. Gaussian entries of variance $1/N$. RIE is applied to $N = 2000$, $M = 4000$, and the results are averaged over 10 runs (error bars are invisible).

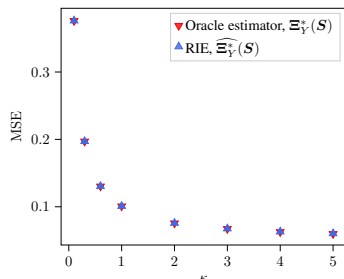

Figure 2: MSE of estimating $Y$. MSE is normalized by the norm of the signal, $\|Y\|_{\mathrm{F}}^2$. $Y$ has uniform spectral density, $\mathcal{U}([1,3])$. $X$ is a shifted Wigner matrix with $c = 3$, and $W$ is a $N \times M$ matrix with i.i.d. Gaussian entries of variance $1/N$. RIE is applied to $N = 2000$, $M = 4000$, and the results are averaged over 10 runs (error bars are invisible).

## 4 Numerical results

### 4.1 Performance of RIE for $X$

We consider the case where $Y, W$ both have i.i.d. Gaussian entries of variance $1/N$, and $X$ is a Wishart matrix, $X = \mathbf{H}\mathbf{H}^{\mathsf{T}}$ with $\mathbf{H} \in \mathbb{R}^{N \times 4N}$ having i.i.d. Gaussian entries of variance $1/N$. For various SNRs, we examine the performance of two proposed estimators, the RIE (7), and the square-root of the estimator (10) (since $X$ is PSD), which is sub-optimal. In figure 1, the MSEs of these algorithmic estimators are compared with the one of Oracle estimator (3). We see that the average performance of the algorithmic RIE $\widehat{\Xi_X^*}(S)$ is very close to the (optimal) Oracle estimator $\Xi_X^*(S)$ (relative errors are small and provided in the appendices) and we believe that the slight mismatch is due to the numerical approximations and finite-size effects. Note that, although the estimator $\sqrt{\widehat{\Xi_{X^2}^*}(S)}$ is sub-optimal, it does not use any prior knowledge of $X$. For more examples, details of the numerical experiments and the relative error of the estimators, we refer to appendix C.3.

### 4.2 Performance of RIE for $Y$

We consider the case where $W$ has i.i.d. Gaussian entries of variance $1/N$, and $X$ is a shifted Wigner matrix with $c = 3$. Matrix $Y$ is constructed as $Y = U_Y \Sigma V_Y^{\mathsf{T}}$ with $U_Y \in \mathbb{R}^{N \times N}, V_Y \in \mathbb{R}^{M \times M}$ are Haar distributed, and the singular values are generated independently from the uniform distribution on $[1, 3]$. MSEs of the RIE (12) and the Oracle estimator (5) are illustrated in figure 2. We see that the performance of the algorithmic RIE $\widehat{\Xi_Y^*}(S)$ is very close to the optimal estimator $\Xi_Y^*(S)$.

**Non-rotational invariant prior**   In another example, which we omit here, with the same settings for $X, W$, we consider the case where $Y$ is a sparse matrix with entries distributed according to Bernoulli-Rademacher prior. The RIE is not optimal in this setting (since the prior is not bi-rotational invariant), however applying a simple thresholding function on the matrix constructed by RIE yields an estimate with lower MSE. This observation suggests that for the case of general priors, the RIEs can provide a spectral initialization for more efficient estimators. For more details and examples, see appendix D.4.

### 4.3 Comparing RIEs of matrix factorization and denoising

The proposed RIEs, namely (7) and (12), simplify greatly when the matrices $W, Y$ are Gaussian, and $X$ is a shifted Wigner matrix. We perform experiments with these priors, where for a given observation matrix $S$, we look at the RIEs of $X, Y$ for the *MF problem*, and simultaneously at the RIE of the product $XY$ as a whole for the *denoising problem* with formulas introduced in [19] (which can also be obtained by taking $X$ to be the identity matrix, see appendix D.3.1). Figure 3

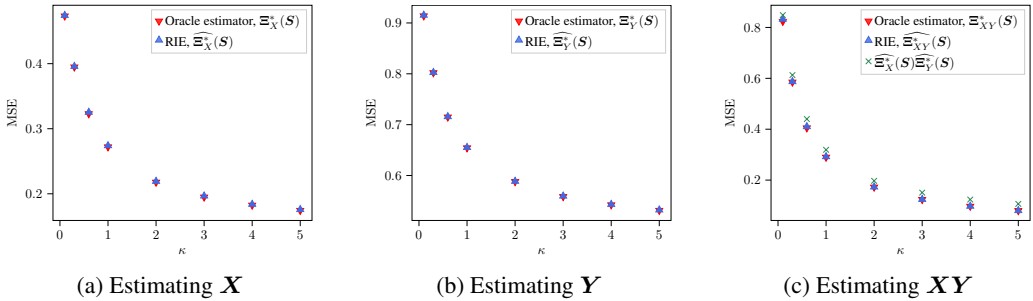

<p>(a) Estimating $\boldsymbol{X}$        (b) Estimating $\boldsymbol{Y}$        (c) Estimating $\boldsymbol{XY}$</p>

Figure 3: MSE of factorization problem. MSE is normalized by the norm of the signal. $\boldsymbol{X}$ is a shifted Wigner matrix with $c = 1$, and both $\boldsymbol{Y}$ and $\boldsymbol{W}$ are $N \times M$ matrices with i.i.d. Gaussian entries of variance $1/N$. RIE is applied to $N = 2000$, $M = 4000$. In each run, the observation matrix $\boldsymbol{S}$ is generated according to (1), and the factors $\boldsymbol{X}, \boldsymbol{Y}$ are estimated simultaneously from $\boldsymbol{S}$. Results are averaged over 10 runs (error bars are invisible).

illustrates these experiments. In particular the MSE of the denoising-RIE matches well the one of the associated Oracle estimator, and as expected is lower than the MSE of the product of MF-RIEs.

## 5 Derivation of the explicit RIEs

In this section, we sketch the derivation of our explicit RIE for $\boldsymbol{X}$. The RIE for $\boldsymbol{Y}$ is derived similarly, but requires more involved analysis and is presented in appendix D. For simplicity, we take the SNR parameter in (1) to be 1, so the model is $\boldsymbol{S} = \boldsymbol{XY} + \boldsymbol{W}$. The optimal eigenvalues are constructed as $\xi_{xi}^* = \sum_{j=1}^{N} \lambda_j \left( \boldsymbol{u}_i^\mathsf{T} \boldsymbol{x}_j \right)^2$. We assume that in the large $N$ limit, $\xi_{xi}^*$ can be approximated by its expectation and we introduce

$$\widehat{\xi}_{xi}^* = \sum_{j=1}^{N} \lambda_j \, \mathbb{E}\left[ \left( \boldsymbol{u}_i^\mathsf{T} \boldsymbol{x}_j \right)^2 \right] \tag{14}$$

where the expectation is over the (left) singular vectors of the observation matrix $\boldsymbol{S}$. Therefore, to compute these eigenvalues, we need to find the mean squared overlap $\mathbb{E}\left[ \left( \boldsymbol{u}_i^\mathsf{T} \boldsymbol{x}_j \right)^2 \right]$ between eigenvectors of $\boldsymbol{X}$ and singular vectors of $\boldsymbol{S}$. In what follows, we will see that (a rescaling of) this quantity can be expressed in terms of $i$-th singular value of $\boldsymbol{S}$ and $j$-th eigenvector of $\boldsymbol{X}$ (and the limiting measures, indeed). Thus, we will use the notation $O_X(\gamma_i, \lambda_j) := N\mathbb{E}\left[ \left( \boldsymbol{u}_i^\mathsf{T} \boldsymbol{x}_j \right)^2 \right]$ in the following. In the next section, we discuss how the overlap can be computed from the resolvent of the "Hermitized" version of $\boldsymbol{S}$.

### 5.1 Relation between overlap and resolvent

Construct the matrix $\boldsymbol{\mathcal{S}} \in \mathbb{R}^{(N+M) \times (N+M)}$ from the observation matrix:

$$\boldsymbol{\mathcal{S}} = \left[ \begin{array}{cc} \boldsymbol{0}_{N \times N} & \boldsymbol{S} \\ \boldsymbol{S}^\mathsf{T} & \boldsymbol{0}_{M \times M} \end{array} \right]$$

By Theorem 7.3.3 in [59], $\boldsymbol{\mathcal{S}}$ has the following eigen-decomposition:

$$\boldsymbol{\mathcal{S}} = \left[ \begin{array}{ccc} \hat{\boldsymbol{U}}_S & \hat{\boldsymbol{U}}_S & \boldsymbol{0} \\ \hat{\boldsymbol{V}}_S^{(1)} & -\hat{\boldsymbol{V}}_S^{(1)} & \boldsymbol{V}_S^{(2)} \end{array} \right] \left[ \begin{array}{ccc} \boldsymbol{\Gamma}_N & \boldsymbol{0} & \boldsymbol{0} \\ \boldsymbol{0} & -\boldsymbol{\Gamma}_N & \boldsymbol{0} \\ \boldsymbol{0} & \boldsymbol{0} & \boldsymbol{0} \end{array} \right] \left[ \begin{array}{ccc} \hat{\boldsymbol{U}}_S & \hat{\boldsymbol{U}}_S & \boldsymbol{0} \\ \hat{\boldsymbol{V}}_S^{(1)} & -\hat{\boldsymbol{V}}_S^{(1)} & \boldsymbol{V}_S^{(2)} \end{array} \right]^\mathsf{T} \tag{15}$$

with $\boldsymbol{V}_S = \left[ \begin{array}{cc} \boldsymbol{V}_S^{(1)} & \boldsymbol{V}_S^{(2)} \end{array} \right]$ in which $\boldsymbol{V}_S^{(1)} \in \mathbb{R}^{M \times N}$. And, $\hat{\boldsymbol{V}}_S^{(1)} = \frac{1}{\sqrt{2}} \boldsymbol{V}_S^{(1)}$, $\hat{\boldsymbol{U}}_S = \frac{1}{\sqrt{2}} \boldsymbol{U}_S$. Eigenvalues of $\boldsymbol{\mathcal{S}}$ are signed singular values of $\boldsymbol{S}$, therefore the limiting eigenvalue distribution of $\boldsymbol{\mathcal{S}}$ (ignoring zero eigenvalues) is the same as the limiting symmetrized singular value distribution of $\boldsymbol{S}$. Define the resolvent of $\boldsymbol{\mathcal{S}}$,

$$\boldsymbol{G}_{\mathcal{S}}(z) = (z\boldsymbol{I} - \boldsymbol{\mathcal{S}})^{-1}$$

We assume that as $N \to \infty$ and $z$ is not too close to the real axis, the matrix $\boldsymbol{G}_{\mathcal{S}}(z)$ concentrates around its mean. Consequently, the value of $\boldsymbol{G}_{\mathcal{S}}(z)$ becomes uncorrelated with the particular

realization of $S$. Specifically, as $N \to \infty$, $G_S(z)$ converges to a deterministic matrix for any fixed value of $z \in \mathbb{C} \backslash \mathbb{R}$ (independent of N). Denote the eigenvectors of $S$ by $\mathbf{s}_i \in \mathbb{R}^{M+N}$, $i = 1, \ldots, M + N$. For $z = x - i\epsilon$ with $x \in \mathbb{R}$ and small $\epsilon$, we have:

$$\boldsymbol{G}_S(x - i\epsilon) = \sum_{k=1}^{2N} \frac{x + i\epsilon}{(x - \tilde{\gamma}_k)^2 + \epsilon^2} \mathbf{s}_k \mathbf{s}_k^\mathsf{T} + \frac{x + i\epsilon}{x^2 + \epsilon^2} \sum_{k=2N+1}^{N+M} \mathbf{s}_k \mathbf{s}_k^\mathsf{T}$$

where $\tilde{\gamma}_k$ are the eigenvalues of $S$, which are in fact the (signed) singular values of $S$, $\tilde{\gamma}_1 = \gamma_1, \ldots, \tilde{\gamma}_N = \gamma_N, \tilde{\gamma}_{N+1} = -\gamma_1, \ldots, \tilde{\gamma}_{2N} = -\gamma_N$.

Define the vectors $\tilde{\boldsymbol{x}}_i = [\boldsymbol{x}_i^\mathsf{T}, \mathbf{0}_M]^\mathsf{T}$ for $\boldsymbol{x}_i$ eigenvectors of $X$. We have

$$\tilde{\boldsymbol{x}}_i^\mathsf{T} \big( \operatorname{Im} \boldsymbol{G}_S(x - i\epsilon) \big) \tilde{\boldsymbol{x}}_i = \sum_{k=1}^{2N} \frac{\epsilon}{(x - \tilde{\gamma}_k)^2 + \epsilon^2} \big( \tilde{\boldsymbol{x}}_i^\mathsf{T} \mathbf{s}_k \big)^2 + \frac{\epsilon}{x^2 + \epsilon^2} \sum_{k=2N+1}^{N+M} \big( \tilde{\boldsymbol{x}}_i^\mathsf{T} \mathbf{s}_k \big)^2 \quad (16)$$

Given the structure of $\mathbf{s}_k$'s in (15), $\big( \tilde{\boldsymbol{x}}_i^\mathsf{T} \mathbf{s}_j \big)^2 = \frac{1}{2} \big( \boldsymbol{x}_i^\mathsf{T} \boldsymbol{u}_j \big)^2 = \big( \tilde{\boldsymbol{x}}_i^\mathsf{T} \mathbf{s}_{j+N} \big)^2$ for $1 \le j \le N$, and the second sum in (16) is zero. We assume that in the limit of large N this quantity concentrates on $O_X(\gamma_j, \lambda_i)$ and depends only on the singular values and eigenvalue pairs $(\gamma_j, \lambda_i)$. We thus have:

$$\tilde{\boldsymbol{x}}_i^\mathsf{T} \big( \operatorname{Im} \boldsymbol{G}_S(x - i\epsilon) \big) \tilde{\boldsymbol{x}}_i \xrightarrow{N \to \infty} \int_{\mathbb{R}} \frac{\epsilon}{(x - t)^2 + \epsilon^2} O_X(t, \lambda_i) \bar{\mu}_S(t) \, dt \quad (17)$$

where the overlap function $O_X(t, \lambda_i)$ is extended (continuously) to arbitrary values within the support of $\bar{\mu}_S$ (the symmetrized limiting singular value distribution of $S$) with the property that $O_X(t, \lambda_i) = O_X(-t, \lambda_i)$ for $t \in \operatorname{supp}(\mu_S)$. Sending $\epsilon \to 0$, we find

$$\tilde{\boldsymbol{x}}_i^\mathsf{T} \big( \operatorname{Im} \boldsymbol{G}_S(x - i\epsilon) \big) \tilde{\boldsymbol{x}}_i \to \pi \bar{\mu}_S(x) O_X(x, \lambda_i) \quad (18)$$

This is a crucial relation as it allows us to study the overlap by means of the resolvent of $S$. In the next section, we establish a connection between this resolvent and the signal $X$, which enables us to determine the optimal eigenvalues values $\widehat{\xi}_{xi}^*$ in terms of the singular values of $S$.

## 5.2 Resolvent relation

To derive the resolvent relation between $S$ and $X$, we fix the matrix $X$ and consider the model

$$S = X U_1 Y V_1^\mathsf{T} + U_2 W V_2^\mathsf{T}$$

with $Y, W \in \mathbb{R}^{N \times M}$ fixed matrices with limiting singular value distribution $\mu_Y, \mu_W$, and $U_1, U_2 \in \mathbb{R}^{N \times N}, V_1, V_2 \in \mathbb{R}^{M \times M}$ independent random Haar matrices. Indeed, if we substitute the SVD of the matrices $Y, W$ in model (1) we find the latter model. Now, the average over the singular vectors of $S$ (with fixed $X$) is equivalent to the average over the matrices $U_1, U_2, V_1, V_2$. In appendix C.1, using the Replica trick, we derive the following relation in the limit $N \to \infty$:

$$\langle \boldsymbol{G}_S(z) \rangle = \left[ \begin{array}{cc} \zeta_3^{-1} \boldsymbol{G}_{X^2} \big( \frac{z - \zeta_1}{\zeta_3} \big) & \mathbf{0} \\ \mathbf{0} & (z - \zeta_2)^{-1} \boldsymbol{I}_M \end{array} \right] \quad (19)$$

with $\zeta_1, \zeta_2, \zeta_3$ satisfying set of equations (41). $\langle . \rangle$ is the expectation w.r.t. the singular vectors of $S$ (or equivalently over $U_1, U_2, V_1, V_2$), and $G_{X^2}$ is the resolvent of $X^2$. As stated earlier, we assume that the resolvent $G_S(z)$ concentrates in the limit $N \to \infty$, therefore we drop the brackets in the following computation.

## 5.3 Overlaps and optimal eigenvalues

From (18), (19), we find:

$$\begin{aligned} O_X(\gamma, \lambda_i) &\approx \frac{1}{\pi \bar{\mu}_S(\gamma)} \operatorname{Im} \lim_{z \to \gamma - i0^+} \boldsymbol{x}_i^\mathsf{T} \zeta_3^{-1} \boldsymbol{G}_{X^2} \Big( \frac{z - \zeta_1}{\zeta_3} \Big) \boldsymbol{x}_i \\ &= \frac{1}{\pi \bar{\mu}_S(\gamma)} \operatorname{Im} \lim_{z \to \gamma - i0^+} \frac{1}{z - \zeta_1 - \zeta_3 \lambda_i^2} \end{aligned} \quad (20)$$

In Fig. 4 we illustrate that the theoretical predictions (20) are in good agreement with numerical simulations for a particular case of $X$ a Wigner matrix, and $Y, W$ with i.i.d. Gaussian entries.

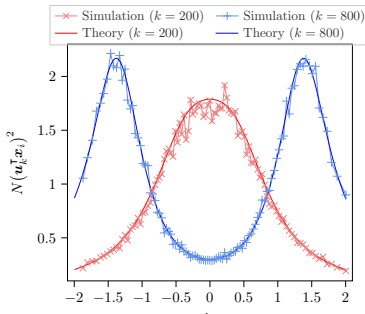

Figure 4: Comparison of the theoretical prediction (20) of the rescaled overlap with the numerical simulation. The rescaled overlap between 200-th and 800-th left singular vector of $S$ and the eigenvectors of $X$ is illustrated. $X = X^{\mathsf{T}} \in \mathbb{R}^{N \times N}$ has i.i.d. Gaussian entries with variance $1/\sqrt{N}$ and is fixed. Both $Y$ and $Z$ are $N \times M$ matrices with i.i.d. Gaussian entries of variance $1/N$. The simulation results are average of 1000 experiments with fixed $X$, and $N = 1000, M = 2000$. Some of the simulation points are dropped for clarity.
One can see that the overlap is an even function of eigenvalues $\lambda_i$, so the optimal eigenvalues $\xi^*_{xi} = \sum_{j=1}^{N} \lambda_j \left( u^{\mathsf{T}}_i x_j \right)^2$ are all zero, as discussed in remark 2.

Once we have the overlap, we can compute the optimal eigenvalues to be

$$\widehat{\xi}^*_{xi} \approx \frac{1}{N} \sum_{j=1}^{N} \lambda_j O_X(\gamma_i, \lambda_j) \approx \frac{1}{\pi \bar{\mu}_S(\gamma_i)} \operatorname{Im} \lim_{z \to \gamma_i - i0^+} \frac{1}{N} \sum_{j=1}^{N} \frac{\lambda_j}{z - \zeta_1 - \zeta_3 \lambda_j^2} \qquad (21)$$

With a bit of algebra, we find the estimator in (7) in the limit $N \to \infty$, see appendix C.2.

## 6   Conclusion

We studied the MF problem with extensive-rank hidden matrices, and proposed explicit (optimal) RIEs to recover the distinct factors. The model we considered, although limited, is the first analytically solvable model in this challenging regime, and we believe it paves the way for future investigations. Extending the methodology developed here to the general case where both matrices can be non-symmetric is an interesting research direction (note that in this case factors can be recovered up to rotations). Moreover, adapting RIEs to incorporate additional structures/ constraints on the signals is a problem with practical importance, that we leave for future investigations.

In general, the MF problem in the extensive-rank regime has not received an in-depth exploration compared to its counterpart in the low-rank regime. While substantial progress has been made in understanding and devising algorithms for low-rank matrices, the challenges posed by extensive-rank matrices remain relatively uncharted. Specifically, the study of general non-rotation invariant priors with i.i.d entries (e.g. a prior supported on the positive real line as in non-negative MF) stands out as an underdeveloped area from the theory side.

## Acknowledgments

We are thankful to Jean Barbier for interesting discussions on related problems. The work of F. P has been supported by Swiss National Science Foundation grant no 200021-204119.

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

# A Posterior mean estimator is in the RIE class

In this section, we show that for rotational invariant priors, the posterior mean estimator is inside the RIE class. For each of the estimators of $\boldsymbol{X}, \boldsymbol{Y}$, we present an equivalent definition of the RIE, then we show that posterior mean estimator satisfies this definition.

## A.1 X Estimator

**Lemma 1.** *Given the observation matrix $\boldsymbol{S}$, let $\hat{\boldsymbol{X}}(\boldsymbol{S})$ be an estimator of $\boldsymbol{X}$. Then $\hat{\boldsymbol{X}}(\boldsymbol{S})$ is a RIE if and only if for any orthogonal matrices $\boldsymbol{U} \in \mathbb{R}^{N \times N}, \boldsymbol{V} \in \mathbb{R}^{M \times M}$:*

$$\hat{\boldsymbol{X}}(\boldsymbol{U}\boldsymbol{S}\boldsymbol{V}^\mathsf{T}) = \boldsymbol{U}\hat{\boldsymbol{X}}(\boldsymbol{S})\boldsymbol{U}^\mathsf{T} \tag{22}$$

*Proof.* If $\hat{\boldsymbol{X}}(\boldsymbol{S})$ is a RIE, then the property (22) clearly follows from the definition (2). Now we turn to the converse.

Suppose that an estimator $\hat{\boldsymbol{X}}(\boldsymbol{S})$ satisfies (22). First, we show that if the observation matrix is diagonal, then the estimator is also diagonal. Consider the observation matrix to be $\boldsymbol{S}^{\mathrm{diag}} = \left[\begin{array}{c|c} \mathrm{diag}(\mathrm{s}_1, \ldots, \mathrm{s_N}) & \boldsymbol{0}_{N \times (M-N)} \end{array}\right]$. Let $\boldsymbol{I}_k^- \in \mathbb{R}^{N \times N}, \boldsymbol{J}_k^- \in \mathbb{R}^{M \times M}$ be diagonal matrices with diagonal entries all one except the $k$-th entry which is $-1$. Note that for $1 \le k \le N$, we have $\boldsymbol{S}^{\mathrm{diag}} = \boldsymbol{I}_k^- \boldsymbol{S}^{\mathrm{diag}} \boldsymbol{J}_k^-$. Moreover, matrices $\boldsymbol{I}_k^-, \boldsymbol{J}_k^-$ are indeed orthogonal. For any $1 \le k \le N$, from the property we have:

$$\hat{\boldsymbol{X}}(\boldsymbol{S}^{\mathrm{diag}}) = \hat{\boldsymbol{X}}(\boldsymbol{I}_k^- \boldsymbol{S}^{\mathrm{diag}} \boldsymbol{J}_k^-) = \boldsymbol{I}_k^- \hat{\boldsymbol{X}}(\boldsymbol{S}^{\mathrm{diag}}) \boldsymbol{I}_k^- \tag{23}$$

This implies that all entries on the $k$-th row and $k$-th column of $\hat{\boldsymbol{X}}(\boldsymbol{S}^{\mathrm{diag}})$ are zero except the $k$-th entry on the diagonal. Since this holds for any $k$, we conclude that $\hat{\boldsymbol{X}}(\boldsymbol{S}^{\mathrm{diag}})$ is diagonal.

Now, for a given general observation matrix with SVD $\boldsymbol{S} = \boldsymbol{U}_S \boldsymbol{\Gamma} \boldsymbol{V}_S^\mathsf{T}$, put $\boldsymbol{U} = \boldsymbol{U}_S^\mathsf{T}, \boldsymbol{V} = \boldsymbol{V}_S^\mathsf{T}$ in the property (22). We have:

$$\hat{\boldsymbol{X}}(\boldsymbol{\Gamma}) = \boldsymbol{U}_S^\mathsf{T} \hat{\boldsymbol{X}}(\boldsymbol{S}) \boldsymbol{U}_S$$

From the argument above, the matrix on the lhs is diagonal. Consequently, the matrix $\boldsymbol{U}_S^\mathsf{T} \hat{\boldsymbol{X}}(\boldsymbol{S}) \boldsymbol{U}_S$ is diagonal which implies that the columns of $\boldsymbol{U}_S$ are eigenvectors of $\hat{\boldsymbol{X}}(\boldsymbol{S})$. Therefore, $\hat{\boldsymbol{X}}(\boldsymbol{S})$ is a RIE. $\square$

Now, we prove that the posterior mean estimator $\hat{\boldsymbol{X}}^*(\boldsymbol{S}) = \mathbb{E}[\boldsymbol{X}|\boldsymbol{S}]$ has the property (22), and therefore belongs to the RIE class. For simplicity, we drop the SNR factor $\sqrt{\kappa}$. For any orthogonal matrices $\boldsymbol{U} \in \mathbb{R}^{N \times N}, \boldsymbol{V} \in \mathbb{R}^{M \times M}$, we have:

$$\begin{aligned}
\mathbb{E}[\boldsymbol{X}|\boldsymbol{U}\boldsymbol{S}\boldsymbol{V}^\mathsf{T}] &= \frac{\int d\boldsymbol{Y}\, d\tilde{\boldsymbol{X}}\, \tilde{\boldsymbol{X}}\, P_X(\tilde{\boldsymbol{X}}) P_Y(\boldsymbol{Y}) P_W(\boldsymbol{U}\boldsymbol{S}\boldsymbol{V}^\mathsf{T} - \tilde{\boldsymbol{X}}\boldsymbol{Y})}{\int d\boldsymbol{Y}\, d\tilde{\boldsymbol{X}}\, P_X(\tilde{\boldsymbol{X}}) P_Y(\boldsymbol{Y}) P_W(\boldsymbol{U}\boldsymbol{S}\boldsymbol{V}^\mathsf{T} - \tilde{\boldsymbol{X}}\boldsymbol{Y})} \\
&\overset{(a)}{=} \frac{\int d\boldsymbol{Y}\, d\tilde{\boldsymbol{X}}\, \boldsymbol{U}\tilde{\boldsymbol{X}}\boldsymbol{U}^\mathsf{T} P_X(\tilde{\boldsymbol{X}}) P_Y(\boldsymbol{Y}) P_W(\boldsymbol{U}\boldsymbol{S}\boldsymbol{V}^\mathsf{T} - \boldsymbol{U}\tilde{\boldsymbol{X}}\boldsymbol{U}^\mathsf{T}\boldsymbol{Y})}{\int d\boldsymbol{Y}\, d\tilde{\boldsymbol{X}}\, P_X(\tilde{\boldsymbol{X}}) P_Y(\boldsymbol{Y}) P_W(\boldsymbol{U}\boldsymbol{S}\boldsymbol{V}^\mathsf{T} - \boldsymbol{U}\tilde{\boldsymbol{X}}\boldsymbol{U}^\mathsf{T}\boldsymbol{Y})} \\
&\overset{(b)}{=} \frac{\int d\boldsymbol{Y}\, d\tilde{\boldsymbol{X}}\, \boldsymbol{U}\tilde{\boldsymbol{X}}\boldsymbol{U}^\mathsf{T} P_X(\tilde{\boldsymbol{X}}) P_Y(\boldsymbol{Y}) P_W(\boldsymbol{U}\boldsymbol{S}\boldsymbol{V}^\mathsf{T} - \boldsymbol{U}\tilde{\boldsymbol{X}}\boldsymbol{U}^\mathsf{T}\boldsymbol{U}\boldsymbol{Y}\boldsymbol{V}^\mathsf{T})}{\int d\boldsymbol{Y}\, d\tilde{\boldsymbol{X}}\, P_X(\tilde{\boldsymbol{X}}) P_Y(\boldsymbol{Y}) P_W(\boldsymbol{U}\boldsymbol{S}\boldsymbol{V}^\mathsf{T} - \boldsymbol{U}\tilde{\boldsymbol{X}}\boldsymbol{U}^\mathsf{T}\boldsymbol{U}\boldsymbol{Y}\boldsymbol{V}^\mathsf{T})} \\
&\overset{(c)}{=} \boldsymbol{U}\left\{\frac{\int d\boldsymbol{Y}\, d\tilde{\boldsymbol{X}}\, \tilde{\boldsymbol{X}}\, P_X(\tilde{\boldsymbol{X}}) P_Y(\boldsymbol{Y}) P_W(\boldsymbol{S} - \tilde{\boldsymbol{X}}\boldsymbol{Y})}{\int d\boldsymbol{Y}\, d\tilde{\boldsymbol{X}}\, P_X(\tilde{\boldsymbol{X}}) P_Y(\boldsymbol{Y}) P_W(\boldsymbol{S} - \tilde{\boldsymbol{X}}\boldsymbol{Y})}\right\}\boldsymbol{U}^\mathsf{T} \\
&= \boldsymbol{U}\mathbb{E}[\boldsymbol{X}|\boldsymbol{S}]\boldsymbol{U}^\mathsf{T}
\end{aligned}$$

where in (a), we changed variables $\tilde{\boldsymbol{X}} \to \boldsymbol{U}\tilde{\boldsymbol{X}}\boldsymbol{U}^\mathsf{T}$, used $|\det \boldsymbol{U}| = 1$, and rotational invariance of $P_X$, $P_X(\tilde{\boldsymbol{X}}) = P_X(\boldsymbol{U}\tilde{\boldsymbol{X}}\boldsymbol{U}^\mathsf{T})$. In (b), we changed variables $\boldsymbol{Y} \to \boldsymbol{U}\boldsymbol{Y}\boldsymbol{V}^\mathsf{T}$, used $|\det \boldsymbol{U}| = |\det \boldsymbol{V}| = 1$, and bi-rotational invariance of $P_Y$, $P_Y(\boldsymbol{Y}) = P_Y(\boldsymbol{U}\boldsymbol{Y}\boldsymbol{V}^\mathsf{T})$. In (c), we used the bi-rotational invariance property of $P_W$, namely $P_W(\boldsymbol{U}\boldsymbol{S}\boldsymbol{V}^\mathsf{T} - \boldsymbol{U}\tilde{\boldsymbol{X}}\boldsymbol{Y}\boldsymbol{V}^\mathsf{T}) = P_W(\boldsymbol{S} - \tilde{\boldsymbol{X}}\boldsymbol{Y})$.

## A.2 Y Estimator

**Lemma 2.** *Given the observation matrix $S$, let $\hat{Y}(S)$ be an estimator for $Y$. Then $\hat{Y}(S)$ is a RIE if and only if for any orthogonal matrices $U \in \mathbb{R}^{N \times N}, V \in \mathbb{R}^{M \times M}$:*

$$\hat{Y}(USV^\intercal) = U\hat{Y}(S)V^\intercal \tag{24}$$

*Proof.* If $\hat{Y}(S)$ is a RIE, then this property clearly follows from the definition (4). Let us now show the converse.

Suppose that an estimator $\hat{Y}(S)$ satisfies (24). First, we show that if the observation matrix is diagonal, then the estimator is also diagonal. Consider the observation matrix to be $S^{\mathrm{diag}} = \left[ \ \mathrm{diag}(s_1, \ldots, s_N) \ | \ \mathbf{0}_{N \times (M-N)} \ \right]$. Let $I_k^- \in \mathbb{R}^{N \times N}, J_k^- \in \mathbb{R}^{M \times M}$ be diagonal matrices with diagonal entries all one except the $k$-th entry which is $-1$. Note that for $1 \leq k \leq N$, we have $S^{\mathrm{diag}} = I_k^- S^{\mathrm{diag}} J_k^-$. Moreover, matrices $I_k^-, J_k^-$ are indeed orthogonal. For any $1 \leq k \leq N$, from the property we have:

$$\hat{Y}(S^{\mathrm{diag}}) = \hat{Y}(I_k^- S^{\mathrm{diag}} J_k^-) = I_k^- \hat{Y}(S^{\mathrm{diag}}) J_k^- \tag{25}$$

This implies that all entries on the $k$-th row and $k$-th column of $\hat{Y}(S^{\mathrm{diag}})$ is zero except the $k$-th entry on the diagonal. Since this holds for any $k$, we conclude that $\hat{Y}(S^{\mathrm{diag}})$ is diagonal.

Now, for a given general observation matrix $S = U_S \Gamma V_S^\intercal$, put $U = U_S^\intercal, V = V_S^\intercal$ in the property (24). We have:

$$\hat{Y}(\Gamma) = U_S^\intercal \hat{Y}(S) V_S$$

From the argument above, the matrix on the lhs is diagonal. Consequently, the matrix $U_S^\intercal \hat{Y}(S) V_S$ is diagonal which implies that the columns of $U_S, V_S$ are the left and right singular vectors of $\hat{Y}(S)$. Therefore, $\hat{Y}(S)$ is a RIE. $\qquad\square$

Now, we prove that the posterior mean estimator $\hat{Y}^*(S) = \mathbb{E}[Y|S]$ has the property (24), and it is inside the RIE class. For simplicity, we drop the SNR factor $\sqrt{\kappa}$. For any orthogonal matrices $U \in \mathbb{R}^{N \times N}, V \in \mathbb{R}^{M \times M}$, we have:

$$
\begin{aligned}
\mathbb{E}[Y|USV^\intercal] &= \frac{\int d X\, d\tilde{Y}\ \tilde{Y}\, P_X(X) P_Y(\tilde{Y}) P_W(USV^\intercal - X\tilde{Y})}{\int d X\, d\tilde{Y}\, P_X(X) P_Y(\tilde{Y}) P_W(USV^\intercal - X\tilde{Y})} \\
&\stackrel{(a)}{=} \frac{\int d X\, d\tilde{Y}\ U\tilde{Y}V^\intercal\, P_X(X) P_Y(\tilde{Y}) P_W(USV^\intercal - XU\tilde{Y}V^\intercal)}{\int d X\, d\tilde{Y}\, P_X(X) P_Y(\tilde{Y}) P_W(USV^\intercal - XU\tilde{Y}V^\intercal)} \\
&\stackrel{(b)}{=} \frac{\int d X\, d\tilde{Y}\ U\tilde{Y}V^\intercal\, P_X(X) P_Y(\tilde{Y}) P_W(USV^\intercal - UXU^\intercal U\tilde{Y}V^\intercal)}{\int d X\, d\tilde{Y}\, P_X(X) P_Y(\tilde{Y}) P_W(USV^\intercal - UXU^\intercal U\tilde{Y}V^\intercal)} \\
&\stackrel{(c)}{=} U\left\{ \frac{\int d X\, d\tilde{Y}\ \tilde{Y}\, P_X(X) P_Y(\tilde{Y}) P_W(S - X\tilde{Y})}{\int d X\, d\tilde{Y}\, P_X(X) P_Y(\tilde{Y}) P_W(S - X\tilde{Y})} \right\} V^\intercal \\
&= U\mathbb{E}[Y|S]V^\intercal
\end{aligned}
$$

where in (a), we changed variables $\tilde{Y} \to U\tilde{Y}V^\intercal$, used $|\det U| = |\det V| = 1$, and bi-rotational invariance of $P_Y$, $P_Y(\tilde{Y}) = P_Y(U\tilde{Y}V^\intercal)$. In (b), we changed variables $X \to UXU^\intercal$, used $|\det U| = 1$, and rotational invariance of $P_X$, $P_X(X) = P_X(UXU^\intercal)$. In (c), we used the bi-rotational invariance property of $P_W$, namely $P_W(USV^\intercal - UX\tilde{Y}V^\intercal) = P_W(S - X\tilde{Y})$.

# B  The replica method for deriving the resolvent relation

In this section we present the replica method used to obtain the resolvent relation. For simplicity of notation we use $G(z) \equiv G_S(z)$ for the resolvent of a random matrix $S$.

First, we express the entries of the resolvent $\boldsymbol{G}(z)$ using the Gaussian integral representation of an inverse matrix [60]:

$$
G_{ij}(z) = \sqrt{\frac{1}{(2\pi)^{N+M}\det(z\boldsymbol{I}-\boldsymbol{S})}} \int \Big( \prod_{k=1}^{M+N} d\eta_k \Big) \eta_i \eta_j \exp\Big\{ -\frac{1}{2}\boldsymbol{\eta}^{\mathsf{T}}(z\boldsymbol{I}-\boldsymbol{S})\boldsymbol{\eta} \Big\}
$$

$$
= \frac{\int \Big( \prod_{k=1}^{M+N} d\eta_k \Big) \eta_i \eta_j \exp\Big\{ -\frac{1}{2}\boldsymbol{\eta}^{\mathsf{T}}(z\boldsymbol{I}-\boldsymbol{S})\boldsymbol{\eta} \Big\}}{\int \Big( \prod_{k=1}^{M+N} d\eta_k \Big) \exp\Big\{ -\frac{1}{2}\boldsymbol{\eta}^{\mathsf{T}}(z\boldsymbol{I}-\boldsymbol{S})\boldsymbol{\eta} \Big\}} \tag{26}
$$

For $z$ not close to the real axis, the resolvent is expected to exhibit self-averaging behavior in the limit of large N, meaning that it will not depend on the particular matrix realization. Thus, we can examine the resolvent $\boldsymbol{G}_{\mathcal{S}}(z)$ by analyzing its ensemble average, denoted by $\langle . \rangle$ in the following.

$$
\langle G_{ij}(z) \rangle = \Big\langle \frac{1}{\mathcal{Z}} \int \Big( \prod_{k=1}^{M+N} d\eta_k \Big) \eta_i \eta_j \exp\Big\{ -\frac{1}{2}\boldsymbol{\eta}^{\mathsf{T}}(z\boldsymbol{I}-\boldsymbol{S})\boldsymbol{\eta} \Big\} \Big\rangle \tag{27}
$$

where $\mathcal{Z}$ is the denominator in (26). Computing the average is, in general, non-trivial. However, the replica method provides us with a technique to overcome this issue by employing the following identity:

$$
\langle G_{ij}(z) \rangle = \lim_{n \to 0} \Big\langle \mathcal{Z}^{n-1} \int \Big( \prod_{k=1}^{M+N} d\eta_k \Big) \eta_i \eta_j \exp\Big\{ -\frac{1}{2}\boldsymbol{\eta}^{\mathsf{T}}(z\boldsymbol{I}-\boldsymbol{S})\boldsymbol{\eta} \Big\} \Big\rangle
$$

$$
= \lim_{n \to 0} \Big\langle \int \Big( \prod_{k=1}^{M+N} \prod_{\tau=1}^{n} d\eta_k^{(\tau)} \Big) \eta_i^{(1)} \eta_j^{(1)} \exp\Big\{ -\frac{1}{2} \sum_{\tau=1}^{n} \boldsymbol{\eta}^{(\tau)\mathsf{T}}(z\boldsymbol{I}-\boldsymbol{S})\boldsymbol{\eta}^{(\tau)} \Big\} \Big\rangle \tag{28}
$$

So, the problem now is reduced to computation of an average over $n$ copies (or replicas) of the initial system (26). After computing the average value (the bracket) in (28), we can perform an analytical continuation of the result to real values of $n$ and then take the limit $n \to 0$. Throughout, we assume as is common in the replica method, that the analytical continuation can be done with only $n$ different sets of points. Of course, this is a totally uncontrolled step that comes with no guarantees.

## C   Derivation of the RIE for $\boldsymbol{X}$

In this section, we consider estimating $\boldsymbol{X}$, and treat both $\boldsymbol{Y}$ and $\boldsymbol{W}$ as noise. We consider $\boldsymbol{X}$ to be fixed, and the observation model:

$$
\boldsymbol{S} = \boldsymbol{X}\boldsymbol{U}_1\boldsymbol{Y}\boldsymbol{V}_1^{\mathsf{T}} + \boldsymbol{U}_2\boldsymbol{W}\boldsymbol{V}_2^{\mathsf{T}} \tag{29}
$$

where $\boldsymbol{Y}, \boldsymbol{W} \in \mathbb{R}^{N\times M}$ are fixed matrices with limiting singular value distribution $\mu_Y, \mu_W$, and $\boldsymbol{U}_1, \boldsymbol{U}_2 \in \mathbb{R}^{N\times N}, \boldsymbol{V}_1, \boldsymbol{V}_2 \in \mathbb{R}^{M\times M}$ are independent random Haar matrices.

Construct the hermitization $\boldsymbol{S} \in \mathbb{R}^{(N+M)\times(N+M)}$ from $S$ as

$$
\boldsymbol{S} = \begin{bmatrix} \boldsymbol{0}_{N\times N} & \boldsymbol{S} \\ \boldsymbol{S}^{\mathsf{T}} & \boldsymbol{0}_{M\times M} \end{bmatrix}
$$

For simplicity of notation, we use $\boldsymbol{T} \equiv \boldsymbol{X}\boldsymbol{U}_1\boldsymbol{Y}\boldsymbol{V}_1^{\mathsf{T}}, \boldsymbol{\mathcal{T}} \in \mathbb{R}^{(N+M)\times(N+M)}$ the hermitization of $\boldsymbol{T}$, and $\widetilde{\boldsymbol{\mathcal{W}}}$ denotes the hermitization of the matrix $\boldsymbol{U}_2\boldsymbol{W}\boldsymbol{V}_2^{\mathsf{T}}$.

## C.1 Resolvent relation

We want to find a relation between $\boldsymbol{G}(z) \equiv \boldsymbol{G}_{\mathcal{S}}(z)$, and the signal matrix $\boldsymbol{X}$. From (28), we have

$$
\langle G_{ij}(z) \rangle = \lim_{n \to \infty} \int \Big( \prod_{k=1}^{N+M} \prod_{\tau=1}^{n} d\eta_k^{(\tau)} \Big) \eta_i^{(1)} \eta_j^{(1)} \Big\langle \exp \Big\{ - \frac{1}{2} \sum_{\tau=1}^{n} \boldsymbol{\eta}^{(\tau)\mathsf{T}} (z\boldsymbol{I} - \boldsymbol{\mathcal{S}}) \boldsymbol{\eta}^{(\tau)} \Big\} \Big\rangle_{\boldsymbol{U}_1, \boldsymbol{U}_2, \boldsymbol{V}_1, \boldsymbol{V}_2}
$$

$$
= \lim_{n \to \infty} \int \Big( \prod_{k=1}^{N+M} \prod_{\tau=1}^{n} d\eta_k^{(\tau)} \Big) \eta_i^{(1)} \eta_j^{(1)} \exp \Big\{ - \frac{z}{2} \sum_{\tau=1}^{n} \boldsymbol{\eta}^{(\tau)\mathsf{T}} \boldsymbol{\eta}^{(\tau)} \Big\}
$$

$$
\times \Big\langle \exp \Big\{ \frac{1}{2} \sum_{\tau=1}^{n} \boldsymbol{\eta}^{(\tau)\mathsf{T}} \boldsymbol{\mathcal{T}} \boldsymbol{\eta}^{(\tau)} \Big\} \Big\rangle_{\boldsymbol{U}_1, \boldsymbol{V}_1} \Big\langle \exp \Big\{ \frac{1}{2} \sum_{\tau=1}^{n} \boldsymbol{\eta}^{(\tau)\mathsf{T}} \widetilde{\boldsymbol{\mathcal{W}}} \boldsymbol{\eta}^{(\tau)} \Big\} \Big\rangle_{\boldsymbol{U}_2, \boldsymbol{V}_2}
$$

$$(30)$$

Split each replica $\boldsymbol{\eta}^{(\tau)}$ into two vectors $\boldsymbol{a}^{(\tau)} \in \mathbb{R}^N, \boldsymbol{b}^{(\tau)} \in \mathbb{R}^M, \boldsymbol{\eta}^{(\tau)} = \begin{bmatrix} \boldsymbol{a}^{(\tau)} \\ \boldsymbol{b}^{(\tau)} \end{bmatrix}$. The exponent in the first bracket in (30) can be written as:

$$
\boldsymbol{\eta}^{(\tau)\mathsf{T}} \boldsymbol{\mathcal{T}} \boldsymbol{\eta}^{(\tau)} = \boldsymbol{a}^{(\tau)\mathsf{T}} \boldsymbol{X} \boldsymbol{U}_1 \boldsymbol{Y} \boldsymbol{V}_1^{\mathsf{T}} \boldsymbol{b}^{(\tau)} + \boldsymbol{b}^{(\tau)\mathsf{T}} \boldsymbol{V}_1 \boldsymbol{Y}^{\mathsf{T}} \boldsymbol{U}_1^{\mathsf{T}} \boldsymbol{X} \boldsymbol{a}^{(\tau)}
$$

$$
= 2 \boldsymbol{a}^{(\tau)\mathsf{T}} \boldsymbol{X} \boldsymbol{U}_1 \boldsymbol{Y} \boldsymbol{V}_1^{\mathsf{T}} \boldsymbol{b}^{(\tau)} \tag{31}
$$

$$
= 2 \operatorname{Tr} \boldsymbol{b}^{(\tau)} \boldsymbol{a}^{(\tau)\mathsf{T}} \boldsymbol{X} \boldsymbol{U}_1 \boldsymbol{Y} \boldsymbol{V}_1^{\mathsf{T}}
$$

Using the formula for the rectangular spherical integral [22] (see Theorem 2 in H.1), we find:

$$
\Big\langle \exp \Big\{ \sum_{\tau=1}^{n} \operatorname{Tr} \boldsymbol{b}^{(\tau)} \boldsymbol{a}^{(\tau)\mathsf{T}} \boldsymbol{X} \boldsymbol{U}_1 \boldsymbol{Y} \boldsymbol{V}_1^{\mathsf{T}} \Big\} \Big\rangle_{\boldsymbol{U}_1, \boldsymbol{V}_1} \approx \exp \Big\{ \frac{N}{2} \sum_{\tau=1}^{n} \mathcal{Q}_{\mu_Y}^{(\alpha)} \Big( \frac{1}{NM} \| \boldsymbol{X} \boldsymbol{a}^{(\tau)} \|^2 \| \boldsymbol{b}^{(\tau)} \|^2 \Big) \Big\}
$$

$$(32)$$

with $\mathcal{Q}_{\mu_Y}^{(\alpha)}(x) = \int_0^x \frac{\mathcal{C}_{\mu_Y}^{(\alpha)}(t)}{t} dt$. In (32), we used that $\boldsymbol{b}^{(\tau)} \boldsymbol{a}^{(\tau)\mathsf{T}} \boldsymbol{X}$ is a rank-one matrix with non-zero singular value $\| \boldsymbol{b}^{(\tau)} \| \| \boldsymbol{X} \boldsymbol{a}^{(\tau)} \|$.

Similarly, for the second bracket in (30) we can write:

$$
\boldsymbol{\eta}^{(\tau)\mathsf{T}} \widetilde{\boldsymbol{\mathcal{W}}} \boldsymbol{\eta}^{(\tau)} = \boldsymbol{a}^{(\tau)\mathsf{T}} \boldsymbol{U}_2 \boldsymbol{W} \boldsymbol{V}_2^{\mathsf{T}} \boldsymbol{b}^{(\tau)} + \boldsymbol{b}^{(\tau)\mathsf{T}} \boldsymbol{V}_2 \boldsymbol{W}^{\mathsf{T}} \boldsymbol{U}_2^{\mathsf{T}} \boldsymbol{a}^{(\tau)}
$$

$$
= 2 \boldsymbol{a}^{(\tau)\mathsf{T}} \boldsymbol{U}_2 \boldsymbol{W} \boldsymbol{V}_2^{\mathsf{T}} \boldsymbol{b}^{(\tau)} \tag{33}
$$

$$
= 2 \operatorname{Tr} \boldsymbol{b}^{(\tau)} \boldsymbol{a}^{(\tau)\mathsf{T}} \boldsymbol{U}_2 \boldsymbol{W} \boldsymbol{V}_2^{\mathsf{T}}
$$

which using the formula of rectangular spherical integrals, implies

$$
\Big\langle \exp \Big\{ \sum_{\tau=1}^{n} \operatorname{Tr} \boldsymbol{b}^{(\tau)} \boldsymbol{a}^{(\tau)\mathsf{T}} \boldsymbol{U}_2 \boldsymbol{W} \boldsymbol{V}_2^{\mathsf{T}} \Big\} \Big\rangle_{\boldsymbol{U}_2, \boldsymbol{V}_2} \approx \exp \Big\{ \frac{N}{2} \sum_{\tau=1}^{n} \mathcal{Q}_{\mu_W}^{(\alpha)} \Big( \frac{1}{NM} \| \boldsymbol{a}^{(\tau)} \|^2 \| \boldsymbol{b}^{(\tau)} \|^2 \Big) \Big\} \tag{34}
$$

From (30), (32), (34), we find:

$$
\langle G_{ij}(z) \rangle = \lim_{n \to \infty} \int \Big( \prod_{k=1}^{N+M} \prod_{\tau=1}^{n} d\eta_k^{(\tau)} \Big) \eta_i^{(1)} \eta_j^{(1)}
$$

$$
\times \exp \Big\{ - \frac{1}{2} \sum_{\tau=1}^{n} z \| \boldsymbol{\eta}^{(\tau)} \|^2 - N \mathcal{Q}_{\mu_Y}^{(\alpha)} \Big( \frac{\| \boldsymbol{X} \boldsymbol{a}^{(\tau)} \|^2 \| \boldsymbol{b}^{(\tau)} \|^2}{NM} \Big) - N \mathcal{Q}_{\mu_W}^{(\alpha)} \Big( \frac{\| \boldsymbol{a}^{(\tau)} \|^2 \| \boldsymbol{b}^{(\tau)} \|^2}{NM} \Big) \Big\}
$$

$$(35)$$

Now, we introduce delta functions $\delta\big(p_1^{(\tau)} - \frac{\| \boldsymbol{a}^{(\tau)} \|^2}{N}\big)$, $\delta\big(p_2^{(\tau)} - \frac{\| \boldsymbol{b}^{(\tau)} \|^2}{M}\big)$, and $\delta\big(p_3^{(\tau)} - \frac{\| \boldsymbol{X} \boldsymbol{a}^{(\tau)} \|^2}{N}\big)$, and using them, the integral in (35) can be written as (for brevity we drop the limit term):

$$
\langle G_{ij}(z) \rangle = \int \Big( \prod_{k=1}^{N+M} \prod_{\tau=1}^{n} d\eta_k^{(\tau)} \Big) \Big( \prod_{\tau=1}^{n} dp_1^{(\tau)} dp_2^{(\tau)} dp_3^{(\tau)} \Big) \eta_i^{(1)} \eta_j^{(1)}
$$

$$
\times \prod_{\tau=1}^{n} \delta\Big( p_1^{(\tau)} - \frac{\| \boldsymbol{a}^{(\tau)} \|^2}{N} \Big) \delta\Big( p_2^{(\tau)} - \frac{\| \boldsymbol{b}^{(\tau)} \|^2}{M} \Big) \delta\Big( p_3^{(\tau)} - \frac{\| \boldsymbol{X} \boldsymbol{a}^{(\tau)} \|^2}{N} \Big) \tag{36}
$$

$$
\times \exp \Big\{ - \frac{1}{2} \sum_{\tau=1}^{n} z \| \boldsymbol{\eta}^{(\tau)} \|^2 - N \mathcal{Q}_{\mu_Y}^{(\alpha)} (p_2^{(\tau)} p_3^{(\tau)}) - N \mathcal{Q}_{\mu_W}^{(\alpha)} (p_1^{(\tau)} p_2^{(\tau)}) \Big\}
$$

In the next step, we replace each delta with its Fourier transform, $\delta\left(p_1^\tau - \frac{1}{N}\|\boldsymbol{a}^\tau\|^2\right) \propto \int d\zeta_1^\tau \exp\left\{-\frac{N}{2}\zeta_1^\tau\left(p_1^\tau - \frac{1}{N}\|\boldsymbol{a}^\tau\|^2\right)\right\}$. After rearranging, we find:

$$\langle G_{ij}(z)\rangle \propto \int \Big(\prod_{\tau=1}^n dp_1^{(\tau)}\, dp_2^{(\tau)}\, dp_3^{(\tau)}\, d\zeta_1^{(\tau)}\, d\zeta_2^{(\tau)}\, d\zeta_3^{(\tau)}\Big)$$

$$\times \exp\left\{\frac{N}{2}\sum_{\tau=1}^n \mathcal{Q}_{\mu_Y}^{(\alpha)}(p_2^{(\tau)}p_3^{(\tau)}) + \mathcal{Q}_{\mu_W}^{(\alpha)}(p_1^{(\tau)}p_2^{(\tau)}) - \zeta_1^{(\tau)}p_1^{(\tau)} - \frac{1}{\alpha}\zeta_2^{(\tau)}p_2^{(\tau)} - \zeta_3^{(\tau)}p_3^{(\tau)}\right\}$$

$$\times \int \Big(\prod_{k=1}^{N+M}\prod_{\tau=1}^n d\eta_k^{(\tau)}\Big)\, \eta_i^{(1)}\eta_j^{(1)}$$

$$\times \exp\left\{-\frac{1}{2}\sum_{\tau=1}^n z\|\boldsymbol{\eta}^{(\tau)}\|^2 - \zeta_1^{(\tau)}\|\boldsymbol{a}^{(\tau)}\|^2 - \zeta_2^{(\tau)}\|\boldsymbol{b}^{(\tau)}\|^2 - \zeta_3^{(\tau)}\|\boldsymbol{X}\boldsymbol{a}^{(\tau)}\|^2\right\}$$

$$\tag{37}$$

The inner integral in (37) is a Gaussian integral, and can be written as:

$$\int \Big(\prod_{k=1}^{N+M}\prod_{\tau=1}^n d\eta_k^{(\tau)}\Big)\, \eta_i^{(1)}\eta_j^{(1)}$$

$$\times \exp\left\{\sum_{\tau=1}^n -\frac{1}{2}\boldsymbol{\eta}^{(\tau)\mathsf{T}}\begin{bmatrix} (z-\zeta_1^{(\tau)})\boldsymbol{I}_N - \zeta_3^{(\tau)}\boldsymbol{X}^2 & \boldsymbol{0} \\ \boldsymbol{0} & (z-\zeta_2^{(\tau)})\boldsymbol{I}_M \end{bmatrix}\boldsymbol{\eta}^{(\tau)}\right\}$$

$$\tag{38}$$

Denote the matrix in the exponent by $\boldsymbol{C}_X^{(\tau)}$. Its determinant reads:

$$\det \boldsymbol{C}_X^{(\tau)} = (z-\zeta_2^{(\tau)})^M \prod_{k=1}^N (z-\zeta_1^{(\tau)} - \zeta_3^{(\tau)}\lambda_k^2)$$

where $\lambda_k$'s are eigenvalues of $\boldsymbol{X}$. So replacing the formula for the Gaussian integrals, (37) can be written as:

$$\langle G_{ij}(z)\rangle \propto \int \Big(\prod_{\tau=1}^n dp_1^{(\tau)}\, dp_2^{(\tau)}\, dp_3^{(\tau)}\, d\zeta_1^{(\tau)}\, d\zeta_2^{(\tau)}\, d\zeta_3^{(\tau)}\Big)\left(\boldsymbol{C}_X^{(1)}{}^{-1}\right)_{ij}$$

$$\times \exp\left\{-\frac{Nn}{2}F_0^X(\boldsymbol{p}_1,\boldsymbol{p}_2,\boldsymbol{p}_3,\boldsymbol{\zeta}_1,\boldsymbol{\zeta}_2,\boldsymbol{\zeta}_3)\right\}$$

$$\tag{39}$$

with

$$F_0^X(\boldsymbol{p}_1,\boldsymbol{p}_2,\boldsymbol{p}_3,\boldsymbol{\zeta}_1,\boldsymbol{\zeta}_2,\boldsymbol{\zeta}_3) = \frac{1}{n}\sum_{\tau=1}^n\left[\frac{1}{N}\sum_{k=1}^N \ln(z-\zeta_1^{(\tau)} - \zeta_3^{(\tau)}\lambda_k^2) + \frac{1}{\alpha}\ln(z-\zeta_2^{(\tau)})\right.$$

$$\left. - \mathcal{Q}_{\mu_Y}^{(\alpha)}(p_2^{(\tau)}p_3^{(\tau)}) - \mathcal{Q}_{\mu_W}^{(\alpha)}(p_1^{(\tau)}p_2^{(\tau)}) + \zeta_1^{(\tau)}p_1^{(\tau)} + \frac{1}{\alpha}\zeta_2^{(\tau)}p_2^{(\tau)} + \zeta_3^{(\tau)}p_3^{(\tau)}\right]$$

$$\tag{40}$$

In the large $N$ limit, the integral in (39) can be computed using the saddle-points of the function $F_0^X$. In the evaluation of this integral, we use the *replica symmetric* ansatz that assumes a saddle-point of the form:

$$\forall \tau \in \{1,\cdots,n\}: \quad \begin{cases} p_1^\tau = p_1, & p_2^\tau = p_2, & p_3^\tau = p_3 \\ \zeta_1^\tau = \zeta_1, & \zeta_2^\tau = \zeta_2, & \zeta_3^\tau = \zeta_3 \end{cases}$$

The saddle point is a solution of the set of equations:

$$\begin{cases} \zeta_1^* = \frac{\mathcal{C}_{\mu_W}^{(\alpha)}(p_1^* p_2^*)}{p_1^*}, \quad \zeta_2^* = \frac{\alpha}{p_2^*}\left(\mathcal{C}_{\mu_W}^{(\alpha)}(p_1^* p_2^*) + \mathcal{C}_{\mu_Y}^{(\alpha)}(p_2^* p_3^*)\right), \quad \zeta_3^* = \frac{\mathcal{C}_{\mu_Y}^{(\alpha)}(p_2^* p_3^*)}{p_3^*} \\[2mm] p_1^* = \frac{1}{\zeta_3^*}\mathcal{G}_{\rho_{X^2}}\left(\frac{z-\zeta_1^*}{\zeta_3^*}\right), \quad p_2^* = \frac{1}{z-\zeta_2^*}, \quad p_3^* = \frac{z-\zeta_1^*}{\zeta_3^{*2}}\mathcal{G}_{\rho_{X^2}}\left(\frac{z-\zeta_1^*}{\zeta_3^*}\right) - \frac{1}{\zeta_3^*} \end{cases}$$

$$\tag{41}$$

Now, since the relation (39) and the solutions (41) hold for arbitrary indices $i, j$, we can state the relation in matrix form. The inverse of $C_X^{*-1}$, and the block structure of $G_S(z)$ are computed in sections H.2. From (111), (112) we have (for sufficiently large $N$):

$$\langle G_S(z)\rangle_{U_1,U_2,V_1,V_2} = \left\langle \begin{bmatrix} \frac{1}{z}I_N + \frac{1}{z}SG_{S^\mathsf{T}S}(z^2)S^\mathsf{T} & SG_{S^\mathsf{T}S}(z^2) \\ G_{S^\mathsf{T}S}^*(z^2)S^\mathsf{T} & zG_{S^\mathsf{T}S}(z^2) \end{bmatrix} \right\rangle$$

$$= \begin{bmatrix} \frac{1}{\zeta_3^*}G_{X^2}\left(\frac{z-\zeta_1^*}{\zeta_3^*}\right) & 0 \\ 0 & \frac{1}{z-\zeta_2^*}I_M \end{bmatrix} \tag{42}$$

With this relation, we proceed to simplify the equations (41).

The normalized trace of the upper-left blocks of $\langle G_S(z)\rangle_{U_1,U_2,V_1,V_2}$ is:

$$\frac{1}{N}\sum_{k=1}^N \left[\frac{1}{z} + \frac{1}{z}\frac{\gamma_k^2}{z^2 - \gamma_k^2}\right] = \frac{1}{z}\frac{1}{N}\sum_{k=1}^N \left[1 + \frac{\gamma_k^2}{z^2 - \gamma_k^2}\right]$$

$$= z\frac{1}{N}\sum_{k=1}^N \frac{1}{z^2 - \gamma_k^2} \tag{43}$$

$$= \frac{1}{2N}\sum_{k=1}^N \left[\frac{1}{z - \gamma_k} + \frac{1}{z + \gamma_k}\right] = \mathcal{G}_{\bar\mu_S}(z)$$

and the normalized trace of the upper-left block in $C_X^{*-1}$ is $\frac{1}{\zeta_3^*}\mathcal{G}_{\rho_{X^2}}\left(\frac{z-\zeta_1^*}{\zeta_3^*}\right) = p_1^*$. Therefore, we have $p_1^* = \mathcal{G}_{\bar\mu_S}(z)$.

The normalized trace of lower-right block of $\langle G_S(z)\rangle_{U_1,U_2,V_1,V_2}$ reads:

$$\frac{1}{M}z\left[\sum_{k=1}^N \frac{1}{z^2 - \gamma_k^2} + (M-N)\frac{1}{z^2}\right] = \frac{N}{M}\mathcal{G}_{\bar\mu_S}(z) + \frac{M-N}{M}\frac{1}{z} = \alpha\mathcal{G}_{\bar\mu_S}(z) + (1-\alpha)\frac{1}{z} \tag{44}$$

and the normalized trace of the lower-right block in $C_X^{*-1}$ is $\frac{1}{z-\zeta_2^*} = p_2^*$. Therefore, we have $p_2^* = \alpha\mathcal{G}_{\bar\mu_S}(z) + (1-\alpha)\frac{1}{z}$. Moreover, we also have that $\zeta_2^* = \alpha z\frac{z\mathcal{G}_{\bar\mu_S}(z)-1}{\alpha z\mathcal{G}_{\bar\mu_S}(z)+1-\alpha}$.

Therefore, the saddle point equations (41) can be rewritten in a simplified form, which does not involve $\rho_{X^2}$, as:

$$\begin{cases} \zeta_1^* = \frac{\mathcal{C}_{\mu_W}^{(\alpha)}(p_1^*p_2^*)}{p_1^*}, & \zeta_2^* = \alpha z\frac{z\mathcal{G}_{\bar\mu_S}(z)-1}{\alpha z\mathcal{G}_{\bar\mu_S}(z)+1-\alpha}, & \zeta_3^* = \frac{\mathcal{C}_{\mu_Y}^{(\alpha)}(p_2^*p_3^*)}{p_3^*} \\[2mm] p_1^* = \mathcal{G}_{\bar\mu_S}(z), & p_2^* = \alpha\mathcal{G}_{\bar\mu_S}(z) + (1-\alpha)\frac{1}{z}, & p_3^* = \frac{z-\zeta_1^*}{\zeta_3^*}\mathcal{G}_{\bar\mu_S}(z) - \frac{1}{\zeta_3^*} \end{cases} \tag{45}$$

Note that $\zeta_1^*, \zeta_2^*$ can be computed from the observation matrix, and we only need to find $\zeta_3^*$ satisfying the following equation:

$$(z - \zeta_1^*)\mathcal{G}_{\bar\mu_S}(z) - 1 = \mathcal{C}_{\mu_Y}^{(\alpha)}\left(\frac{1}{\zeta_3^*}\left[\alpha\mathcal{G}_{\bar\mu_S}(z) + \frac{1-\alpha}{z}\right]\left[(z-\zeta_1^*)\mathcal{G}_{\bar\mu_S}(z) - 1\right]\right) \tag{46}$$

## C.2 Overlaps and optimal eigenvalues

We restate the relation between the resolvent and the overlaps from the main text (18). For $\tilde{x}_i = [x_i^\mathsf{T}, 0_M]^\mathsf{T}$ with $x_i$ eigenvectors of $X$, we have:

$$\tilde{x}_i^\mathsf{T}\left(\mathrm{Im}\,G_S(x - i\epsilon)\right)\tilde{x}_i \approx \pi\bar\mu_S(x)O_X(x, \lambda_i) \tag{47}$$

From (47), (42), we find:

$$O_X(\gamma, \lambda_i) \approx \frac{1}{\pi\bar\mu_S(\gamma)}\,\mathrm{Im}\,\lim_{z\to\gamma-i0^+} x_i^\mathsf{T}\,\zeta_3^{*-1}G_{X^2}\left(\frac{z-\zeta_1^*}{\zeta_3^*}\right)x_i$$

$$= \frac{1}{\pi\bar\mu_S(\gamma)}\,\mathrm{Im}\,\lim_{z\to\gamma-i0^+} \frac{1}{z - \zeta_1^* - \zeta_3^*\lambda_i^2} \tag{48}$$

Once we have the overlap, we can compute the optimal eigenvalues from (14) in section 5. Note that, until now we had absorbed $\sqrt{\kappa}$ into $\boldsymbol{X}$. Therefore, we should use (48) with $O_X(\gamma, \sqrt{\kappa}\lambda_i)$. This leads to:

$$
\begin{aligned}
\widehat{\xi^*_{x\,i}} &\approx \frac{1}{N}\sum_{j=1}^{N}\lambda_j O_X(\gamma_i, \sqrt{\kappa}\lambda_j) \\[2mm]
&\approx \frac{1}{\pi\bar{\mu}_S(\gamma_i)}\operatorname{Im}\lim_{z\to\gamma_i-\mathrm{i}0^+}\frac{1}{N}\sum_{j=1}^{N}\frac{\lambda_j}{z-\zeta_1^*-\zeta_3^*\kappa\lambda_j^2} \\[2mm]
&= \frac{1}{\pi\bar{\mu}_S(\gamma_i)}\operatorname{Im}\lim_{z\to\gamma_i-\mathrm{i}0^+}\frac{1}{\kappa\zeta_3^*}\frac{1}{N}\sum_{j=1}^{N}\frac{\lambda_j}{\frac{z-\zeta_1^*}{\kappa\zeta_3^*}-\lambda_j^2} \\[2mm]
&= \frac{1}{\kappa\pi\bar{\mu}_S(\gamma_i)}\operatorname{Im}\lim_{z\to\gamma_i-\mathrm{i}0^+}\frac{1}{\zeta_3^*}\left(\frac{1}{2}\frac{1}{N}\sum_{j=1}^{N}\frac{1}{\sqrt{\frac{z-\zeta_1^*}{\kappa\zeta_3^*}}-\lambda_j}-\frac{1}{2}\frac{1}{N}\sum_{j=1}^{N}\frac{1}{\sqrt{\frac{z-\zeta_1^*}{\kappa\zeta_3^*}}+\lambda_j}\right) \\[2mm]
&\approx \frac{1}{\kappa\pi\bar{\mu}_S(\gamma_i)}\operatorname{Im}\lim_{z\to\gamma_i-\mathrm{i}0^+}\left\{\frac{1}{2}\frac{1}{\zeta_3^*}\mathcal{G}_{\rho_X}\left(\sqrt{\frac{z-\zeta_1^*}{\kappa\zeta_3^*}}\right)-\frac{1}{2}\frac{1}{\zeta_3^*}\mathcal{G}_{\rho-X}\left(\sqrt{\frac{z-\zeta_1^*}{\kappa\zeta_3^*}}\right)\right\} \\[2mm]
&= \frac{1}{2\kappa\pi\bar{\mu}_S(\gamma_i)}\operatorname{Im}\lim_{z\to\gamma_i-\mathrm{i}0^+}\left\{\frac{1}{\zeta_3^*}\left[\mathcal{G}_{\rho_X}\left(\sqrt{\frac{z-\zeta_1^*}{\kappa\zeta_3^*}}\right)+\mathcal{G}_{\rho_X}\left(-\sqrt{\frac{z-\zeta_1^*}{\kappa\zeta_3^*}}\right)\right]\right\}
\end{aligned}
\tag{49}
$$

### C.2.1 Estimating $\boldsymbol{X}^2$

The resolvent relation we have found in (42) is in terms of $\boldsymbol{G}_{X^2}$. Therefore, like other RIEs in other problems [14, 19], we can express the estimator for $\boldsymbol{X}^2$ without any knowledge about $\rho_X$ or $\rho_{X^2}$. One can see that, the optimal RIE for $\boldsymbol{X}^2$ is constructed in the same way as for $\boldsymbol{X}$ with eigenvalues denoted by $\widehat{\xi^*_{x^2\,i}}$. To compute the optimal eigenvalues, we absorb $\sqrt{\kappa}$ into $\boldsymbol{X}$ and we use the exact expression in (48). In the end, we only need to divide by $\kappa$ to find an estimator for the true $\boldsymbol{X}^2$.

$$
\begin{aligned}
\widehat{\xi^*_{x^2\,i}} &\approx \frac{1}{N}\sum_{j=1}^{N}\lambda_j^2 O_X(\gamma_i, \lambda_j) \\[2mm]
&\approx \frac{1}{\pi\bar{\mu}_S(\gamma_i)}\operatorname{Im}\lim_{z\to\gamma_i-\mathrm{i}0^+}\frac{1}{N}\sum_{j=1}^{N}\frac{\lambda_j^2}{z-\zeta_1^*-\zeta_3^*\lambda_j^2} \\[2mm]
&= \frac{1}{\pi\bar{\mu}_S(\gamma_i)}\operatorname{Im}\lim_{z\to\gamma_i-\mathrm{i}0^+}\frac{1}{\zeta_3^*}\frac{1}{N}\sum_{j=1}^{N}\frac{\lambda_j^2}{\frac{z-\zeta_1^*}{\zeta_3^*}-\lambda_j^2} \\[2mm]
&= \frac{1}{\pi\bar{\mu}_S(\gamma_i)}\operatorname{Im}\lim_{z\to\gamma_i-\mathrm{i}0^+}-\frac{1}{\zeta_3^*}\frac{1}{N}\sum_{j=1}^{N}\frac{\frac{z-\zeta_1^*}{\zeta_3^*}-\lambda_j^2-\frac{z-\zeta_1^*}{\zeta_3^*}}{\frac{z-\zeta_1^*}{\zeta_3^*}-\lambda_j^2} \\[2mm]
&= \frac{1}{\pi\bar{\mu}_S(\gamma_i)}\operatorname{Im}\lim_{z\to\gamma_i-\mathrm{i}0^+}-\frac{1}{\zeta_3^*}\frac{1}{N}\sum_{j=1}^{N}\left[1-\frac{z-\zeta_1^*}{\zeta_3^*}\frac{1}{\frac{z-\zeta_1^*}{\zeta_3^*}-\lambda_j^2}\right] \\[2mm]
&\approx \frac{1}{\pi\bar{\mu}_S(\gamma_i)}\operatorname{Im}\lim_{z\to\gamma_i-\mathrm{i}0^+}-\frac{1}{\zeta_3^*}+\frac{z-\zeta_1^*}{\zeta_3^{*2}}\mathcal{G}_{\rho_{X^2}}\left(\frac{z-\zeta_1^*}{\zeta_3^*}\right) \\[2mm]
&\overset{(a)}{=} \frac{1}{\pi\bar{\mu}_S(\gamma_i)}\operatorname{Im}\lim_{z\to\gamma_i-\mathrm{i}0^+}p_3^* \\[2mm]
&\overset{(b)}{=} \frac{1}{\pi\bar{\mu}_S(\gamma_i)}\operatorname{Im}\lim_{z\to\gamma_i-\mathrm{i}0^+}\frac{z-\zeta_1^*}{\zeta_3^*}\mathcal{G}_{\bar{\mu}_S}(z)-\frac{1}{\zeta_3^*}
\end{aligned}
\tag{50}
$$

where in (a) we used (41), and for (b) we used (45). Thus, the optimal eigenvalues for $\boldsymbol{X}^2$ read:

$$
\widehat{\xi^*_{x^2\,i}} = \frac{1}{\kappa}\frac{1}{\pi\bar{\mu}_S(\gamma_i)}\operatorname{Im}\lim_{z\to\gamma_i-\mathrm{i}0^+}\frac{z-\zeta_1^*}{\zeta_3^*}\mathcal{G}_{\bar{\mu}_S}(z)-\frac{1}{\zeta_3^*}
\tag{51}
$$

Note that the parameters $\zeta_1^*, \zeta_3^*$ can be computed from (45), (46), without the knowledge of $\rho_X$ or $\rho_{X^2}$.

**Remark 3.** *The main barrier to find an estimator for $X$ is that the resolvent relation (42) is in terms of $\mathcal{G}_{\rho_{X^2}}$. Moreover, in the estimator for $X$, second equality in (49), we have the sum $\sum_{j=1}^{N} \frac{\lambda_j}{z - \zeta_1^* - \kappa \zeta_3^* \lambda_j^2}$ which cannot be written in terms of $\mathcal{G}_{\rho_{X^2}}$.*

**Remark 4.** *If we add the assumption that the matrix $X$ is positive semi-definite, without any further knowledge on the prior, we can use $\sqrt{\widehat{\xi_{x^2}^*}_i}$ for the eigenvalues of $\Xi_X(S)$. However, note that, this estimator is sub-optimal for $X$ as $\sqrt{\sum_{j=1}^{N} \lambda_j^2 \left( u_i^\mathsf{T} x_j \right)^2} \neq \sum_{j=1}^{N} \lambda_j \left( u_i^\mathsf{T} x_j \right)^2$.*

### C.3 Numerical Examples

In this section, we will illustrate the derived formulas (42), (48), and (49) with numerical experiments.

We consider matrices $Y, W \in \mathbb{R}^{N \times M}$ to have i.i.d. Gaussian entries, so $\mathcal{C}_{\mu_Y}^{(\alpha)}(z) = \mathcal{C}_{\mu_W}^{(\alpha)}(z) = \frac{1}{\alpha} z$ which leads to a simplification of saddle point equations (45):

$$\begin{cases} \zeta_1^* = \frac{1}{\alpha} p_2^*, \quad \zeta_2^* = \alpha z \frac{z \mathcal{G}_{\bar{\mu}_S}(z) - 1}{\alpha z \mathcal{G}_{\bar{\mu}_S}(z) + 1 - \alpha}, \quad \zeta_3^* = \frac{1}{\alpha} p_2^* \\ p_1^* = \mathcal{G}_{\bar{\mu}_S}(z), \quad p_2^* = \alpha \mathcal{G}_{\bar{\mu}_S}(z) + (1 - \alpha) \frac{1}{z}, \quad p_3^* = \frac{z - \zeta_1^*}{\zeta_3^*} \mathcal{G}_{\bar{\mu}_S}(z) - \frac{1}{\zeta_3^*} \end{cases} \tag{52}$$

#### C.3.1 Resolvent relation

We take $\kappa = 1$. In model (29), without loss of generality we can consider $X$ to be diagonal. In figures 5 and 6 respectively, we consider the $X$ to be a diagonal matrix obtained by taking the eigenvalues of a Wigner matrix and a Wishart matrix respectively.

Note that $\mu_S$ and $\mathcal{G}_{\bar{\mu}_S}(z)$ can be computed analytically using tools from random matrix theory, but the computation is highly involved. In our experiments, we use instead a numerical estimation of $\mathcal{G}_{\bar{\mu}_S}(z)$ obtained from the observation matrix with the help of a Cauchy kernel to compute the parameters $\zeta_1^*, \zeta_3^*$ (see section G, and [58] for details on the Cauchy kernel method).

Unlike the simpler models [15] for which the fluctuations are derived to be of the order $1/\sqrt{N}$, based on our derivation we cannot assess the order of fluctuations. However, from our numerics we observe that the fluctuations are of the order $o(N)$. Moreover, fluctuations near the edge points of density are larger (in particular for the last row in both figures 5, 6), which is due to the fact that the limiting measures have higher fluctuations on their edge-points.

Another observation, from comparison of figures 5, 6, is that the fluctuations for the first example are relatively larger than the second one. One possible guess could be that this is due to the symmetry of $\rho_X$ in the first example. However based on more extensive numerical observations (which we omit here) we speculate that this issue is in fact related to the existence of small eigenvalues of $X$. In other words, if $X$ has eigenvalue 0 or small eigenvalues, we have higher fluctuations in the relation (47).

#### C.3.2 Overlaps

To illustrate the formula for the overlap (48), we fix the matrix $X$ and run experiments over various realization of the model (29). For each experiment, we record the overlap of $k$-th left singular vector of $S$ and the eigenvectors of $X$. To compute the theoretical prediction, we find $\zeta_1^* = \zeta_3^*$ for $z = \bar{\gamma}_k - i0^+$ where $\bar{\gamma}_k$ is the average of $k$-th singular value of $S$ in the experiments.

To find $\zeta_1^* = \zeta_3^*$, we use the set of equations (41) which for $Y, W$ Gaussian can be written as:

$$\begin{cases} \zeta_1^* = \frac{1}{\alpha} p_2^*, \quad \zeta_2^* = p_1^* + p_3^*, \quad \zeta_3^* = \frac{1}{\alpha} p_2^* \\ p_1^* = \frac{1}{\zeta_1^*} \mathcal{G}_{\rho_{X^2}} \left( \frac{z}{\zeta_1^*} - 1 \right), \quad p_2^* = \frac{1}{z - \zeta_2^*}, \quad p_3^* = \frac{z - \zeta_1^*}{\zeta_1^{*2}} \mathcal{G}_{\rho_{X^2}} \left( \frac{z}{\zeta_1^*} - 1 \right) - \frac{1}{\zeta_1^*} \end{cases} \tag{53}$$

Now we proceed to simplify the solution above:

$$\zeta_2^* = p_1^* + p_3^* = \frac{z}{\zeta_1^{*2}} \mathcal{G}_{\rho_{X^2}} \left( \frac{z}{\zeta_1^*} - 1 \right) - \frac{1}{\zeta_1^*}$$

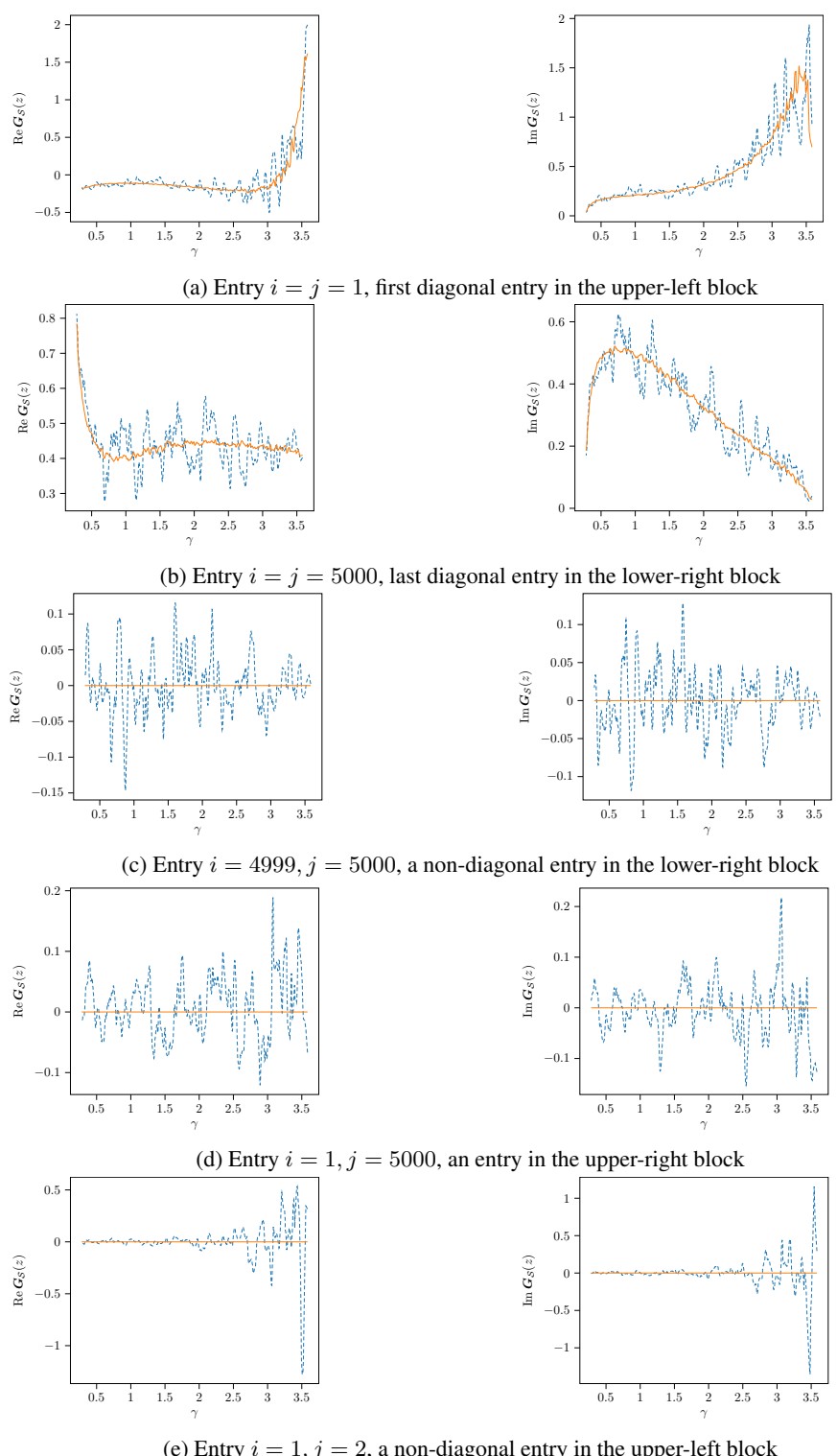

(a) Entry $i = j = 1$, first diagonal entry in the upper-left block

(b) Entry $i = j = 5000$, last diagonal entry in the lower-right block

(c) Entry $i = 4999, j = 5000$, a non-diagonal entry in the lower-right block

(d) Entry $i = 1, j = 5000$, an entry in the upper-right block

(e) Entry $i = 1, j = 2$, a non-diagonal entry in the upper-left block

Figure 5: Illustration of (42). $\boldsymbol{X}$ is diagonal matrix from the eigenvalues of a Wigner matrix and $\boldsymbol{Y}, \boldsymbol{Z}$ are Gaussian matrices with $N = 2000, M = 3000$. The empirical estimate of $\boldsymbol{G}_\mathcal{S}(z)$ (dashed blue line) is computed for $z = \gamma_i - \mathrm{i}\sqrt{\frac{1}{2N}}$ for $1 \le i \le N$. Theoretical estimate (solid orange line) computed from the rhs of (42) with parameters obtained from the generated matrix. Note that, the theoretical estimate has also fluctuations because the parameters $\zeta_1^*, \zeta_3^*$ are given by the numerical estimate of $\mathcal{G}_{\bar{\mu}_\mathcal{S}}(z)$.

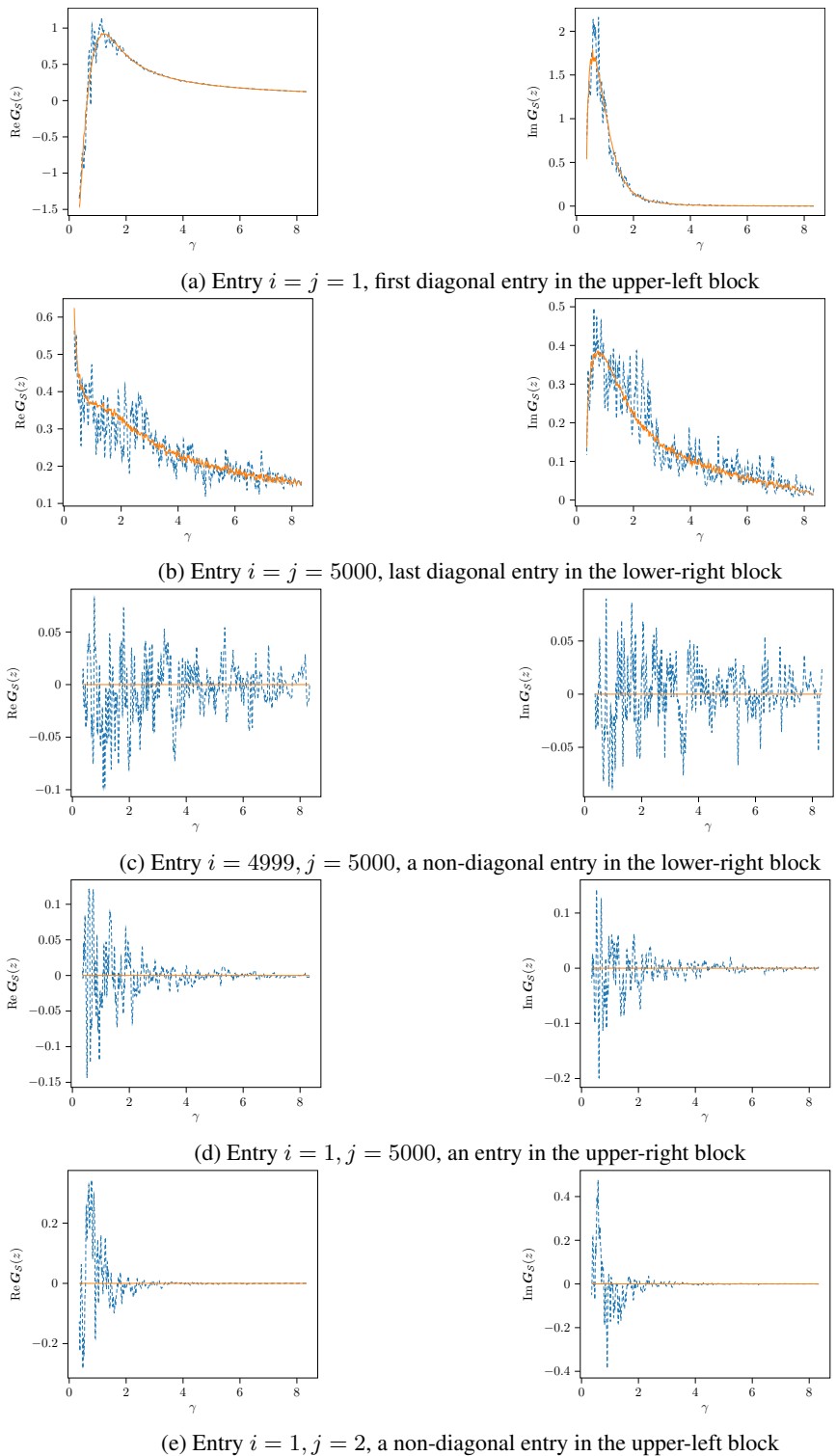

(a) Entry $i = j = 1$, first diagonal entry in the upper-left block

(b) Entry $i = j = 5000$, last diagonal entry in the lower-right block

(c) Entry $i = 4999, j = 5000$, a non-diagonal entry in the lower-right block

(d) Entry $i = 1, j = 5000$, an entry in the upper-right block

(e) Entry $i = 1, j = 2$, a non-diagonal entry in the upper-left block

Figure 6: Illustration of (42). $\boldsymbol{X}$ is diagonal matrix from the eigenvalues of a Wishart matrix with aspect ratio $^1\!/_2$ and $\boldsymbol{Y}, \boldsymbol{Z}$ are Gaussian matrices with $N = 2000, M = 3000$. The empirical estimate of $\boldsymbol{G}_{\mathcal{S}}(z)$ (dashed blue line) is computed for $z = \gamma_i - \mathsf{i}\sqrt{\frac{1}{2N}}$ for $1 \leq i \leq N$. The Theoretical estimate (solid orange line) is computed from the rhs of (42) with parameters obtained from the generated matrix. Note that, the theoretical estimate has also fluctuations because the parameters $\zeta_1^*, \zeta_3^*$ are given by the numerical estimate of $\mathcal{G}_{\bar{\mu}_S}(z)$.

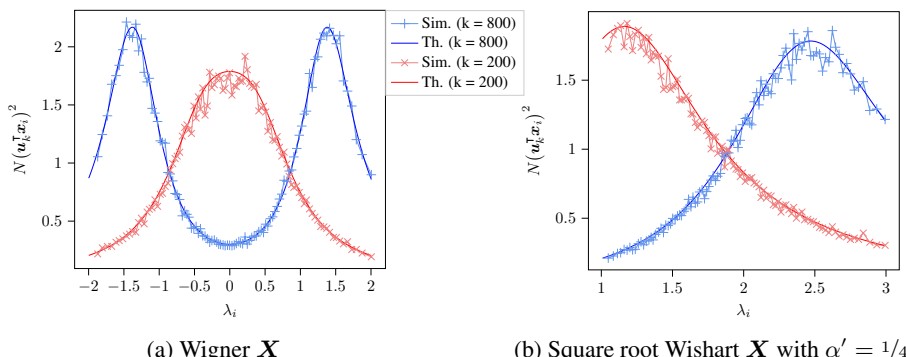

(a) Wigner $\boldsymbol{X}$          (b) Square root Wishart $\boldsymbol{X}$ with $\alpha' = 1/4$

Figure 7: Computation of the rescaled overlap. Both $\boldsymbol{Y}$ and $\boldsymbol{W}$ are $N \times M$ matrices with i.i.d. Gaussian entries of variance $1/N$, and aspect ratio $N/M = 1/2$. The simulation results are averaged over 1000 experiments with fixed $\boldsymbol{X}$, and $N = 1000, M = 2000$. Some of the simulation points are dropped for clarity.

$$p_2^* = \frac{1}{z - \zeta_2^*} = \frac{\zeta_1^*}{\zeta_1^* z - \frac{z}{\zeta_1^*} \mathcal{G}_{\rho_{X^2}}\left(\frac{z}{\zeta_1^*} - 1\right) + 1}$$

$$
\begin{aligned}
\zeta_1^* = \frac{1}{\alpha} p_2^* \Longrightarrow \zeta_1^* z - \frac{z}{\zeta_1^*} \mathcal{G}_{\rho_{X^2}}\left(\frac{z}{\zeta_1^*} - 1\right) + 1 &= \frac{1}{\alpha} \\
\Rightarrow \mathcal{G}_{\rho_{X^2}}\left(\frac{z}{\zeta_1^*} - 1\right) &= \zeta_1^{*2} + \left(1 - \frac{1}{\alpha}\right) \frac{\zeta_1^*}{z} \\
\Rightarrow \frac{z}{\zeta_1^*} - 1 &= \mathcal{G}_{\rho_{X^2}}^{-1}\left(\zeta_1^{*2} + \left(1 - \frac{1}{\alpha}\right) \frac{\zeta_1^*}{z}\right) \\
\Rightarrow \frac{z}{\zeta_1^*} - 1 - \frac{1}{\zeta_1^{*2} + \left(1 - \frac{1}{\alpha}\right) \frac{\zeta_1^*}{z}} &= \mathcal{R}_{\rho_{X^2}}\left(\zeta_1^{*2} + \left(1 - \frac{1}{\alpha}\right) \frac{\zeta_1^*}{z}\right)
\end{aligned}
\tag{54}
$$

Thus, $\zeta_1^*$ is the solution to (54). For each example, we solve this equation and compare the obtained theoretical overlap against the average over the experiments.

**Wigner $\boldsymbol{X}$.** Let $\boldsymbol{X} \in \mathbb{R}^{N \times N}$ be a Wigner matrix, then $\mathcal{R}_{\rho_{X^2}}(z) = \frac{1}{1-z}$. Solving (54), we can compute the overlap using (48). In Fig. 7a, we compare the theoretical computation with simulations for $N = 1000, M = 2000$. As in previous cases $\bar{\mu}_S(\gamma)$ is approximated using a Cauchy kernel [58].

**Square root Wishart $\boldsymbol{X}$.** Let $\boldsymbol{X} \in \mathbb{R}^{N \times N}$ be the square root of a Wishart matrix $\boldsymbol{X} = \sqrt{\frac{1}{N} \mathbf{H} \mathbf{H}^{\mathsf{T}}}$ with $\mathbf{H} \in \mathbb{R}^{N \times N'}$ having i.i.d. Gaussian entries. Then $\mathcal{R}_{\rho_{X^2}}(z) = \frac{1}{\alpha'} \frac{1}{1-z}, \alpha' = N/N'$. Solving (54), we can compute the overlap using (48). In Fig. 7b, we compare the theoretical computation with simulations for $N = 1000, N' = 4000, M = 2000$.

### C.3.3 RIE performance

In this section, we investigate the performance of our proposed estimators for $\boldsymbol{X}$. We compare performances of the optimal RIE (49) with the one of Oracle estimator (3). Moreover, we illustrate the performance of the estimator for $\boldsymbol{X}^2$ (50), and the sub-optimal estimator of $\boldsymbol{X}$ derived from it, see remark 4.

For $\boldsymbol{Y}, \boldsymbol{W}$ with Gaussian i.i.d. entries, (51) simplifies to:

$$
\begin{aligned}
\widehat{\xi^*_{x^2 i}} &= \frac{1}{\kappa}\frac{1}{\pi\bar{\mu}_S(\gamma_i)}\,\mathrm{Im}\,\lim_{z\to\gamma_i - i0^+} \frac{z-\zeta_1^*}{\zeta_3^*}\mathcal{G}_{\bar{\mu}_S}(z) - \frac{1}{\zeta_3^*}\\
&= \frac{1}{\kappa}\frac{1}{\pi\bar{\mu}_S(\gamma_i)}\,\mathrm{Im}\,\lim_{z\to\gamma_i - i0^+} \frac{z}{\zeta_1^*}\mathcal{G}_{\bar{\mu}_S}(z) - \mathcal{G}_{\bar{\mu}_S} - \frac{1}{\zeta_1^*}\\
&= \frac{1}{\kappa}\frac{1}{\pi\bar{\mu}_S(\gamma_i)}\,\mathrm{Im}\,\lim_{z\to\gamma_i - i0^+} \frac{z}{\mathcal{G}_{\bar{\mu}_S}(z) + \frac{1-\alpha}{\alpha}\frac{1}{z}}\mathcal{G}_{\bar{\mu}_S}(z) - \mathcal{G}_{\bar{\mu}_S}(z) - \frac{1}{\mathcal{G}_{\bar{\mu}_S}(z) + \frac{1-\alpha}{\alpha}\frac{1}{z}}\\
&= \frac{1}{\kappa}\frac{1}{\pi\bar{\mu}_S(\gamma_i)}\,\mathrm{Im}\,\Bigg\{ \frac{\gamma_i}{\pi\mathsf{H}[\bar{\mu}_S](\gamma_i) + \pi i\bar{\mu}_S(\gamma_i) + \frac{1-\alpha}{\alpha}\frac{1}{\gamma_i}}\big(\pi\mathsf{H}[\bar{\mu}_S](\gamma_i) + \pi i\bar{\mu}_S(\gamma_i)\big)\\
&\qquad\qquad - \big(\pi\mathsf{H}[\bar{\mu}_S](\gamma_i) + \pi i\bar{\mu}_S(\gamma_i)\big) - \frac{1}{\pi\mathsf{H}[\bar{\mu}_S](\gamma_i) + \pi i\bar{\mu}_S(\gamma_i) + \frac{1-\alpha}{\alpha}\frac{1}{\gamma_i}}\Bigg\}\\
&= \frac{1}{\kappa}\frac{1}{\pi\bar{\mu}_S(\gamma_i)}\,\pi\bar{\mu}_S(\gamma_i)\Bigg( -1 + \frac{1}{\alpha\Big(\pi^2\bar{\mu}_S(\gamma_i)^2 + \big(\pi\mathsf{H}[\bar{\mu}_S](\gamma_i) + \frac{-1+\frac{1}{\alpha}}{\gamma_i}\big)^2\Big)}\Bigg)\\
&= \frac{1}{\kappa}\Bigg[ -1 + \frac{1}{\alpha\Big(\pi^2\bar{\mu}_S(\gamma_i)^2 + \big(\pi\mathsf{H}[\bar{\mu}_S](\gamma_i) + \frac{-1+\frac{1}{\alpha}}{\gamma_i}\big)^2\Big)}\Bigg]
\end{aligned}
$$
(55)

For our first example, we consider two priors for $\boldsymbol{X}$:

**Shifted Wigner $\boldsymbol{X}$.** We consider $\boldsymbol{X} = \boldsymbol{F} + c\boldsymbol{I}$ where $\boldsymbol{F} = \boldsymbol{F}^\intercal \in \mathbb{R}^{N\times N}$ has i.i.d. entries with variance $1/N$, and $c \neq 0$ is a real number. Then, the spectrum of $\boldsymbol{X}$ is a shifted version of the Wigner law

$$
\rho_X(\lambda) = \frac{\sqrt{4-(\lambda-c)^2}}{2\pi}, \quad \text{for } c-2 < \lambda < c+2,
$$

and the Stieltjes transform reads:

$$
\mathcal{G}_{\rho_X}(z) = \frac{z - c - \sqrt{(z-2-c)(z+2-c)}}{2}
$$

**Wishart $\boldsymbol{X}$.** Take $\boldsymbol{X} = \frac{1}{N}\mathbf{H}\mathbf{H}^\intercal$ with $\mathbf{H} \in \mathbb{R}^{N\times N'}$ having i.i.d. Gaussian entries, with $N/N' = \alpha' \leq 1$. Then, the spectrum of $\boldsymbol{X}$ is the renowned *Marchenko-Pastur* distribution:

$$
\rho_X(\lambda) = \frac{\sqrt{\big[\lambda - \big(\frac{1}{\sqrt{\alpha'}}-1\big)^2\big]\big[\big(\frac{1}{\sqrt{\alpha'}}+1\big)^2 - \lambda\big]}}{2\pi\lambda}, \quad \text{for } \big(\frac{1}{\sqrt{\alpha'}}-1\big)^2 < \lambda < \big(\frac{1}{\sqrt{\alpha'}}+1\big)^2,
$$

and the Stieltjes transform reads:

$$
\mathcal{G}_{\rho_X}(z) = \frac{z - \big(\frac{1}{\alpha'}-1\big) - \sqrt{\big[z - \big(\frac{1}{\sqrt{\alpha'}}-1\big)^2\big]\big[z - \big(\frac{1}{\sqrt{\alpha'}}+1\big)^2\big]}}{2z}
$$

In Figure 8, the MSE of Oracle estimator, RIE (49), and $\sqrt{\boldsymbol{X}^2}$-RIE is illustrated for shifted Wigner $\boldsymbol{X}$ with $c=3$, and Wishart with aspect-ratio $\alpha' = 1/4$. We see that the performance of RIE is close to the one of Oracle estimator, which implies the optimality of the proposed estimator (49). Moreover, we observe the sub-optimality of estimating $\boldsymbol{X}$ using $\sqrt{\widehat{\boldsymbol{\Xi}^*_{X^2}}(\boldsymbol{S})}$. Note that, in the low-SNR regime, the estimated eigenvalues $\widehat{\xi^*_{x^2 i}}$ might be negative which makes the estimator $\sqrt{\widehat{\boldsymbol{\Xi}^*_{X^2}}(\boldsymbol{S})}$ undefined, so the MSE is not depicted in this case.

In Figure 9, the MSE of estimating $\boldsymbol{X}^2$ is shown. We see that in the high-SNR regimes the RIE (55) has the same performance as the Oracle estimator.

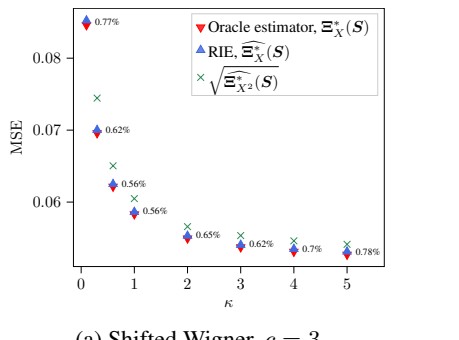
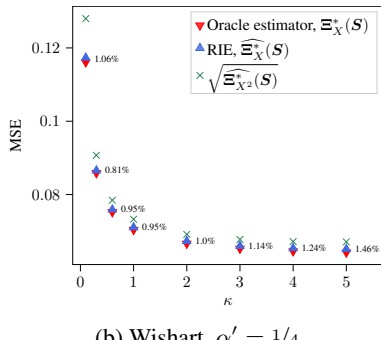

(a) Shifted Wigner, $c = 3$         (b) Wishart, $\alpha' = 1/4$

Figure 8: Estimating $\boldsymbol{X}$. The MSE is normalized by the norm of the signal, $\|\boldsymbol{X}\|_F^2$. Both $\boldsymbol{Y}$ and $\boldsymbol{W}$ are $N \times M$ matrices with i.i.d. Gaussian entries of variance $1/N$, and aspect ratio $N/M = 1/2$. The RIE is applied to $N = 2000$, $M = 4000$, and the results are averaged over 10 runs (error bars are invisible). Average relative error between RIE $\widehat{\boldsymbol{\Xi}}_X^*(\boldsymbol{S})$ and Oracle estimator is also reported.

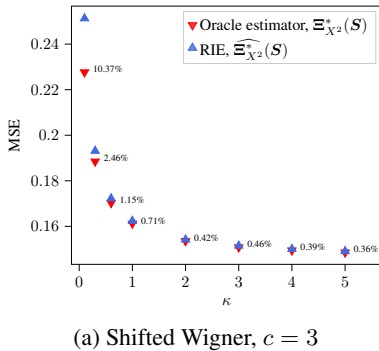
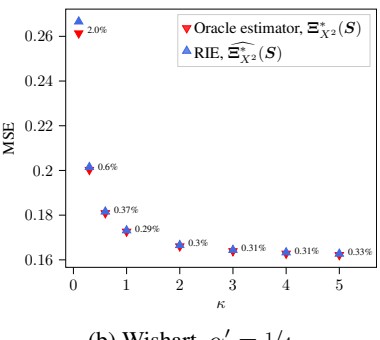

(a) Shifted Wigner, $c = 3$         (b) Wishart, $\alpha' = 1/4$

Figure 9: Estimating $\boldsymbol{X}^2$. The MSE is normalized by the norm of the signal, $\|\boldsymbol{X}^2\|_F^2$. Both $\boldsymbol{Y}$ and $\boldsymbol{W}$ are $N \times M$ matrices with i.i.d. Gaussian entries of variance $1/N$, and aspect ratio $N/M = 1/2$. The RIE is applied to $N = 2000$, $M = 4000$, and the results are averaged over 10 runs (error bars are invisible). Average relative error between RIE $\widehat{\boldsymbol{\Xi}}_X^*(\boldsymbol{S})$ and Oracle estimator is also reported.

**Bernoulli spectral distribution.** In this case, the matrix $\boldsymbol{X}$ is constructed as $\boldsymbol{X} = \boldsymbol{U}_X \boldsymbol{\Lambda} \boldsymbol{U}_X^\mathsf{T}$ with $\boldsymbol{U}_X$ a $N \times N$ orthogonal matrix distributed according to Haar measure on orthogonal matrices, and $\boldsymbol{\Lambda} = \operatorname{diag}(\boldsymbol{\lambda})$ where $\boldsymbol{\lambda}$ has i.i.d. Bernoulli elements. Thus, $\rho_X = p\delta_0 + (1-p)\delta_{+1}$ for $p \in (0,1)$, and the Stieltjes transform is:

$$\mathcal{G}_{\rho_X}(z) = p\frac{1}{z} + (1-p)\frac{1}{z-1}$$

For this prior, we have that $\boldsymbol{X} = \boldsymbol{X}^2$, so both estimators $\widehat{\boldsymbol{\Xi}}_X^*(\boldsymbol{S})$ and $\widehat{\boldsymbol{\Xi}}_{X^2}^*(\boldsymbol{S})$ should have the same performance. However, note that $\widehat{\boldsymbol{\Xi}}_{X^2}^*(\boldsymbol{S})$ does not use any knowledge of $\rho_X$. In Figure 10, the MSE is illustrated for these two estimators for two sparsity parameter, $p = 0.5$ and $0.9$. We observe that, except for for the low-SNR regimes, both estimators have the same MSE. The poor performance of $\widehat{\boldsymbol{\Xi}}_{X^2}^*(\boldsymbol{S})$ in the low-SNR regimes might be due to the fact that, some of the estimated eigenvalues $\widehat{\xi}_{x^2,i}^*$ are negative although the true eigenvalue is 0. This makes the estimation more difficult for the sparser prior, see Figure 10b. However, this problem is resolved in $\widehat{\boldsymbol{\Xi}}_X^*(\boldsymbol{S})$ by taking the knowledge of $\mathcal{G}_{\rho_X}(z)$ into account.

**Effect of aspect-ratio $\alpha$.** In Figure 11, we consider $\boldsymbol{X}$ to be shifted Wigner with $c = 3$, and the MSE is depicted for various values of the aspect-ratio $\alpha$. As expected, as $M$ increases ($\alpha$ decreases) and we have more observation or more data samples, the estimation error decreases.

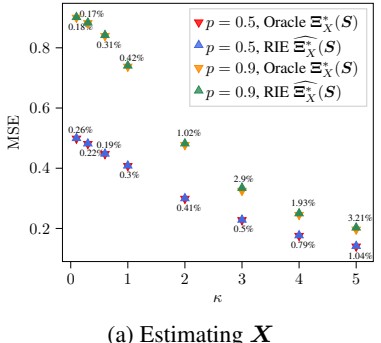
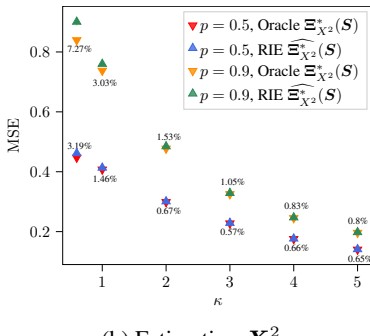

(a) Estimating $\boldsymbol{X}$         (b) Estimating $\boldsymbol{X}^2$

Figure 10: Estimating $\boldsymbol{X}$ and $\boldsymbol{X}^2$ with Bernoulli spectral prior distribution. The MSE is normalized by the norm of the signal, $\|\boldsymbol{X}\|_{\mathrm{F}}^2 = \|\boldsymbol{X}^2\|_{\mathrm{F}}^2$. Both $\boldsymbol{Y}$ and $\boldsymbol{W}$ are $N \times M$ matrices with i.i.d. Gaussian entries of variance $1/N$, and aspect ratio $N/M = 1/2$. The RIE is applied to $N = 2000, M = 4000$, and the results are averaged over 10 runs (error bars are invisible). Average relative error between RIE $\widehat{\boldsymbol{\Xi}}_X^*(\boldsymbol{S})$ and Oracle estimator is also reported.

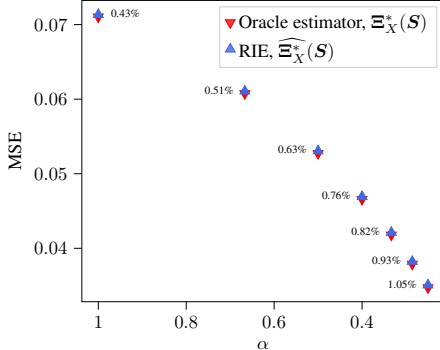

Figure 11: MSE of estimating $\boldsymbol{X}$ as a function of aspect-ratio $\alpha$, prior on $\boldsymbol{X}$ is shifted Wigner with $c = 3$, and $\kappa = 5$. MSE is normalized by the norm of the signal, $\|\boldsymbol{X}\|_{\mathrm{F}}^2$. Both $\boldsymbol{Y}$ and $\boldsymbol{W}$ are $N \times M$ matrices with i.i.d. Gaussian entries of variance $1/N$. The RIE is applied to $N = 2000, M = 1/\alpha N$, and the results are averaged over 10 runs (error bars are invisible). Average relative error between RIE $\widehat{\boldsymbol{\Xi}}_X^*(\boldsymbol{S})$ and Oracle estimator is also reported.

# D    Estimating $\boldsymbol{Y}$

In this section, we present the derivation of the optimal RIE for $\boldsymbol{Y}$. For simplicity, the SNR parameter in (1) is absorbed into $\boldsymbol{Y}$, so the model is $\boldsymbol{S} = \boldsymbol{X}\boldsymbol{Y} + \boldsymbol{W}$. Therefore, the final estimator should be divided by $1/\sqrt{\kappa}$ to give an estimate of the original $\boldsymbol{Y}$.

The optimal singular values are constructed as $\xi_{y_i}^* = \sum_{j=1}^{N} \sigma_j \left( \boldsymbol{u}_i^\mathsf{T} \boldsymbol{y}_j^{(l)} \right) \left( \boldsymbol{v}_i^\mathsf{T} \boldsymbol{y}_j^{(r)} \right)$. We assume that, for large $N$, $\xi_{y_i}^*$ can be approximated by its expectation:

$$\widehat{\xi}_{y_i}^* \approx \sum_{j=1}^{N} \sigma_j \, \mathbb{E}\left[ \left( \boldsymbol{u}_i^\mathsf{T} \boldsymbol{y}_j^{(l)} \right) \left( \boldsymbol{v}_i^\mathsf{T} \boldsymbol{y}_j^{(r)} \right) \right]$$

where the expectation is over the singular vectors of the observation matrix $\boldsymbol{S}$. Therefore, to compute the optimal singular values, we need to find the mean overlap $\mathbb{E}\left[ \left( \boldsymbol{u}_i^\mathsf{T} \boldsymbol{y}_j^{(l)} \right) \left( \boldsymbol{v}_i^\mathsf{T} \boldsymbol{y}_j^{(r)} \right) \right]$ between singular vectors of $\boldsymbol{Y}$ and singular vectors of $\boldsymbol{S}$. In the following we will see that (a rescaling of) this quantity can be expressed in terms of $i$-th singular value of $\boldsymbol{S}$ and $j$-th singular value of $\boldsymbol{Y}$ (and the limiting measures, indeed). Thus, we will use the notation $O_Y(\gamma_i, \sigma_j) := N\mathbb{E}\left[ \left( \boldsymbol{u}_i^\mathsf{T} \boldsymbol{y}_j^{(l)} \right) \left( \boldsymbol{v}_i^\mathsf{T} \boldsymbol{y}_j^{(r)} \right) \right]$ in what follows. In the nest section, we discuss how the overlap can be computed from the resolvent of the Hermitized matrix of $\boldsymbol{S}$.

## D.1 Relation between overlap and the resolvent

Construct the matrix $\boldsymbol{\mathcal{S}} \in \mathbb{R}^{(N+M)\times(N+M)}$ from the observation matrix:

$$\boldsymbol{\mathcal{S}} = \left[ \begin{array}{cc} \mathbf{0}_{N\times N} & \boldsymbol{S} \\ \boldsymbol{S}^\intercal & \mathbf{0}_{M\times M} \end{array} \right]$$

By Theorem 7.3.3 in [59], $\boldsymbol{\mathcal{S}}$ has the following eigen-decomposition:

$$\boldsymbol{\mathcal{S}} = \left[ \begin{array}{ccc} \hat{\boldsymbol{U}}_S & \hat{\boldsymbol{U}}_S & \mathbf{0} \\ \hat{\boldsymbol{V}}_S^{(1)} & -\hat{\boldsymbol{V}}_S^{(1)} & \boldsymbol{V}_S^{(2)} \end{array} \right] \left[ \begin{array}{ccc} \boldsymbol{\Gamma}_N & \mathbf{0} & \mathbf{0} \\ \mathbf{0} & -\boldsymbol{\Gamma}_N & \mathbf{0} \\ \mathbf{0} & \mathbf{0} & \mathbf{0} \end{array} \right] \left[ \begin{array}{ccc} \hat{\boldsymbol{U}}_S & \hat{\boldsymbol{U}}_S & \mathbf{0} \\ \hat{\boldsymbol{V}}_S^{(1)} & -\hat{\boldsymbol{V}}_S^{(1)} & \boldsymbol{V}_S^{(2)} \end{array} \right]^\intercal \tag{56}$$

with $\boldsymbol{V}_S = \left[ \begin{array}{cc} \boldsymbol{V}_S^{(1)} & \boldsymbol{V}_S^{(2)} \end{array} \right]$ in which $\boldsymbol{V}_S^{(1)} \in \mathbb{R}^{M\times N}$. And, $\hat{\boldsymbol{V}}_S^{(1)} = \frac{1}{\sqrt{2}}\boldsymbol{V}_S^{(1)}$, $\hat{\boldsymbol{U}}_S = \frac{1}{\sqrt{2}}\boldsymbol{U}_S$. Eigenvalues of $\boldsymbol{\mathcal{S}}$ are signed singular values of $\boldsymbol{S}$, therefore the limiting eigenvalue distribution of $\boldsymbol{\mathcal{S}}$ (ignoring zero eigenvalues) is the same as the limiting symmetrized singular value distribution of $\boldsymbol{S}$.

Define the resolvent of $\boldsymbol{\mathcal{S}}$

$$\boldsymbol{G}_\mathcal{S}(z) = \left( z\boldsymbol{I} - \boldsymbol{\mathcal{S}} \right)^{-1}$$

Denote the eigenvectors of $\boldsymbol{\mathcal{S}}$ by $\mathbf{s}_i \in \mathbb{R}^{M+N}$, $i = 1, \ldots, M+N$. For $z = x - i\epsilon$ with $x \in \mathbb{R}$ and $\epsilon \gg \frac{1}{N}$, we have:

$$\boldsymbol{G}_\mathcal{S}(x - i\epsilon) = \sum_{k=1}^{2N} \frac{x + i\epsilon}{(x - \tilde{\gamma}_k)^2 + \epsilon^2}\mathbf{s}_k\mathbf{s}_k^\intercal + \frac{x + i\epsilon}{x^2 + \epsilon^2} \sum_{k=2N+1}^{N+M} \mathbf{s}_k\mathbf{s}_k^\intercal$$

where $\tilde{\gamma}_k$ are the eigenvalues of $\boldsymbol{\mathcal{S}}$, which are in fact the (signed) singular values of $\boldsymbol{S}$, $\tilde{\gamma}_1 = \gamma_1, \ldots, \tilde{\gamma}_N = \gamma_N, \tilde{\gamma}_{N+1} = -\gamma_1, \ldots, \tilde{\gamma}_{2N} = -\gamma_N$.

Define the vectors $\boldsymbol{r}_i = \left[ \begin{array}{c} \mathbf{0}_N \\ \boldsymbol{y}_i^{(r)} \end{array} \right], \boldsymbol{l}_i = \left[ \begin{array}{c} \boldsymbol{y}_i^{(l)} \\ \mathbf{0}_M \end{array} \right]$ for $\boldsymbol{y}_i^{(r)}, \boldsymbol{y}_i^{(l)}$ right/ left singular vectors of $\boldsymbol{Y}$, we have

$$\boldsymbol{r}_i^\intercal\left( \mathrm{Im}\,\boldsymbol{G}_\mathcal{S}(x - i\epsilon) \right)\boldsymbol{l}_i = \sum_{k=1}^{2N} \frac{\epsilon}{(x - \tilde{\gamma}_k)^2 + \epsilon^2}\left( \boldsymbol{r}_i^\intercal\mathbf{s}_k \right)\left( \boldsymbol{l}_i^\intercal\mathbf{s}_k \right) + \frac{x + i\epsilon}{x^2 + \epsilon^2} \sum_{k=2N+1}^{N+M} \left( \boldsymbol{r}_i^\intercal\mathbf{s}_k \right)\left( \boldsymbol{l}_i^\intercal\mathbf{s}_k \right) \tag{57}$$

Given the structure of $\mathbf{s}_k$'s in (56), we have:

$$\left( \boldsymbol{r}_i^\intercal\mathbf{s}_k \right)\left( \boldsymbol{l}_i^\intercal\mathbf{s}_k \right) = \begin{cases} \frac{1}{2}\left( \boldsymbol{u}_k^\intercal\boldsymbol{y}_i^{(l)} \right)\left( \boldsymbol{v}_k^\intercal\boldsymbol{y}_i^{(r)} \right) & \text{for } 1 \leq k \leq N \\ -\frac{1}{2}\left( \boldsymbol{u}_{k-N}^\intercal\boldsymbol{y}_i^{(l)} \right)\left( \boldsymbol{v}_{k-N}^\intercal\boldsymbol{y}_i^{(r)} \right) & \text{for } N+1 \leq k \leq 2N \\ 0 & \text{for } 2N+1 \leq k \leq N+M \end{cases}$$

In the limit of large N, the latter quantity is also self-averaging, due to the fact that as $N \to \infty$, these overlaps exhibit asymptotic independence, enabling the law of large numbers to be applied here. We can thus state that:

$$\boldsymbol{r}_i^\intercal\left( \mathrm{Im}\,\boldsymbol{G}_\mathcal{S}(x - i\epsilon) \right)\boldsymbol{l}_i \xrightarrow{N\to\infty} \int_\mathbb{R} \frac{\epsilon}{(x - t)^2 + \epsilon^2}O_Y(t, \sigma_i)\bar{\mu}_S(t)\, dt \tag{58}$$

where the overlap function $O_Y(t, \lambda_i)$ is extended (continuously) to arbitrary values within the support of $\bar{\mu}_S$ with the property that $O_Y(-t, \lambda_i) = -O_Y(t, \lambda_i)$ for $t \in \mathrm{supp}(\mu_S)$. Sending $\epsilon \to 0$, we find

$$\boldsymbol{r}_i^\intercal\left( \mathrm{Im}\,\boldsymbol{G}_\mathcal{S}(x - i\epsilon) \right)\boldsymbol{l}_i \approx \pi\bar{\mu}_S(x)O_Y(x, \sigma_i) \tag{59}$$

In the next section, we establish a connection between the resolvent $\boldsymbol{G}_\mathcal{S}(z)$ and the signal $\boldsymbol{Y}$, which enables us to determine the overlap and consequently the optimal singular values values $\widehat{\xi}_{y_i}^*$ in terms of the singular values of the observation matrix $\boldsymbol{S}$.

## D.2 Resolvent relation for $Y$

In this section, we consider estimating $Y$, and treat both $X$ and $W$ as noise. We consider the model to be:
$$S = OXO^{\mathsf{T}}Y + UWV^{\mathsf{T}} \tag{60}$$
where $X = X^{\mathsf{T}} \in \mathbb{R}^{N \times N}, W \in \mathbb{R}^{N \times M}$ are fixed matrices with limiting eigenvalue/singular value distribution $\rho_X, \mu_W$, and $O, U \in \mathbb{R}^{N \times N}, V \in \mathbb{R}^{M \times M}$ are independent random Haar matrices. For simplicity of notation, we use $T \equiv OXO^{\mathsf{T}}Y$, and $\mathcal{T} \in \mathbb{R}^{(N+M) \times (N+M)}$ the hermitization of $T$. And $\widetilde{\mathcal{W}}$ denotes the hermitization of the matrix $UWV^{\mathsf{T}}$.

As in the case for $X$, we express the entries of $G(z) \equiv G_S(z)$ using Gaussian integral representation, and after applying the replica trick (28), we find:

$$\langle G_{ij}(z) \rangle = \lim_{n \to \infty} \int \Big( \prod_{k=1}^{N+M} \prod_{\tau=1}^{n} d\eta_k^{(\tau)} \Big) \eta_i^{(1)} \eta_j^{(1)} \Big\langle \exp\Big\{ -\frac{1}{2} \sum_{\tau=1}^{n} \eta^{(\tau)\mathsf{T}} (zI - S) \eta^{(\tau)} \Big\} \Big\rangle_{O,U,V}$$

$$= \lim_{n \to \infty} \int \Big( \prod_{k=1}^{N+M} \prod_{\tau=1}^{n} d\eta_k^{(\tau)} \Big) \eta_i^{(1)} \eta_j^{(1)} \exp\Big\{ -\frac{z}{2} \sum_{\tau=1}^{n} \eta^{(\tau)\mathsf{T}} \eta^{(\tau)} \Big\}$$

$$\times \Big\langle \exp\Big\{ \frac{1}{2} \sum_{\tau=1}^{n} \eta^{(\tau)\mathsf{T}} \mathcal{T} \eta^{(\tau)} \Big\} \Big\rangle_O \Big\langle \exp\Big\{ \frac{1}{2} \sum_{\tau=1}^{n} \eta^{(\tau)\mathsf{T}} \widetilde{\mathcal{W}} \eta^{(\tau)} \Big\} \Big\rangle_{U,V} \tag{61}$$

Split each replica $\eta^{(\tau)}$ into two vectors $a^{(\tau)} \in \mathbb{R}^N, b^{(\tau)} \in \mathbb{R}^M, \eta^{(\tau)} = \begin{bmatrix} a^{(\tau)} \\ b^{(\tau)} \end{bmatrix}$. The exponent in the first bracket in (61) can be written as :

$$\eta^{(\tau)\mathsf{T}} \mathcal{T} \eta^{(\tau)} = a^{(\tau)\mathsf{T}} OXO^{\mathsf{T}} Y b^{(\tau)} + b^{(\tau)\mathsf{T}} Y^{\mathsf{T}} OXO^{\mathsf{T}} a^{(\tau)}$$

$$= \operatorname{Tr} OXO^{\mathsf{T}} (\underbrace{Y b^{(\tau)} a^{(\tau)\mathsf{T}} + a^{(\tau)} b^{(\tau)\mathsf{T}} Y^{\mathsf{T}}}_{\tilde{Y}^{(\tau)}}) \tag{62}$$

where $\tilde{Y}^{(\tau)}$ is a symmetric $N \times N$ matrix with two non-zero eigenvalues $a^{(\tau)\mathsf{T}} Y b^{(\tau)} \pm \|a^{(\tau)}\| \|Y b^{(\tau)}\|$ by lemma 3.

Using the formula for the spherical integral [21] (see Theorem 1 in H.1), we find:

$$\Big\langle \exp\Big\{ \frac{1}{2} \sum_{\tau=1}^{n} \operatorname{Tr} OXO^{\mathsf{T}} \tilde{Y}^{(\tau)} \Big\} \Big\rangle_O \approx \exp\Big\{ \frac{N}{2} \sum_{\tau=1}^{n} \mathcal{P}_{\rho_X} \Big( \frac{1}{N} (a^{(\tau)\mathsf{T}} Y b^{(\tau)} + \|a^{(\tau)}\| \|Y b^{(\tau)}\|) \Big)$$

$$+ \mathcal{P}_{\rho_X} \Big( \frac{1}{N} (a^{(\tau)\mathsf{T}} Y b^{(\tau)} - \|a^{(\tau)}\| \|Y b^{(\tau)}\|) \Big) \Big\} \tag{63}$$

By the same computation as previous section, for the second bracket we have:

$$\Big\langle \exp\Big\{ \sum_{\tau=1}^{n} \operatorname{Tr} b^{(\tau)} a^{(\tau)\mathsf{T}} UWV^{\mathsf{T}} \Big\} \Big\rangle_{U,V} \approx \exp\Big\{ \frac{N}{2} \sum_{\tau=1}^{n} \mathcal{Q}_{\mu_W}^{(\alpha)} \Big( \frac{1}{NM} \|a^{(\tau)}\|^2 \|b^{(\tau)}\|^2 \Big) \Big\} \tag{64}$$

From (61), (63), (64), we find:

$$\langle G_{ij}(z) \rangle = \lim_{n \to \infty} \int \Big( \prod_{k=1}^{N+M} \prod_{\tau=1}^{n} d\eta_k^{(\tau)} \Big) \eta_i^{(1)} \eta_j^{(1)}$$

$$\times \exp\Big\{ -\frac{1}{2} \sum_{\tau=1}^{n} \Big[ z\|\eta^{(\tau)}\|^2 - N\mathcal{Q}_{\mu_W}^{(\alpha)} \Big( \frac{1}{NM} \|a^{(\tau)}\|^2 \|b^{(\tau)}\|^2 \Big)$$

$$- N\mathcal{P}_{\rho_X} \Big( \frac{1}{N} (a^{(\tau)\mathsf{T}} Y b^{(\tau)} + \|a^{(\tau)}\| \|Y b^{(\tau)}\|) \Big)$$

$$- N\mathcal{P}_{\rho_X} \Big( \frac{1}{N} (a^{(\tau)\mathsf{T}} Y b^{(\tau)} - \|a^{(\tau)}\| \|Y b^{(\tau)}\|) \Big) \Big] \Big\} \tag{65}$$

Now, we introduce delta functions (for brevity we drop the limit term):

$$
\langle G_{ij}(z)\rangle = \int \Big(\prod_{k=1}^{N+M}\prod_{\tau=1}^{n} d\eta_k^{(\tau)}\Big)\Big(\prod_{\tau=1}^{n} dp_1^{(\tau)}\,dq_2^{(\tau)}\,dq_3^{(\tau)}\,dq_4^{(\tau)}\Big)\,\eta_i^{(1)}\eta_j^{(1)}
$$

$$
\times \prod_{\tau=1}^{n} \delta\big(q_1^{(\tau)} - \tfrac{1}{N}\|\boldsymbol{a}^{(\tau)}\|^2\big)\,\delta\big(q_2^{(\tau)} - \tfrac{1}{M}\|\boldsymbol{b}^{(\tau)}\|^2\big)
$$

$$
\times \delta\big(q_3^{(\tau)} - \tfrac{1}{N}\|\boldsymbol{Y}\boldsymbol{b}^{(\tau)}\|^2\big)\,\delta\big(q_4^{(\tau)} - \tfrac{1}{N}\boldsymbol{a}^{(\tau)\mathsf{T}}\boldsymbol{Y}\boldsymbol{b}^{(\tau)}\big) \tag{66}
$$

$$
\times \exp\Big\{ -\tfrac{1}{2}\sum_{\tau=1}^{n} z\|\boldsymbol{\eta}^{(\tau)}\|^2 - N\mathcal{Q}_{\mu W}^{(\alpha)}\big(q_1^{(\tau)}q_2^{(\tau)}\big)
$$

$$
- N\mathcal{P}_{\rho X}\big(q_4^{(\tau)} + \sqrt{q_1^{(\tau)}q_3^{(\tau)}}\big) - N\mathcal{P}_{\rho X}\big(q_4^{(\tau)} - \sqrt{q_1^{(\tau)}q_3^{(\tau)}}\big)\Big\}
$$

In the next step, we replace each delta with its Fourier transform. Note that for the parameters $q_1, q_2, q_3$ we use $\delta\big(q_1^{\tau} - \tfrac{1}{N}\|\boldsymbol{a}^{\tau}\|^2\big) \propto \int d\beta_1^{\tau}\exp\Big\{ -\tfrac{N}{2}\beta_1^{\tau}\big(q_1^{\tau} - \tfrac{1}{N}\|\boldsymbol{a}^{\tau}\|^2\big)\Big\}$, and for $q_4$ we use $\delta\big(q_4^{(\tau)} - \tfrac{1}{N}\boldsymbol{a}^{(\tau)\mathsf{T}}\boldsymbol{Y}\boldsymbol{b}^{(\tau)}\big) \propto \int d\beta_1^{\tau}\exp\Big\{ -N\beta_1^{\tau}\big(q_4^{(\tau)} - \tfrac{1}{N}\boldsymbol{a}^{(\tau)\mathsf{T}}\boldsymbol{Y}\boldsymbol{b}^{(\tau)}\big)\Big\}$. After rearranging, we find:

$$
\langle G_{ij}(z)\rangle \propto \int \Big(\prod_{\tau=1}^{n} dq_1^{(\tau)}\,dq_2^{(\tau)}\,dq_3^{(\tau)}\,dq_4^{(\tau)}\,d\beta_1^{(\tau)}\,d\beta_2^{(\tau)}\,d\beta_3^{(\tau)}\,d\beta_4^{(\tau)}\Big)
$$

$$
\times \exp\Big\{ \tfrac{N}{2}\sum_{\tau=1}^{n}\mathcal{Q}_{\mu W}^{(\alpha)}\big(q_1^{(\tau)}q_2^{(\tau)}\big) + \mathcal{P}_{\rho X}\big(q_4^{(\tau)} + \sqrt{q_1^{(\tau)}q_3^{(\tau)}}\big) + \mathcal{P}_{\rho X}\big(q_4^{(\tau)} - \sqrt{q_1^{(\tau)}q_3^{(\tau)}}\big)
$$

$$
- \beta_1^{(\tau)}q_1^{(\tau)} - \tfrac{1}{\alpha}\beta_2^{(\tau)}q_2^{(\tau)} - \beta_3^{(\tau)}q_3^{(\tau)} - 2\beta_4^{(\tau)}q_4^{(\tau)}\Big\}
$$

$$
\times \int \Big(\prod_{k=1}^{N+M}\prod_{\tau=1}^{n} d\eta_k^{(\tau)}\Big)\,\eta_i^{(1)}\eta_j^{(1)}\exp\Big\{ -\tfrac{1}{2}\sum_{\tau=1}^{n} z\|\boldsymbol{\eta}^{(\tau)}\| - \beta_1^{(\tau)}\|\boldsymbol{a}^{(\tau)}\|^2 - \beta_2^{(\tau)}\|\boldsymbol{b}^{(\tau)}\|^2
$$

$$
- \beta_3^{(\tau)}\|\boldsymbol{Y}\boldsymbol{b}^{(\tau)}\|^2 - 2\beta_4^{(\tau)}\boldsymbol{a}^{(\tau)\mathsf{T}}\boldsymbol{Y}\boldsymbol{b}^{(\tau)}\Big\}
$$

$$
\tag{67}
$$

The inner integral is a Gaussian integral, and can be written as:

$$
\int \Big(\prod_{k=1}^{N+M}\prod_{\tau=1}^{n} d\eta_k^{(\tau)}\Big)\,\eta_i^{(1)}\eta_j^{(1)}
$$

$$
\times \exp\Big\{ \sum_{\tau=1}^{n} -\tfrac{1}{2}\boldsymbol{\eta}^{(\tau)\mathsf{T}}\begin{bmatrix} (z - \beta_1^{(\tau)})\boldsymbol{I}_N & -\beta_4^{(\tau)}\boldsymbol{Y} \\ -\beta_4^{(\tau)}\boldsymbol{Y}^{\mathsf{T}} & (z - \beta_2^{(\tau)})\boldsymbol{I}_M - \beta_3^{(\tau)}\boldsymbol{Y}^{\mathsf{T}}\boldsymbol{Y} \end{bmatrix}\boldsymbol{\eta}^{(\tau)}\Big\} \tag{68}
$$

Denote the matrix in the exponent by $\boldsymbol{C}_Y^{(\tau)}$. Using the formula for determinant of block matrices (see proposition 2.8.4 in [61]), we have::

$$
\det \boldsymbol{C}_Y^{(\tau)} = \det\Big[(z - \beta_1^{(\tau)})\boldsymbol{I}_N - \beta_4^{(\tau)2}\boldsymbol{Y}\big((z - \beta_2^{(\tau)})\boldsymbol{I}_M - \beta_3^{(\tau)}\boldsymbol{Y}^{\mathsf{T}}\boldsymbol{Y}\big)^{-1}\boldsymbol{Y}^{\mathsf{T}}\Big]
$$

$$
\times \det\Big[(z - \beta_2^{(\tau)})\boldsymbol{I}_M - \beta_3^{(\tau)}\boldsymbol{Y}^{\mathsf{T}}\boldsymbol{Y}\Big]
$$

$$
= \prod_{k=1}^{N}\Big[z - \beta_1^{(\tau)} - \beta_4^{(\tau)2}\frac{\sigma_k^2}{z - \beta_2^{(\tau)} - \beta_3^{(\tau)}\sigma_k^2}\Big]\prod_{k=1}^{N}\big(z - \beta_2^{(\tau)} - \beta_3^{(\tau)}\sigma_k^2\big)\big(z - \beta_2^{(\tau)}\big)^{M-N}
$$

$$
= \big(z - \beta_2^{(\tau)}\big)^{M-N}\prod_{k=1}^{N}\Big[(z - \beta_1^{(\tau)})(z - \beta_2^{(\tau)} - \beta_3^{(\tau)}\sigma_k^2) - \beta_4^{(\tau)2}\sigma_k^2\Big]
$$

$$
= \big(z - \beta_2^{(\tau)}\big)^{M-N}\prod_{k=1}^{N}\Big[(z - \beta_1^{(\tau)})(z - \beta_2^{(\tau)}) - \big(\beta_4^{(\tau)2} + \beta_3^{(\tau)}(z - \beta_1^{(\tau)})\big)\sigma_k^2\Big]
$$

where $\sigma_k$'s are the singular values of $\boldsymbol{Y}$. So computing the Gaussian integrals, (67) can be written as:

$$\langle G_{ij}(z) \rangle \propto \int \Big( \prod_{\tau=1}^{n} dq_1^{(\tau)} \, dq_2^{(\tau)} \, dq_3^{(\tau)} \, dq_4^{(\tau)} \, d\beta_1^{(\tau)} \, d\beta_2^{(\tau)} \, d\beta_3^{(\tau)} \, d\beta_4^{(\tau)} \Big) \big( \boldsymbol{C}_Y^{(1)^{-1}} \big)_{ij}$$

$$\times \exp \Big\{ - \frac{Nn}{2} F_0^Y (\boldsymbol{q}_1, \boldsymbol{q}_2, \boldsymbol{q}_3, \boldsymbol{q}_4, \boldsymbol{\beta}_1, \boldsymbol{\beta}_2, \boldsymbol{\beta}_3, \boldsymbol{\beta}_4) \Big\}$$

(69)

with

$$F_0^Y (\boldsymbol{q}_1, \boldsymbol{q}_2, \boldsymbol{q}_3, \boldsymbol{q}_4, \boldsymbol{\beta}_1, \boldsymbol{\beta}_2, \boldsymbol{\beta}_3, \boldsymbol{\beta}_4) = \frac{1}{n} \sum_{\tau=1}^{n} \Big[ \Big( \frac{1}{\alpha} - 1 \Big) \ln(z - \beta_2^{(\tau)})$$

$$+ \frac{1}{N} \sum_{k=1}^{N} \ln \Big( (z - \beta_1^{(\tau)})(z - \beta_2^{(\tau)}) - (\beta_4^{(\tau)^2} + \beta_3^{(\tau)}(z - \beta_1^{(\tau)})) \sigma_k^2 \Big)$$

$$- \mathcal{Q}_{\mu_W}^{(\alpha)}(q_1^{(\tau)} q_2^{(\tau)}) - \mathcal{P}_{\rho_X} \big( q_4^{(\tau)} + \sqrt{q_1^{(\tau)} q_3^{(\tau)}} \big) - \mathcal{P}_{\rho_X} \big( q_4^{(\tau)} - \sqrt{q_1^{(\tau)} q_3^{(\tau)}} \big)$$

$$+ \beta_1^{(\tau)} q_1^{(\tau)} + \frac{1}{\alpha} \beta_2^{(\tau)} q_2^{(\tau)} + \beta_3^{(\tau)} q_3^{(\tau)} + 2 \beta_4^{(\tau)} q_4^{(\tau)} \Big]$$

(70)

We will evaluate the integral (67) using saddle-points of the function $F_0^Y$. From the *replica symmetric ansatz* at the saddle-point we have:

$$\forall \tau \in \{1, \cdots, n\}: \quad \begin{cases} q_1^\tau = q_1, & q_2^\tau = q_2, & q_3^\tau = q_3, & q_4^\tau = q_4 \\ \beta_1^\tau = \beta_1, & \beta_2^\tau = \beta_2, & \beta_3^\tau = \beta_3, & \beta_4^\tau = \beta_4 \end{cases}$$

Finally, we find the solution to be:

$$\begin{cases} \beta_1^* = \frac{\mathcal{C}_{\mu_W}^{(\alpha)}(q_1^* q_2^*)}{q_1^*} + \frac{1}{2} \sqrt{\frac{q_3^*}{q_1^*}} \Big( \mathcal{R}_{\rho_X} \big( q_4^* + \sqrt{q_1^* q_3^*} \big) - \mathcal{R}_{\rho_X} \big( q_4^* - \sqrt{q_1^* q_3^*} \big) \Big) \\ \beta_2^* = \alpha \frac{\mathcal{C}_{\mu_W}^{(\alpha)}(q_1^* q_2^*)}{q_2^*} \\ \beta_3^* = \frac{1}{2} \sqrt{\frac{q_1^*}{q_3^*}} \Big( \mathcal{R}_{\rho_X} \big( q_4^* + \sqrt{q_1^* q_3^*} \big) - \mathcal{R}_{\rho_X} \big( q_4^* - \sqrt{q_1^* q_3^*} \big) \Big) \\ \beta_4^* = \frac{1}{2} \Big( \mathcal{R}_{\rho_X} \big( q_4^* + \sqrt{q_1^* q_3^*} \big) + \mathcal{R}_{\rho_X} \big( q_4^* - \sqrt{q_1^* q_3^*} \big) \Big) \\ q_1^* = \frac{(z - \beta_2^*) \beta_4^{*2}}{Z_2(z)^2} \mathcal{G}_{\rho_Y} \big( \frac{Z_1(z)}{Z_2(z)} \big) + \frac{\beta_3^*}{Z_2(z)} \\ q_2^* = \alpha \frac{z - \beta_1^*}{Z_2(z)} \mathcal{G}_{\rho_Y} \big( \frac{Z_1(z)}{Z_2(z)} \big) + \frac{1 - \alpha}{z - \beta_2^*} \\ q_3^* = \frac{(z - \beta_1^*) Z_1(z)}{Z_2(z)^2} \mathcal{G}_{\rho_Y} \big( \frac{Z_1(z)}{Z_2(z)} \big) - \frac{z - \beta_1^*}{Z_2(z)} \\ q_4^* = \frac{\beta_4^* Z_1(z)}{Z_2(z)^2} \mathcal{G}_{\rho_Y} \big( \frac{Z_1(z)}{Z_2(z)} \big) - \frac{\beta_4^*}{Z_2(z)} \end{cases} \quad \text{with} \quad \begin{cases} Z_1(z) = (z - \beta_1^*)(z - \beta_2^*) \\ Z_2(z) = \beta_4^{*2} + \beta_3^*(z - \beta_1^*) \end{cases}$$

(71)

where $\rho_Y$ is the limiting eigenvalue distribution of $\boldsymbol{Y}\boldsymbol{Y}^\intercal$.

The relation (69) and the solutions (71) hold for arbitrary indices $i, j$, so we can state the relation in the matrix form. Computing the inverse of $\boldsymbol{C}_Y^{*-1}$ (see section H.2), we have:

$$\langle \boldsymbol{G}_\mathcal{S}(z) \rangle_{\boldsymbol{O},\boldsymbol{U},\boldsymbol{V}} = \Bigg\langle \begin{bmatrix} \frac{1}{z} \boldsymbol{I}_N + \frac{1}{z} \boldsymbol{S} \boldsymbol{G}_{S^\intercal S}(z^2) \boldsymbol{S}^\intercal & \boldsymbol{S} \boldsymbol{G}_{S^\intercal S}(z^2) \\ \boldsymbol{G}_{S^\intercal S}(z^2) \boldsymbol{S}^\intercal & z \boldsymbol{G}_{S^\intercal S}(z^2) \end{bmatrix} \Bigg\rangle$$

$$= \begin{bmatrix} \frac{1}{z - \beta_1^*} \boldsymbol{I}_N + \frac{\beta_4^{*2}}{(z - \beta_1^*) Z_2(z)} \boldsymbol{Y} \boldsymbol{G}_{Y^\intercal Y} \big( \frac{Z_1(z)}{Z_2(z)} \big) \boldsymbol{Y}^\intercal & \frac{\beta_4^*}{Z_2(z)} \boldsymbol{Y} \boldsymbol{G}_{Y^\intercal Y} \big( \frac{Z_1(z)}{Z_2(z)} \big) \\ \frac{\beta_4^*}{Z_2(z)} \boldsymbol{G}_{Y^\intercal Y} \big( \frac{Z_1(z)}{Z_2(z)} \big) \boldsymbol{Y}^\intercal & \frac{z - \beta_1^*}{Z_2(z)} \boldsymbol{G}_{Y^\intercal Y} \big( \frac{Z_1(z)}{Z_2(z)} \big) \end{bmatrix}$$

(72)

With this relation, we can further simplify the solution (71).

We start with comparing the trace of upper-left block in (72). The normalized trace of the first block in $\langle \boldsymbol{G}_\mathcal{S}(z) \rangle_{\boldsymbol{O},\boldsymbol{U},\boldsymbol{V}}$ is computed in (43) to be $\mathcal{G}_{\bar{\mu}_\mathcal{S}}(z)$. The normalized trace of the upper-left block in

$C_Y^{*\,-1}$ is:

$$\frac{1}{N} \operatorname{Tr} \left[ (z - \beta_1^*)^{-1} \boldsymbol{I}_N + \frac{\beta_4^{*2}}{(z - \beta_1^*) Z_2(z)} \boldsymbol{Y} \boldsymbol{G}_{Y^\top Y}\left(\frac{Z_1(z)}{Z_2(z)}\right) \boldsymbol{Y}^\top \right]$$

$$= \frac{1}{N} \frac{1}{z - \beta_1^*} \sum_{k=1}^{N} \left[ 1 + \frac{\beta_4^{*2}}{Z_2(z)} \frac{\sigma_k^2}{\frac{Z_1(z)}{Z_2(z)} - \sigma_k^2} \right]$$

$$= \frac{1}{N} \frac{1}{z - \beta_1^*} \sum_{k=1}^{N} \left[ \frac{\beta_4^{*2} Z_1(z)}{Z_2^2(z)} \frac{1}{\frac{Z_1(z)}{Z_2(z)} - \sigma_k^2} + 1 - \frac{\beta_4^{*2}}{Z_2(z)} \right] \qquad (73)$$

$$= \frac{1}{N} \frac{1}{z - \beta_1^*} \frac{\beta_4^{*2} Z_1(z)}{Z_2^2(z)} \sum_{k=1}^{N} \frac{1}{\frac{Z_1(z)}{Z_2(z)} - \sigma_k^2} + \frac{1}{z - \beta_1^*} \frac{\beta_3^*(z - \beta_1^*)}{Z_2(z)}$$

$$= \frac{(z - \beta_2^*)\beta_4^{*2}}{Z_2(z)^2} \mathcal{G}_{\rho_Y}\left(\frac{Z_1(z)}{Z_2(z)}\right) + \frac{\beta_3^*}{Z_2(z)}$$

$$= q_1^*$$

Thus, $q_1^* = \mathcal{G}_{\bar{\mu}_S}(z)$.

The normalized trace of the lower-right block of $\langle \boldsymbol{G}_{\mathcal{S}}(z) \rangle_{\boldsymbol{O},\boldsymbol{U},\boldsymbol{V}}$ is $\alpha \mathcal{G}_{\bar{\mu}_S}(z) + (1 - \alpha)\frac{1}{z}$ (see (44)).
The normalized trace of the lower-right block in $C_Y^{*\,-1}$ is:

$$\frac{1}{M} \operatorname{Tr} \left[ \frac{z - \beta_1^*}{Z_2(z)} \boldsymbol{G}_{Y^\top Y}\left(\frac{Z_1(z)}{Z_2(z)}\right) \right] = \frac{1}{M} \frac{z - \beta_1^*}{Z_2(z)} \sum_{k=1}^{N} \frac{1}{\frac{Z_1(z)}{Z_2(z)} - \sigma_k^2} + \frac{M - N}{M} \frac{z - \beta_1^*}{Z_2(z)} \frac{Z_2(z)}{Z_1(z)}$$

$$= \frac{N}{M} \frac{1}{N} \frac{z - \beta_1^*}{Z_2(z)} \sum_{k=1}^{N} \frac{1}{\frac{Z_1(z)}{Z_2(z)} - \sigma_k^2} + \frac{M - N}{M} \frac{z - \beta_1^*}{Z_1(z)} \qquad (74)$$

$$= \alpha \frac{z - \beta_1^*}{Z_2(z)} \mathcal{G}_{\rho_Y}\left(\frac{Z_1(z)}{Z_2(z)}\right) + \frac{1 - \alpha}{z - \beta_2^*}$$

$$= q_2^*$$

So, $q_2^* = \alpha \mathcal{G}_{\bar{\mu}_S}(z) + (1 - \alpha)\frac{1}{z}$.

With a bit of algebra, we can express the parameters $q_3^*, q_4^*$ in terms of $q_1^*, \beta_1^*, \beta_4^*$:

$$q_3^* = \frac{(z - \beta_1^*)^2}{\beta_4^{*2}} q_1^* - \frac{z - \beta_1^*}{\beta_4^{*2}}, \qquad q_4^* = \frac{z - \beta_1^*}{\beta_4^*} q_1^* - \frac{1}{\beta_4^*} \qquad (75)$$

Therefore, the solution can be written without involving $\mathcal{G}_{\rho_Y}$, as:

$$\begin{cases} \beta_1^* = \frac{\mathcal{C}_{\mu_W}^{(\alpha)}(q_1^* q_2^*)}{q_1^*} + \frac{1}{2}\sqrt{\frac{q_3^*}{q_1^*}}\left(\mathcal{R}_{\rho_X}\left(q_4^* + \sqrt{q_1^* q_3^*}\right) - \mathcal{R}_{\rho_X}\left(q_4^* - \sqrt{q_1^* q_3^*}\right)\right) \\ \beta_2^* = \alpha \frac{\mathcal{C}_{\mu_W}^{(\alpha)}(q_1^* q_2^*)}{q_2^*} \\ \beta_3^* = \frac{1}{2}\sqrt{\frac{q_1^*}{q_3^*}}\left(\mathcal{R}_{\rho_X}\left(q_4^* + \sqrt{q_1^* q_3^*}\right) - \mathcal{R}_{\rho_X}\left(q_4^* - \sqrt{q_1^* q_3^*}\right)\right) \\ \beta_4^* = \frac{1}{2}\left(\mathcal{R}_{\rho_X}\left(q_4^* + \sqrt{q_1^* q_3^*}\right) + \mathcal{R}_{\rho_X}\left(q_4^* - \sqrt{q_1^* q_3^*}\right)\right) \\ q_1^* = \mathcal{G}_{\bar{\mu}_S}(z) \\ q_2^* = \alpha \mathcal{G}_{\bar{\mu}_S}(z) + (1 - \alpha)\frac{1}{z} \\ q_3^* = \frac{(z - \beta_1^*)^2}{\beta_4^{*2}} \mathcal{G}_{\bar{\mu}_S}(z) - \frac{z - \beta_1^*}{\beta_4^{*2}} \\ q_4^* = \frac{z - \beta_1^*}{\beta_4^*} \mathcal{G}_{\bar{\mu}_S}(z) - \frac{1}{\beta_4^*} \end{cases} \qquad (76)$$

**Remark 5.** *The simplifications in (75) are derived with the assumption that $\beta_4^* \neq 0$. However, in the initial set of equations (71), if $\rho_X$ is symmetric measure then $\beta_4^* = q_4^* = 0$ is a solution. If $\rho_X$ is symmetric, then $\mathcal{R}_{\rho_X}(-z) = -\mathcal{R}_{\rho_X}(z)$, and plugging $q_4^* = 0$ in the expression for $\beta_4^*$ in (71), we find that $\beta_4^* = 0$.*

## D.3 Overlaps and the optimal singular values

From (59), (72), we find:

$$
\begin{aligned}
O_Y(\gamma, \sigma_i) &\approx \frac{1}{\pi\bar\mu_S(\gamma)} \operatorname{Im} \lim_{z\to\gamma-\mathrm{i}0^+} \frac{\beta_4^*}{Z_2(z)} \boldsymbol{y}_i^{(r)\mathsf{T}} \boldsymbol{G}_{Y^\mathsf{T}Y}\Big(\frac{Z_1(z)}{Z_2(z)}\Big) \boldsymbol{Y}^\mathsf{T}\boldsymbol{y}_i^{(l)} \\
&= \frac{1}{\pi\bar\mu_S(\gamma)} \operatorname{Im} \lim_{z\to\gamma-\mathrm{i}0^+} \beta_4^* \frac{\sigma_i}{Z_1(z) - Z_2(z)\sigma_i^2}
\end{aligned}
\tag{77}
$$

From the overlap, we can compute the optimal singular values:

$$
\begin{aligned}
\widehat{\xi}^*_{y_i} &\approx \frac{1}{N}\sum_{j=1}^N \sigma_j O_Y(\gamma_i, \sigma_j) \\
&\approx \frac{1}{\pi\bar\mu_S(\gamma_i)} \operatorname{Im} \lim_{z\to\gamma_i-\mathrm{i}0^+} \frac{1}{N}\sum_{j=1}^N \beta_4^* \frac{\sigma_j^2}{Z_1(z) - Z_2(z)\sigma_j^2} \\
&= \frac{1}{\pi\bar\mu_S(\gamma_i)} \operatorname{Im} \lim_{z\to\gamma_i-\mathrm{i}0^+} \frac{1}{N}\frac{\beta_4^*}{Z_2(z)}\sum_{j=1}^N \frac{\sigma_j^2}{\frac{Z_1(z)}{Z_2(z)} - \sigma_j^2} \\
&= \frac{1}{\pi\bar\mu_S(\gamma_i)} \operatorname{Im} \lim_{z\to\gamma_i-\mathrm{i}0^+} \frac{1}{N}\frac{\beta_4^*}{Z_2(z)}\sum_{j=1}^N \left[\frac{\frac{Z_1(z)}{Z_2(z)}}{\frac{Z_1(z)}{Z_2(z)} - \sigma_j^2} - 1\right] \\
&\approx \frac{1}{\pi\bar\mu_S(\gamma_i)} \operatorname{Im} \lim_{z\to\gamma_i-\mathrm{i}0^+} \frac{\beta_4^* Z_1(z)}{Z_2(z)^2}\mathcal{G}_{\rho_Y}\Big(\frac{Z_1(z)}{Z_2(z)}\Big) - \frac{\beta_4^*}{Z_2(z)} \\
&= \frac{1}{\pi\bar\mu_S(\gamma_i)} \operatorname{Im} \lim_{z\to\gamma_i-\mathrm{i}0^+} q_4^*
\end{aligned}
\tag{78}
$$

where in the last equality we used the solution we have found in (71). Note that, based on (76), we do not need to have any knowledge about $\rho_Y$ to compute $q_4^*$. In the end, we need to divide the estimator by $\sqrt{\kappa}$ as we have absorbed it into $\boldsymbol{Y}$.

### D.3.1 Recovering the rectangular RIE for a denoising problem

Note that if in the model (60), we put $\boldsymbol{X} = \boldsymbol{I}$ the model reduces to the additive denoising of $\boldsymbol{Y}$, and we recover the estimator recently proposed in [19] for the rectangular case.

For $\boldsymbol{X} = \boldsymbol{I}$, $\mathcal{R}_{\rho_X}(z) = 1$, so (76) reduces to:

$$
\begin{cases}
\beta_1^* = \frac{\mathcal{C}_{\mu_W}^{(\alpha)}(q_1^* q_2^*)}{q_1^*}, & \beta_2^* = \alpha\frac{\mathcal{C}_{\mu_W}^{(\alpha)}(q_1^* q_2^*)}{q_2^*}, & \beta_3^* = 0, \quad \beta_4^* = 1 \\
q_1^* = \mathcal{G}_{\bar\mu_S}(z), \quad q_2^* = \alpha\mathcal{G}_{\bar\mu_S}(z) + (1-\alpha)\frac{1}{z} \\
q_3^* = (z - \beta_1^*)^2 \mathcal{G}_{\bar\mu_S}(z) - (z - \beta_1^*), \quad q_4^* = (z - \beta_1^*)\mathcal{G}_{\bar\mu_S}(z) - 1
\end{cases}
\tag{79}
$$

From (78), we have:

$$
\begin{aligned}
\widehat{\xi}^*_{y\,i} &= \frac{1}{\pi\bar{\mu}_S(\gamma_i)}\,\mathrm{Im}\,\lim_{z\to\gamma_i-\mathrm{i}0^+} q_4^* \\
&= \frac{1}{\pi\bar{\mu}_S(\gamma_i)}\,\mathrm{Im}\,\lim_{z\to\gamma_i-\mathrm{i}0^+} z\mathcal{G}_{\bar{\mu}_S}(z) - \beta_1^*\mathcal{G}_{\bar{\mu}_S}(z) - 1 \\
&= \frac{1}{\pi\bar{\mu}_S(\gamma_i)}\,\mathrm{Im}\,\lim_{z\to\gamma_i-\mathrm{i}0^+} z\mathcal{G}_{\bar{\mu}_S}(z) - \frac{\mathcal{C}^{(\alpha)}_{\mu_W}(q_1^*q_2^*)}{q_1^*}\mathcal{G}_{\bar{\mu}_S}(z) - 1 \\
&= \frac{1}{\pi\bar{\mu}_S(\gamma_i)}\,\mathrm{Im}\,\lim_{z\to\gamma_i-\mathrm{i}0^+} z\mathcal{G}_{\bar{\mu}_S}(z) - \mathcal{C}^{(\alpha)}_{\mu_W}(q_1^*q_2^*) - 1 \\
&= \frac{1}{\pi\bar{\mu}_S(\gamma_i)}\,\mathrm{Im}\,\lim_{z\to\gamma_i-\mathrm{i}0^+} z\mathcal{G}_{\bar{\mu}_S}(z) - \mathcal{C}^{(\alpha)}_{\mu_W}\Big(\mathcal{G}_{\bar{\mu}_S}(z)\big(\alpha\mathcal{G}_{\bar{\mu}_S}(z) + (1-\alpha)\tfrac{1}{z}\big)\Big) - 1 \\
&\overset{(a)}{=} \frac{1}{\pi\bar{\mu}_S(\gamma_i)}\mathrm{Im}\left[\gamma_i\mathcal{G}_{\bar{\mu}_S}(\gamma_i - \mathrm{i}0^+) - \mathcal{C}^{(\alpha)}_{\mu_W}\Big(\frac{1}{\gamma_i}\mathcal{G}_{\bar{\mu}_S}(\gamma_i - \mathrm{i}0^+)\big(1-\alpha+\alpha\gamma_i\mathcal{G}_{\bar{\mu}_S}(\gamma_i - \mathrm{i}0^+)\big)\Big)\right] \\
&\overset{(b)}{=} \gamma_i - \frac{1}{\pi\bar{\mu}_S(\gamma_i)}\mathrm{Im}\,\mathcal{C}^{(\alpha)}_{\mu_W}\Big(\frac{1-\alpha}{\gamma_i}\pi\mathsf{H}[\bar{\mu}_S](\gamma_i) + \alpha\big(\pi\mathsf{H}[\bar{\mu}_S](\gamma_i)\big)^2 - \alpha\big(\pi\bar{\mu}_S(\gamma_i)\big)^2 \\
&\qquad\qquad\qquad\qquad\qquad\qquad + \mathrm{i}\pi\bar{\mu}_S(\gamma_i)\big(\frac{1-\alpha}{\gamma_i} + 2\alpha\pi\mathsf{H}[\bar{\mu}_S](\gamma_i)\big)\Big)
\end{aligned}
$$
(80)

where in (a) we used the analyticity of rectangular R-transform [57], and in (b), we used Plemelj formula (6). Note that, the final estimator should be divided by the $\sqrt{\kappa}$.

## D.4  Examples

Throughout the numerical experiments, we consider the matrix $W$ to have i.i.d. Gaussian entries with variance $1/N$, so $\mathcal{C}^{(\alpha)}_{\mu_W}(z) = \frac{1}{\alpha}z$. And, $X = F + cI$ where $F = F^\mathsf{T} \in \mathbb{R}^{N\times N}$ has i.i.d. entries with variance $1/N$, and $c \neq 0$ is a real number, so $\mathcal{R}_{\rho_X}(z) = z + c$. With these choices, the solution (76) simplifies to:

$$
\begin{cases}
\beta_1^* = \frac{1}{\alpha}q_2^* + q_3^*, & \beta_2^* = q_1^*, & \beta_3^* = q_1^*, & \beta_4^* = q_4^* + c \\
q_1^* = \mathcal{G}_{\bar{\mu}_S}(z), & q_2^* = \alpha\mathcal{G}_{\bar{\mu}_S}(z) + (1-\alpha)\frac{1}{z} \\
q_3^* = \frac{(z-\beta_1^*)^2}{\beta_4^{*2}}\mathcal{G}_{\bar{\mu}_S}(z) - \frac{z-\beta_1^*}{\beta_4^{*2}}, & q_4^* = \frac{z-\beta_1^*}{\beta_4^*}\mathcal{G}_{\bar{\mu}_S}(z) - \frac{1}{\beta_4^*}
\end{cases}
$$
(81)

Note that in (81), $q_1^*, q_2^*$ are given in terms of the observation, so to find the solution we only need to find the parameters $q_3^*, q_4^*$. In (81), one can see that we have the relation $q_3^* = \frac{z-\beta_1^*}{\beta_4^*}q_4^*$. Writing the parameters $\beta_1^*, \beta_4^*$ in terms of $q_2^*, q_3^*, q_4^*$, after a bit of algebra we have the following relation:

$$
q_3^* = \frac{z - \frac{1}{\alpha}q_2^*}{2q_4^* + c}q_4^*
$$
(82)

In the expression for $q_4^*$ in (81), using (82) we can rewrite $\beta_1^*, \beta_4^*$ in terms of $q_2^*, q_4^*$. After some manipulations we find that $q_4^*$ is the solution to the following cubic equation:

$$
2x^3 + 3c\,x^2 + \left[c^2 + 2 - \Big(z - \mathcal{G}_{\bar{\mu}_S}(z) - \frac{1-\alpha}{\alpha}\frac{1}{z}\Big)\mathcal{G}_{\bar{\mu}_S}(z)\right]x - c\left[\Big(z - \mathcal{G}_{\bar{\mu}_S}(z) - \frac{1-\alpha}{\alpha}\frac{1}{z}\Big)\mathcal{G}_{\bar{\mu}_S}(z) - 1\right] = 0
$$
(83)

Based on our numerical simulations, we pick the following root for $q_4^*$:

$$
q_4^* = -\frac{c}{2} - \frac{12 - 3c^2 + 6A}{3\sqrt[3]{B}} + \frac{\sqrt[3]{B}}{12}
$$
(84)

with

$$
A = \mathcal{G}_{\bar{\mu}_S}(z)^2 - \frac{\mathcal{G}_{\bar{\mu}_S}(z)}{z}\Big(1 - \frac{1}{\alpha}\Big) - \mathcal{G}_{\bar{\mu}_S}(z)z
$$

$$
B = -216cA + 4\sqrt{4\big(12 - 3c^2 + 6A\big)^3 + 54^2c^2A^2}
$$

Once we have $q_4^*$, we can find $q_3^*$ using (82). In the end, $\beta_1^*, \cdots, \beta_4^*$ can be evaluated. Note that, for the RIE, only $q_4^*$ is required. Other parameters are used to evaluate the resolvent relation (72) and the overlap (77).

### D.4.1 Resolvent relation

We take $\kappa = 1$. In model (60), without loss of generality we can consider $\boldsymbol{Y}$ to be diagonal.

In figure 12, $\boldsymbol{Y}$ is the diagonal matrix obtained from the singular values of a Gaussian matrix with i.i.d. entries of variance $1/N$. In figure 13, the non-zero entries (on main diagonal) of $\boldsymbol{Y}$ are uniformly distributed in $[1, 3]$. As in previous cases, $\mu_S, \mathcal{G}_{\bar{\mu}_S}(z)$ are estimated numerically using Cauchy kernel, from which the parameters $\beta_1^*, \cdots, \beta_4^*$ are computed.

### D.4.2 Overlap

To illustrate the formula for the overlap (77), we fix the matrix $\boldsymbol{Y}$ and run experiments over various realization of the model (60). For each experiment, we record the overlap of $k$-th singular vectors left and right) of $\boldsymbol{S}$ and singular vectors of $\boldsymbol{Y}$. To compute the theoretical prediction, we evaluate the parameters $\beta_1^*, \beta_2^*, \beta_3^*, \beta_4^*$, for $z = \bar{\gamma}_k - \mathrm{i}0^+$ where $\bar{\gamma}_k$ is the average of $k$-th singular value of $\boldsymbol{S}$ in the experiments.

In figure 14a, the overlap is shown for $\boldsymbol{Y}$ with i.i.d. Gaussian entries of variance $\frac{1}{N}$, so $\mu_Y$ is the Marchenko-Pastur law with aspect-ratio $\alpha$. In figure 14b, matrix $\boldsymbol{Y}$ is constructed as $\boldsymbol{Y} = \boldsymbol{U}_Y \boldsymbol{\Sigma} \boldsymbol{V}_Y^\mathsf{T}$, where $\boldsymbol{U}_Y \in \mathbb{R}^{N \times N}, \boldsymbol{V}_Y \in \mathbb{R}^{M \times M}$ are Haar distributed orthogonal matrices, and singular values $\sigma_1, \cdots, \sigma_N$ are chosen independently uniformly from $[1, 3]$, so $\mu_Y = \mathcal{U}\big([1, 3]\big)$.

### D.4.3 RIE performance

In this section, we investigate the performance of our proposed estimators for $\boldsymbol{Y}$. To construct the RIE for $\boldsymbol{Y}$, we only need $q_4^*$ which we use (84). We compare performances of the optimal RIE (78) with the one of oracle estimator (5).

In figures 15,16, the MSE of RIE and the oracle estimator is plotted for three cases of priors: $\boldsymbol{Y}$ with Gaussian entries, $\boldsymbol{Y}$ with uniform spectral density, and $\boldsymbol{Y}$ with Bernoulli spectral density. In all cases, observe that the RIE has the same performance as the oracle estimator.

**Effect of aspect-ratio $\alpha$.** In Figure 17, we take $\boldsymbol{Y}$ to have Gaussian entries (with variance $\frac{1}{N}$), and the MSE is depicted for various values of the aspect-ratio $\alpha$. We see that as $M$ increases ($\alpha$ decreases) the estimation error (of $\boldsymbol{Y}$) decreases.

**Sparse $\boldsymbol{Y}$: a non-rotation invariant example.** We consider $\boldsymbol{Y}$ to have i.i.d. entries from the Bernoulli-Rademacher distribution,

$$Y_{i,j} = \begin{cases} +\frac{1}{\sqrt{N}} & \text{with probability } \frac{1-p}{2} \\ 0 & \text{with probability } p \\ -\frac{1}{\sqrt{N}} & \text{with probability } \frac{1-p}{2} \end{cases}, \qquad \forall \quad 1 \leq i \leq N, \quad 1 \leq j \leq M$$

With the normalization $1/\sqrt{N}$, the spectrum of $\boldsymbol{Y}$ does not grow with the dimension and has a finite support, thus we can apply our estimator to reconstruct $\boldsymbol{Y}$. *Note that the prior of $\boldsymbol{Y}$ is not rotationally invariant, and neither the oracle estimator nor the RIE are optimal.* Therefore, taking the prior into account, we apply a thresholding function on the entries of the matrix obtained from the RIE, $\widehat{\boldsymbol{\Xi}}_Y^*(\boldsymbol{S})$. We apply the following function on each entry of the estimator:

$$f_h(x) = \begin{cases} +\frac{1}{\sqrt{N}} & \text{if } x > \frac{h}{\sqrt{N}} \\ 0 & \text{if } |x| \leq \frac{h}{\sqrt{N}} \\ -\frac{1}{\sqrt{N}} & \text{if } x < -\frac{h}{\sqrt{N}} \end{cases}, \qquad \text{for } h \in [0, 1]$$

In figure 18, the MSE of the oracle estimator, RIE, and RIE+$f_p(x)$ (with $h = p$) is plotted. A few remarks on this figure are in order. First, RIEs are not limited to rotationally invariant priors and can give non-trivial estimates for non-rotationally invariant priors, although they are sub-optimal. The

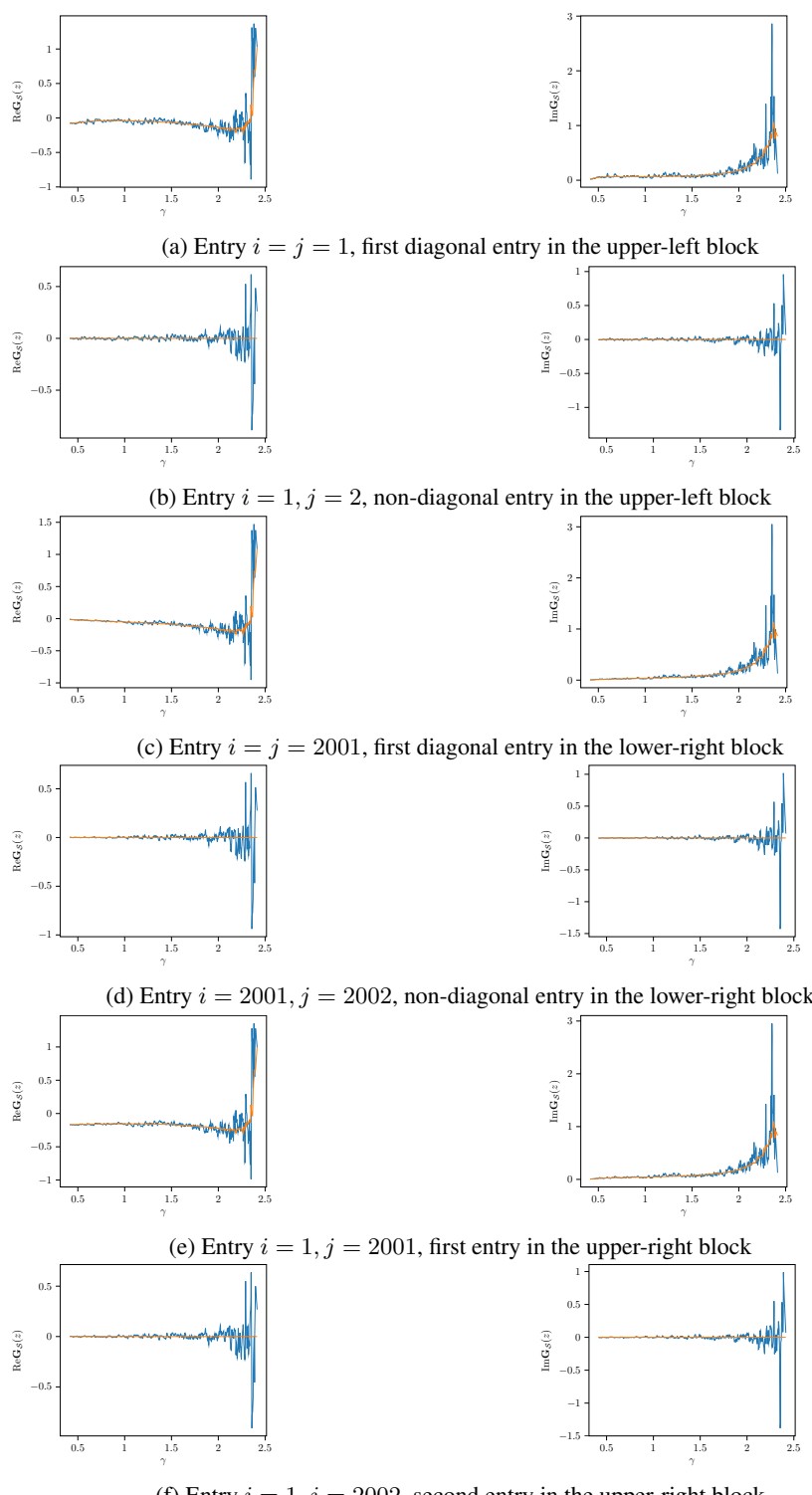

(a) Entry $i = j = 1$, first diagonal entry in the upper-left block

(b) Entry $i = 1, j = 2$, non-diagonal entry in the upper-left block

(c) Entry $i = j = 2001$, first diagonal entry in the lower-right block

(d) Entry $i = 2001, j = 2002$, non-diagonal entry in the lower-right block

(e) Entry $i = 1, j = 2001$, first entry in the upper-right block

(f) Entry $i = 1, j = 2002$, second entry in the upper-right block

Figure 12: Illustration of (72). $\boldsymbol{Y} \in \mathbb{R}^{N \times M}$ is a diagonal matrix obtained from the singular values of a $N \times M$ matrix with i.i.d. entries of variance $1/N$, $\boldsymbol{X} = \boldsymbol{X}^{\mathsf{T}}$ is shifted Wigner matrix with $c = 3$, and $\boldsymbol{Z}$ is a Gaussian matrices with. The empirical estimate of $\boldsymbol{G}_{\mathcal{S}}(z)$ (dashed blue line) is computed for $z = \gamma_i - \mathrm{i}\sqrt{\frac{1}{2N}}$ for $1 \le i \le N$, for $N = 2000$, $M = 4000$. Theoretical one (solid orange line) is computed from the rhs of (72) with parameters computed from the generated matrix. Note that, the theoretical one has also fluctuations because the parameters $\beta_1^*, \cdots \beta_4^*$ are computed from the numerical estimate of $\mathcal{G}_{\bar{\mu}_{\mathcal{S}}}(z)$.

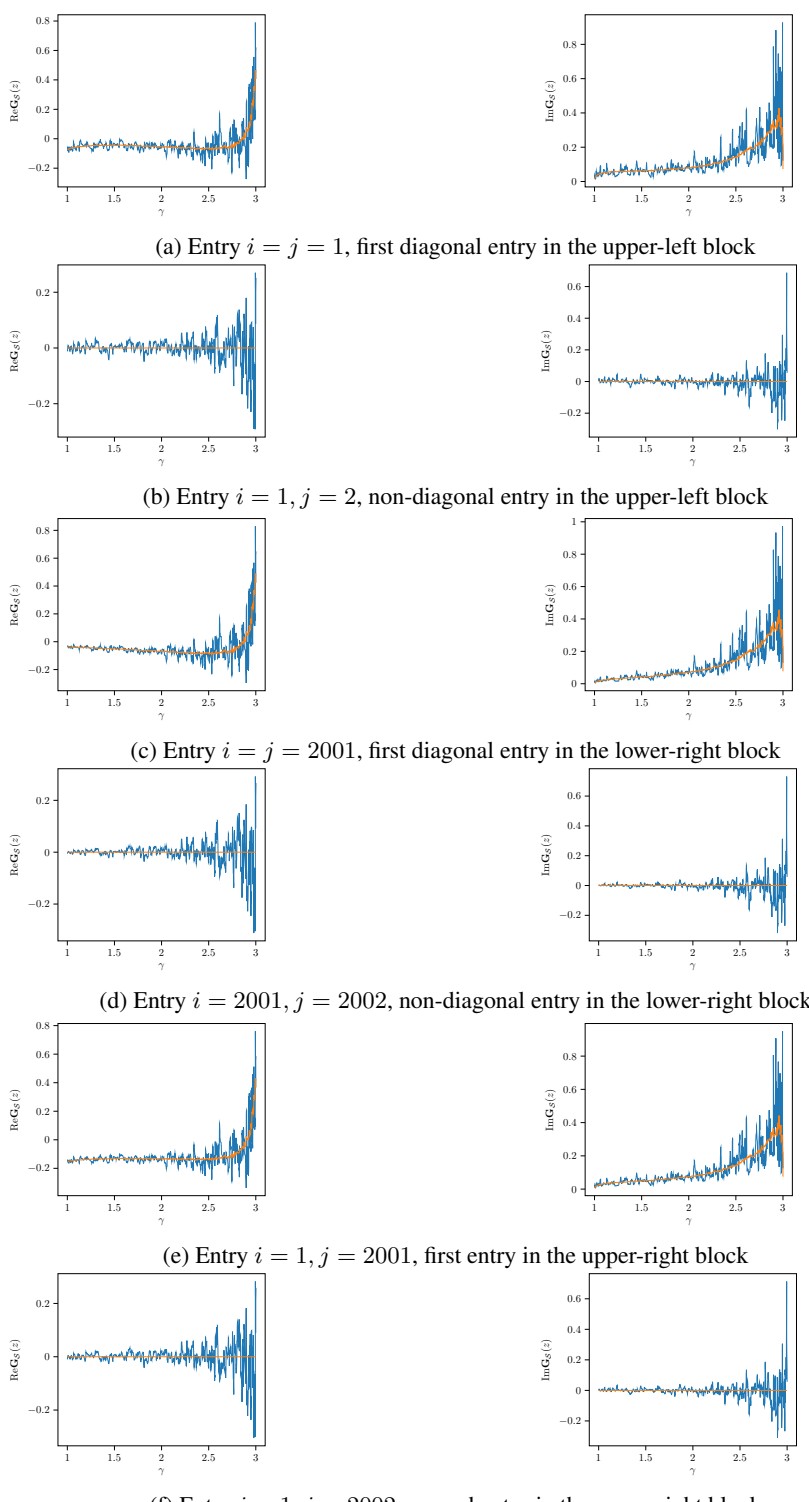

(a) Entry $i = j = 1$, first diagonal entry in the upper-left block

(b) Entry $i = 1, j = 2$, non-diagonal entry in the upper-left block

(c) Entry $i = j = 2001$, first diagonal entry in the lower-right block

(d) Entry $i = 2001, j = 2002$, non-diagonal entry in the lower-right block

(e) Entry $i = 1, j = 2001$, first entry in the upper-right block

(f) Entry $i = 1, j = 2002$, second entry in the upper-right block

Figure 13: Illustration of (72). $\boldsymbol{Y} \in \mathbb{R}^{N \times M}$ is a diagonal matrix with (main) diagonal entries uniformly distributed in $[1, 3]$, $\boldsymbol{X} = \boldsymbol{X}^{\mathsf{T}}$ is shifted Wigner matrix with $c = 3$, and $\boldsymbol{Z}$ is a Gaussian matrices with. The empirical estimate of $\boldsymbol{G}_{\mathcal{S}}(z)$ (dashed blue line) is computed for $z = \gamma_i - \mathrm{i}\sqrt{\frac{1}{2N}}$ for $1 \leq i \leq N$, for $N = 2000$, $M = 4000$. Theoretical one (solid orange line) is computed from the rhs of (72) with parameters computed from the generated matrix. Note that, the theoretical one has also fluctuations because the parameters $\beta_1^*, \cdots \beta_4^*$ are computed from the numerical estimate of $\mathcal{G}_{\bar{\mu}_S}(z)$.

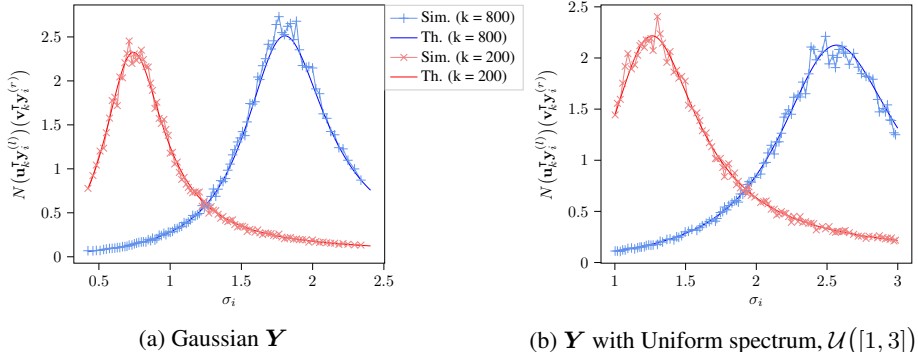

(a) Gaussian $Y$                    (b) $Y$ with Uniform spectrum, $\mathcal{U}\big([1,3]\big)$

Figure 14: Computation of the rescaled overlap. $X$ is a shifted Wigner matrix with $c = 3$, and $W$ has i.i.d. Gaussian entries of variance $1/N$, and $N/M = 1/2$. The simulation results are average of 1000 experiments with fixed $Y$, and $N = 1000, M = 2000$. Some of the simulation points are dropped for clarity.

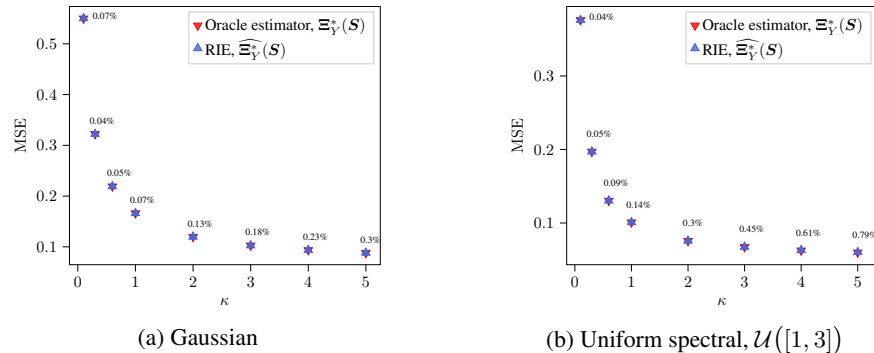

(a) Gaussian                    (b) Uniform spectral, $\mathcal{U}\big([1,3]\big)$

Figure 15: Estimating $Y$. MSE is normalized by the norm of the signal, $\|Y\|_{\mathrm{F}}^2$. $X$ is a shifted Wigner matrix with $c = 3$, and $W$ has i.i.d. Gaussian entries of variance $1/N$, and $N/M = 1/2$. The RIE is applied to $N = 2000, M = 4000$, and the results are averaged over 10 runs (error bars are invisible). Average relative error between RIE $\widehat{\Xi}_Y^*(S)$ and Oracle estimator is also reported.

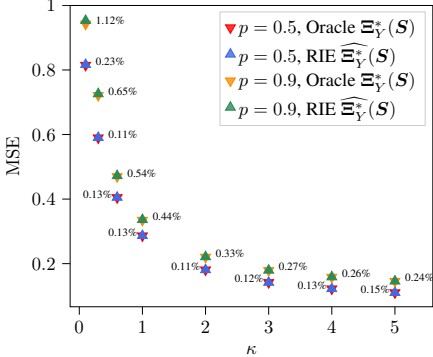

Figure 16: Estimating $Y$ with Bernoulli spectral prior. MSE is normalized by the norm of the signal, $\|Y\|_{\mathrm{F}}^2$. $Y$ has Bernoulli spectral distribution with parameter $p$. $X$ is a shifted Wigner matrix with $c = 3$, and $W$ has i.i.d. Gaussian entries of variance $1/N$, and $N/M = 1/2$. The RIE is applied to $N = 2000, M = 4000$, and the results are averaged over 10 runs (error bars are invisible). Average relative error between RIE $\widehat{\Xi}_Y^*(S)$ and Oracle estimator is also reported.

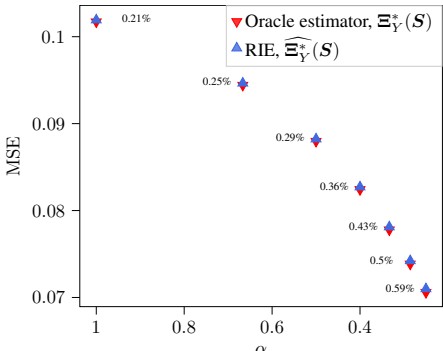

Figure 17: MSE of estimating $Y$ as a function of aspect-ratio $\alpha$, $Y$ has Gaussain entries of variance $1/N$, and $\kappa = 5$. MSE is normalized by the norm of the signal, $\|Y\|_F^2$. $X$ is a shifted Wigner matrix with $c = 3$, and $W$ has i.i.d. Gaussian entries of variance $1/N$. The RIE is applied to $N = 2000$, $M = 1/\alpha N$, and the results are averaged over 10 runs (error bars are invisible). Average relative error between RIE $\widehat{\Xi_Y^*}(S)$ and Oracle estimator is also reported.

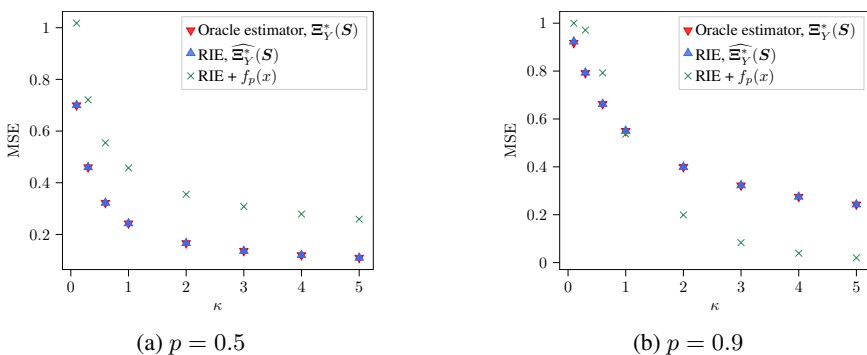

(a) $p = 0.5$                    (b) $p = 0.9$

Figure 18: Estimating $Y$ with Bernoulli-Rademacher *entries*. MSE is normalized by the norm of the signal, $\|Y\|_F^2$. $X$ is a shifted Wigner matrix with $c = 3$, and $W$ has i.i.d. Gaussian entries of variance $1/N$, and $N/M = 1/2$. The RIE is applied to $N = 2000$, $M = 4000$, and the results are averaged over 10 runs (error bars are invisible).

RIE's output can be refined, or used as a warmed-up initialization for other algorithms to get a better estimate.

In figure 19, for one experiment, the MSE is plotted for RIE and RIE+$f(x)$ with the best $h$ among $\{0, 0.1, \cdots, 1\}$. We observe that for the particular case of Bernoulli-Rademacher prior, the thresholding stage can improve the MSE when SNR is greater than 1, however the parameter $h$ should be chosen properly.

## E    Comparison of RIEs for MF and denoising

For estimating $X$, we have derived the estimator (49) for general priors $\rho_X, \mu_Y, \mu_W$. This estimator simplifies greatly, with parameters in (52), when both $\mu_Y, \mu_W$ are Marchenko-Pastur distribution, i.e. both $Y, W$ having i.i.d. Gaussian entries of variance $1/N$. Similarly, although the RIE for $Y$ in (78) is derived for the general priors, it reduces to a rather simple estimator if $\rho_X, \mu_W$ are taken to be shifted Wigner, and Marchenko-Pastur distribution, respectively. Therefore, in our numerical examples on factorization problem, we consider $X$ to be a shifted Wigner matrix, and $Y, W$ to be Gaussian matrices.

In each experiment, the factors $X, Y$ are estimated simultaneously using RIE from the observation matrix $S$. In addition to the MSE of estimating each factor, we also compute the MSE of estimating the product $XY$. We compare the MSE of the product with the MSE of the oracle estimator and the RIE introduced in [19] for the denoising problem. The oracle estimator for the denoising is

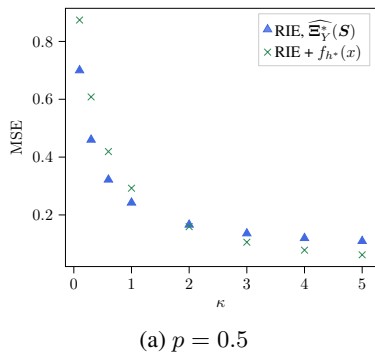
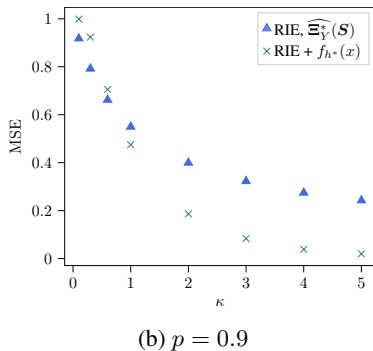

(a) $p = 0.5$                                      (b) $p = 0.9$

Figure 19: Estimating $Y$ with Bernoulli-Rademacher *entries*. MSE is normalized by the norm of the signal, $\|Y\|_F^2$. $X$ is a shifted Wigner matrix with $c = 3$, and $W$ has i.i.d. Gaussian entries of variance $1/N$, and $N/M = 1/2$. The RIE is applied to $N = 2000$, $M = 4000$, and thresholding function is applied with the best $h$ among $\{0, 0.1, \cdots, 1\}$. Results are averaged over 10 runs (error bars are invisible).

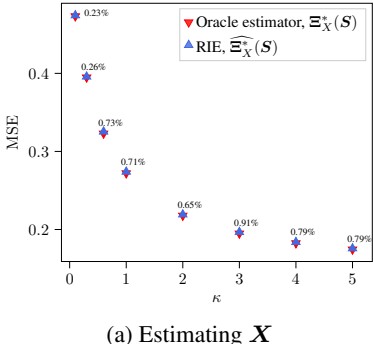
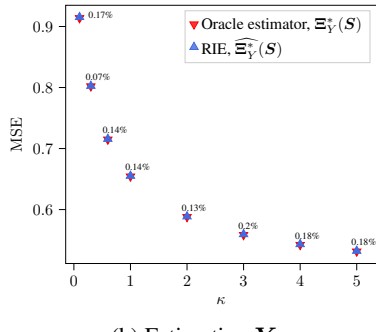

(a) Estimating $X$                                 (b) Estimating $Y$

Figure 20: MSE of factorization problem. MSE is normalized by the norm of the signal. $X$ is a shifted Wigner matrix with $c = 1$, and both $Y$ and $W$ are $N \times M$ matrices with i.i.d. Gaussian entries of variance $1/N$, and $N/M = 1/2$. The RIE is applied to $N = 2000$, $M = 4000$. In each run, the observation matrix $S$ is generated according to (1), and the factors $X$, $Y$ are estimated simultaneously from $S$. Results are averaged over 10 runs (error bars are invisible). Average relative error between RIEs and Oracle estimators is also reported.

constructed as:

$$\Xi_{XY}^*(S) = \sum_{i=1}^{N} \xi_{xy_i}^* \, u_i v_i^\mathsf{T}, \quad \xi_{xy_i}^* = u_i^\mathsf{T} X Y v_i \tag{85}$$

where $u_i, v_i$'s are left/right singular vectors of $S$. In the RIE proposed in [19], the singular values are estimated by (see section D.3.1)

$$\widehat{\xi_{xy_i}^*} = \frac{1}{\sqrt{\kappa}} \left[ \gamma_i - \frac{1}{\pi \bar{\mu}_S(\gamma_i)} \mathrm{Im} \, \mathcal{C}_{\mu_W}^{(\alpha)} \left( \frac{1-\alpha}{\gamma_i} \pi \mathsf{H}[\bar{\mu}_S](\gamma_i) + \alpha \big( \pi \mathsf{H}[\bar{\mu}_S](\gamma_i) \big)^2 - \alpha \big( \pi \bar{\mu}_S(\gamma_i) \big)^2 \right. \right.$$
$$\left. \left. + \mathrm{i} \pi \bar{\mu}_S(\gamma_i) \big( \frac{1-\alpha}{\gamma_i} + 2\alpha \pi \mathsf{H}[\bar{\mu}_S](\gamma_i) \big) \right) \right] \tag{86}$$

Note that, in general the MSE of the denoising RIE $\widehat{\Xi_{XY}^*}(S)$, is less than the MSE of the prdouct of the estimated factors $\widehat{\Xi_X^*}(S)\widehat{\Xi_Y^*}(S)$.

In figures 20,21, the MSE of estimating the factors is illustrated for $c = 1$ and $c = 3$ respectively. The MSE of estimating the product is shown in figure 22.

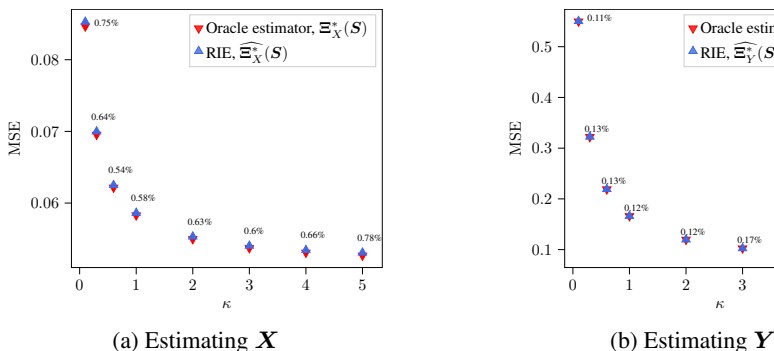

(a) Estimating $X$            (b) Estimating $Y$

Figure 21: MSE of factorization problem. MSE is normalized by the norm of the signal. $X$ is a shifted Wigner matrix with $c = 3$, and both $Y$ and $W$ are $N \times M$ matrices with i.i.d. Gaussian entries of variance $1/N$, and $N/M = 1/2$. The RIE is applied to $N = 2000$, $M = 4000$. In each run, the observation matrix $S$ is generated according to (1), and the factors $X$, $Y$ are estimated simultaneously from $S$. Results are averaged over 10 runs (error bars are invisible). Average relative error between RIEs and Oracle estimators is also reported.

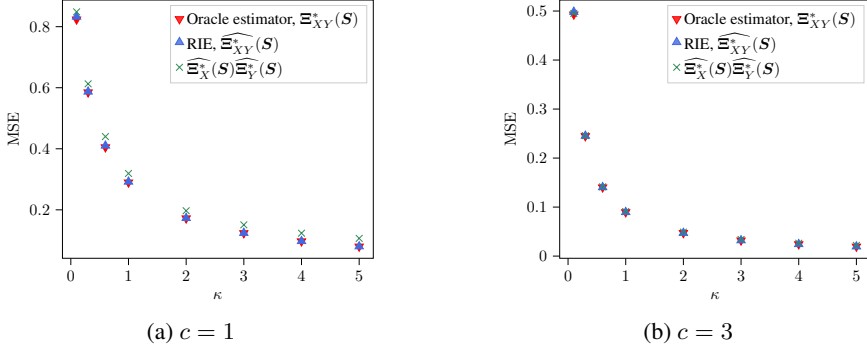

(a) $c = 1$            (b) $c = 3$

Figure 22: MSE of the product of the factors. MSE is normalized by the norm of the signal $\|XY\|_F^2$. $X$ is a shifted Wigner matrix with $c = 1, c = 3$, and both $Y$ and $W$ are $N \times M$ matrices with i.i.d. Gaussian entries of variance $1/N$, and $N/M = 1/2$. The RIE is applied to $N = 2000$, $M = 4000$. Results are averaged over 10 runs (error bars are invisible).

## F    Case of $\alpha \geq 1$

In this section we consider the case where $M \leq N$ and $N/M \to \alpha \geq 1$ as $N \to \infty$. Throughout this section $\Gamma \in \mathbb{R}^{N \times M}$ is a (tall) matrix with $\Gamma_M$ in its upper $M \times M$ block, and the rest zero entries. $\Gamma_M$ is diagonal matrix constructed from $\gamma \in \mathbb{R}^M$ which are the singular values of $S$.

Similar to the case of $\alpha \leq 1$, resolvent of the matrix $\mathcal{S} \in \mathbb{R}^{(N+M) \times (N+M)}$ plays a central role in deriving the RIEs. For the case of $M \geq N$, with $S = U_S \Gamma V_S^\mathsf{T}$, the matrix $\mathcal{S}$ has the following eigen-decomposition:

$$\mathcal{S} = \begin{bmatrix} \hat{U}_S^{(1)} & -\hat{U}_S^{(1)} & U_S^{(2)} \\ \hat{V}_S & -\hat{V}_S & 0 \end{bmatrix} \begin{bmatrix} \Gamma_M & 0 & 0 \\ 0 & -\Gamma_M & 0 \\ 0 & 0 & 0 \end{bmatrix} \begin{bmatrix} \hat{U}_S^{(1)} & -\hat{U}_S^{(1)} & U_S^{(2)} \\ \hat{V}_S & -\hat{V}_S & 0 \end{bmatrix}^\mathsf{T} \quad (87)$$

with $U_S = \begin{bmatrix} U_S^{(1)} & U_S^{(2)} \end{bmatrix}$ in which $U_S^{(1)} \in \mathbb{R}^{N \times M}$. And, $\hat{U}_S^{(1)} = \frac{1}{\sqrt{2}} U_S^{(1)}$, $\hat{V}_S = \frac{1}{\sqrt{2}} V_S$. The resolvent of $\mathcal{S}$ can be written as:

$$G_{\mathcal{S}}(x - \mathrm{i}\epsilon) = \sum_{k=1}^{2M} \frac{x + \mathrm{i}\epsilon}{(x - \tilde{\gamma}_k)^2 + \epsilon^2} \mathbf{s}_k \mathbf{s}_k^\mathsf{T} + \frac{x + \mathrm{i}\epsilon}{x^2 + \epsilon^2} \sum_{k=2M+1}^{M+N} \mathbf{s}_k \mathbf{s}_k^\mathsf{T}$$

where $\tilde{\gamma}_k$ are the eigenvalues of $\mathcal{S}$, which are in fact the (signed) singular values of $S$, $\tilde{\gamma}_1 = \gamma_1, \ldots, \tilde{\gamma}_M = \gamma_M, \tilde{\gamma}_{M+1} = -\gamma_1, \ldots, \tilde{\gamma}_{2M} = -\gamma_M$.

## F.1 Estimating $X$

The RIE for $X$ is constructed in the same way as in the case of $\alpha \leq 1$, (2). However, in the present case the observation matrix $S$ has $M$ (non-trivially zero) singular values and we need to estimate $N$ eigenvalues for the RIE. As it will be clear, the $N - M$ eigenvalues are chosen to be equal.

### F.1.1 Relation between overlap and the resolvent

Define the vectors $\tilde{x}_i = [x_i^\mathsf{T}, \mathbf{0}_M]^\mathsf{T}$ for $x_i$ eigenvectors of $X$. We have

$$\tilde{x}_i^\mathsf{T}\big(\operatorname{Im}G_S(x - \mathrm{i}\epsilon)\big)\tilde{x}_i = \sum_{k=1}^{2M} \frac{\epsilon}{(x - \tilde{\gamma}_k)^2 + \epsilon^2}\big(\tilde{x}_i^\mathsf{T}\mathbf{s}_k\big)^2 + \frac{\epsilon}{x^2 + \epsilon^2}\sum_{k=2M+1}^{M+N}\big(\tilde{x}_i^\mathsf{T}\mathbf{s}_k\big)^2 \qquad (88)$$

Given the structure of $\mathbf{s}_k$'s in (87), we have:

$$\big(\tilde{x}_i^\mathsf{T}\mathbf{s}_k\big)^2 = \begin{cases} \frac{1}{2}\big(x_i^\mathsf{T}u_k\big)^2 & \text{for } 1 \leq k \leq M \\ \frac{1}{2}\big(x_i^\mathsf{T}u_{k-M}\big)^2 & \text{for } M+1 \leq k \leq 2M \\ \big(x_i^\mathsf{T}u_{k-M}\big)^2 & \text{for } 2M+1 \leq k \leq M+N \end{cases}$$

We assume that in the limit of large N this quantity concentrates on $O_X(\gamma_j, \lambda_i)$ and depends only on the singular values and eigenvalue pairs $(\gamma_j, \lambda_i)$. This assumption implies that the singular vectors associated with 0 singular values ($u_j$ for $M + 1 \leq j \leq N$) all have the same overlap with the eigenvectors of $X$, $O_X(0, \lambda_i)$. We thus have:

$$\tilde{x}_i^\mathsf{T}\big(\operatorname{Im}G_S(x - \mathrm{i}\epsilon)\big)\tilde{x}_i \xrightarrow{N\to\infty} \frac{1}{\alpha}\int_{\mathbb{R}}\frac{\epsilon}{(x-t)^2 + \epsilon^2}O_X(t, \lambda_i)\bar{\mu}_S(t)\,dt + \big(1 - \frac{1}{\alpha}\big)\frac{\epsilon}{x^2 + \epsilon^2}O_X(0, \lambda_i) \tag{89}$$

where the overlap function $O_X(t, \lambda_i)$ is extended (continuously) to arbitrary values within the support of $\bar{\mu}_S$ (the symmetrized limiting singular value distribution of $S$) with the property that $O_X(t, \lambda_i) = O_X(-t, \lambda_i)$ for $t \in \operatorname{supp}(\mu_S)$. Sending $\epsilon \to 0$, we find

$$\tilde{x}_i^\mathsf{T}\big(\operatorname{Im}G_S(x - \mathrm{i}\epsilon)\big)\tilde{x}_i \to \frac{1}{\alpha}\pi\bar{\mu}_S(x)O_X(x, \lambda_i) + \big(1 - \frac{1}{\alpha}\big)\pi\delta(x)O_X(x, \lambda_i) \tag{90}$$

### F.1.2 Resolvent relation

We derive the resolvent relation for the same model as in (29). The derivation is similar to the procedure explained in section C.1, and we omit here. The final resolvent relation is the same as (42), with parameters satisfying:

$$\begin{cases} \zeta_1^* = \frac{1}{\alpha}\frac{\mathcal{C}_{\mu_W}^{(1/\alpha)}(p_1^*p_2^*)}{p_1^*}, \quad \zeta_2^* = \frac{1}{p_2^*}\big(\mathcal{C}_{\mu_W}^{(1/\alpha)}(p_1^*p_2^*) + \mathcal{C}_{\mu_Y}^{(1/\alpha)}(p_2^*p_3^*)\big), \quad \zeta_3^* = \frac{1}{\alpha}\frac{\mathcal{C}_{\mu_Y}^{(1/\alpha)}(p_2^*p_3^*)}{p_3^*} \\[2mm] p_1^* = \frac{1}{\zeta_3^*}\mathcal{G}_{\rho_{X^2}}\big(\frac{z-\zeta_1^*}{\zeta_3^*}\big), \quad p_2^* = \frac{1}{z-\zeta_2^*}, \quad p_3^* = \frac{z-\zeta_1^*}{\zeta_3^{*2}}\mathcal{G}_{\rho_{X^2}}\big(\frac{z-\zeta_1^*}{\zeta_3^*}\big) - \frac{1}{\zeta_3^*} \end{cases} \tag{91}$$

Again, with the same procedure as (43),(44), the saddle point equations (91) can be rewritten in a simplified form, which does not involve $\rho_{X^2}$, as:

$$\begin{cases} \zeta_1^* = \frac{1}{\alpha}\frac{\mathcal{C}_{\mu_W}^{(1/\alpha)}(p_1^*p_2^*)}{p_1^*}, \quad \zeta_2^* = z - \frac{1}{\mathcal{G}_{\bar{\mu}_S}(z)}, \quad \zeta_3^* = \frac{1}{\alpha}\frac{\mathcal{C}_{\mu_Y}^{(1/\alpha)}(p_2^*p_3^*)}{p_3^*} \\[2mm] p_1^* = \frac{1}{\alpha}\mathcal{G}_{\bar{\mu}_S}(z) + \big(1 - \frac{1}{\alpha}\big)\frac{1}{z}, \quad p_2^* = \mathcal{G}_{\bar{\mu}_S}(z), \quad p_3^* = \frac{z-\zeta_1^*}{\alpha\zeta_3^*}\mathcal{G}_{\bar{\mu}_S}(z) + \frac{z-\zeta_1^*}{\zeta_3^*}\big(1 - \frac{1}{\alpha}\big)\frac{1}{z} - \frac{1}{\zeta_3^*} \end{cases} \tag{92}$$

with $\bar{\mu}_S$ the limiting ESD of non-trivial singular values of $S$. Note that $\zeta_1^*, \zeta_2^*$ can be computed from the observation matrix, and we only need to find $\zeta_3^*$ satisfying the following equation:

$$(z - \zeta_1^*)\big[\frac{1}{\alpha}\mathcal{G}_{\bar{\mu}_S}(z) + \big(1 - \frac{1}{\alpha}\big)\frac{1}{z}\big] - 1 = \frac{1}{\alpha}\mathcal{C}_{\mu_Y}^{(1/\alpha)}\Big(\frac{1}{\zeta_3^*}\mathcal{G}_{\bar{\mu}_S}(z)(z - \zeta_1^*)\big[\frac{1}{\alpha}\mathcal{G}_{\bar{\mu}_S}(z) + \big(1 - \frac{1}{\alpha}\big)\frac{1}{z}\big]\Big) \tag{93}$$

Note that both sets of equations (90), (92) and (47), (45) match for $\alpha = 1$.

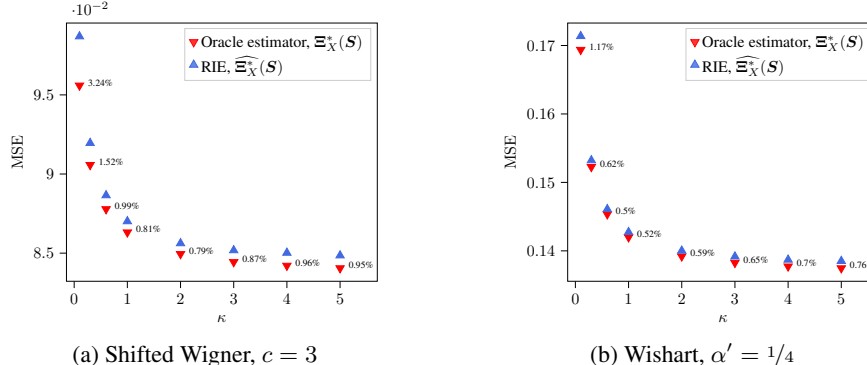

(a) Shifted Wigner, $c = 3$           (b) Wishart, $\alpha' = 1/4$

Figure 23: Estimating $\boldsymbol{X}$. The MSE is normalized by the norm of the signal, $\|\boldsymbol{X}\|_{\mathrm{F}}^2$. Both $\boldsymbol{Y}$ and $\boldsymbol{W}$ are $N \times M$ matrices with i.i.d. Gaussian entries of variance $1/N$, and aspect ratio $N/M = 2$. The RIE is applied to $N = 2000$, $M = 1000$, and the results are averaged over 10 runs (error bars are invisible). Average relative error between RIE $\widehat{\boldsymbol{\Xi}_X^*}(\boldsymbol{S})$ and Oracle estimator is also reported.

### F.1.3   Overlaps and optimal eigenvalues

From (90), (42), for $\gamma$ a non-trivially zero singular value of $\boldsymbol{S}$ we find:

$$
\begin{aligned}
O_X(\gamma, \lambda_i) &\approx \frac{\alpha}{\pi \bar{\mu}_S(\gamma)} \operatorname{Im} \lim_{z \to \gamma - \mathrm{i}0^+} \boldsymbol{x}_i^\mathsf{T} \zeta_3^{*-1} \boldsymbol{G}_{X^2}\Big(\frac{z - \zeta_1^*}{\zeta_3^*}\Big) \boldsymbol{x}_i \\
&= \frac{\alpha}{\pi \bar{\mu}_S(\gamma)} \operatorname{Im} \lim_{z \to \gamma - \mathrm{i}0^+} \frac{1}{z - \zeta_1^* - \zeta_3^* \lambda_i^2}
\end{aligned}
\tag{94}
$$

And, in the case of $M > N$, for zero singular values we have:

$$
\begin{aligned}
O_X(0, \lambda_i) &\approx \frac{\alpha}{(\alpha - 1)\pi} \operatorname{Im} \lim_{z \to -\mathrm{i}0^+} \boldsymbol{x}_i^\mathsf{T} \zeta_3^{*-1} \boldsymbol{G}_{X^2}\Big(\frac{z - \zeta_1^*}{\zeta_3^*}\Big) \boldsymbol{x}_i \\
&= \frac{\alpha}{(\alpha - 1)\pi} \operatorname{Im} \lim_{z \to -\mathrm{i}0^+} \frac{1}{z - \zeta_1^* - \zeta_3^* \lambda_i^2}
\end{aligned}
\tag{95}
$$

Finally, the optimal eigenvalues can be derived in the same way as in (49). For $1 \le i \le M$, we have:

$$
\widehat{\xi}_{xi}^* = \frac{\alpha}{2\kappa\pi \bar{\mu}_S(\gamma_i)} \operatorname{Im} \lim_{z \to \gamma_i - \mathrm{i}0^+} \left\{ \frac{1}{\zeta_3^*} \Big[ \mathcal{G}_{\rho_X}\Big(\sqrt{\frac{z - \zeta_1^*}{\kappa \zeta_3^*}}\Big) + \mathcal{G}_{\rho_X}\Big(-\sqrt{\frac{z - \zeta_1^*}{\kappa \zeta_3^*}}\Big) \Big] \right\}
\tag{96}
$$

And, for all $M + 1 \le i \le N$:

$$
\widehat{\xi}_{xi}^* = \frac{\alpha}{2\kappa(\alpha - 1)\pi} \operatorname{Im} \lim_{z \to -\mathrm{i}0^+} \left\{ \frac{1}{\zeta_3^*} \Big[ \mathcal{G}_{\rho_X}\Big(\sqrt{\frac{z - \zeta_1^*}{\kappa \zeta_3^*}}\Big) + \mathcal{G}_{\rho_X}\Big(-\sqrt{\frac{z - \zeta_1^*}{\kappa \zeta_3^*}}\Big) \Big] \right\}
\tag{97}
$$

### F.1.4   Numerical Examples

For matrices $\boldsymbol{Y}, \boldsymbol{W} \in \mathbb{R}^{N \times M}$ with i.i.d. Gaussian entries of variance $1/N$ and $M > N$, we have that $\mathcal{C}_{\mu_Y}^{(1/\alpha)}(z) = \mathcal{C}_{\mu_W}^{(1/\alpha)}(z) = z$ which leads to a simplification of equations (92):

$$
\begin{cases}
\zeta_1^* = \frac{1}{\alpha} p_2^*, \quad \zeta_2^* = z - \frac{1}{\mathcal{G}_{\bar{\mu}_S}(z)}, \quad \zeta_3^* = \frac{1}{\alpha} p_2^* \\[2mm]
p_1^* = \frac{1}{\alpha} \mathcal{G}_{\bar{\mu}_S}(z) + \Big(1 - \frac{1}{\alpha}\Big)\frac{1}{z}, \quad p_2^* = \mathcal{G}_{\bar{\mu}_S}(z), \quad p_3^* = \frac{z - \zeta_1^*}{\alpha \zeta_3^*} \mathcal{G}_{\bar{\mu}_S}(z) + \frac{z - \zeta_1^*}{\zeta_3^*}\Big(1 - \frac{1}{\alpha}\Big)\frac{1}{z} - \frac{1}{\zeta_3^*}
\end{cases}
\tag{98}
$$

Therefore, $\zeta_1^* = \zeta_3^* = \frac{1}{\alpha} \mathcal{G}_{\bar{\mu}_S}(z)$.

In Figure 23, the MSE of the Oracle estimator and the RIE (96), (97) is illustrated for shifted Wigner $\boldsymbol{X}$ with $c = 3$, and Wishart with aspect-ratio $\alpha' = 1/4$.

**Effect of aspect-ratio $\alpha$.**   In Figure 24, we take $\boldsymbol{X}$ to be a shifted Wigner matrix with $c = 3$, and the MSE is depicted for various values of the aspect-ratio $\alpha > 1$. We see that as $M$ decreases ($\alpha$ increases) the estimation error (of $\boldsymbol{Y}$) increases.

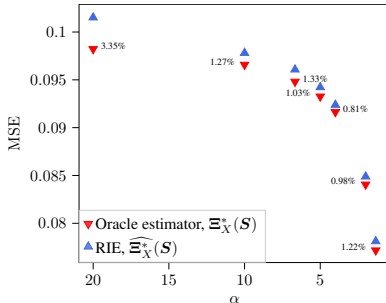

Figure 24: MSE of estimating $X$ as a function of aspect-ratio $\alpha > 1$, prior on $X$ is shifted Wigner with $c = 3$, and $\kappa = 5$. MSE is normalized by the norm of the signal, $\|X\|_F^2$. Both $Y$ and $W$ are $N \times M$ matrices with i.i.d. Gaussian entries of variance $1/N$. The RIE is applied to $N = 2000$, $M = 1/\alpha N$, and the results are averaged over 10 runs (error bars are invisible). Average relative error between RIE $\widehat{\Xi}_X^*(S)$ and Oracle estimator is also reported.

## F.2 Estimating $Y$

### F.2.1 Relation between overlap and the resolvent

For the vectors $r_i = \begin{bmatrix} \mathbf{0}_N \\ y_i^{(r)} \end{bmatrix}$, $l_i = \begin{bmatrix} y_i^{(l)} \\ \mathbf{0}_M \end{bmatrix}$ with $y_i^{(r)}, y_i^{(l)}$ right/ left singular vectors of $Y$, we have

$$r_i^\mathsf{T}\big(\operatorname{Im} G_S(x - i\epsilon)\big)l_i = \sum_{k=1}^{2M} \frac{\epsilon}{(x - \tilde{\gamma}_k)^2 + \epsilon^2}\big(r_i^\mathsf{T}s_k\big)\big(l_i^\mathsf{T}s_k\big) + \frac{\epsilon}{x^2 + \epsilon^2}\sum_{k=2M+1}^{M+N}\big(r_i^\mathsf{T}s_k\big)\big(l_i^\mathsf{T}s_k\big) \tag{99}$$

Given the structure of $s_k$'s in (87), we have:

$$\big(r_i^\mathsf{T}s_k\big)\big(l_i^\mathsf{T}s_k\big) = \begin{cases} \frac{1}{2}\big(u_k^\mathsf{T}y_i^{(l)}\big)\big(v_k^\mathsf{T}y_i^{(r)}\big) & \text{for } 1 \le k \le M \\ -\frac{1}{2}\big(u_{k-M}^\mathsf{T}y_i^{(l)}\big)\big(v_{k-M}^\mathsf{T}y_i^{(r)}\big) & \text{for } M + 1 \le k \le 2M \\ 0 & \text{for } 2M + 1 \le k \le N + M \end{cases}$$

Therefore, in the limit $N \to \infty$, we have:

$$r_i^\mathsf{T}\big(\operatorname{Im} G_S(x - i\epsilon)\big)l_i \xrightarrow{N\to\infty} \frac{1}{\alpha}\int_{\mathbb{R}} \frac{\epsilon}{(x - t)^2 + \epsilon^2}O_Y(t, \sigma_i)\bar{\mu}_S(t)\,dt \tag{100}$$

where the overlap function $O_Y(t, \lambda_i)$ is extended (continuously) to arbitrary values within the support of $\bar{\mu}_S$ with the property that $O_Y(-t, \lambda_i) = -O_Y(t, \lambda_i)$ for $t \in \operatorname{supp}(\mu_S)$. Sending $\epsilon \to 0$, we find

$$r_i^\mathsf{T}\big(\operatorname{Im} G_S(x - i\epsilon)\big)l_i \approx \frac{1}{\alpha}\pi\bar{\mu}_S(x)O_Y(x, \sigma_i) \tag{101}$$

### F.2.2 Resolvent relation

The resolvent relation for the model (60) with $M < N$ is the same as in (72) with parameters satisfying:

$$
\begin{cases}
\beta_1^* = \frac{1}{\alpha}\frac{\mathcal{C}_{\mu_W}^{(\alpha)}(q_1^* q_2^*)}{q_1^*} + \frac{1}{2}\sqrt{\frac{q_3^*}{q_1^*}}\Big(\mathcal{R}_{\rho_X}\big(q_4^* + \sqrt{q_1^* q_3^*}\big) - \mathcal{R}_{\rho_X}\big(q_4^* - \sqrt{q_1^* q_3^*}\big)\Big) \\
\beta_2^* = \frac{\mathcal{C}_{\mu_W}^{(\alpha)}(q_1^* q_2^*)}{q_2^*} \\
\beta_3^* = \frac{1}{2}\sqrt{\frac{q_1^*}{q_3^*}}\Big(\mathcal{R}_{\rho_X}\big(q_4^* + \sqrt{q_1^* q_3^*}\big) - \mathcal{R}_{\rho_X}\big(q_4^* - \sqrt{q_1^* q_3^*}\big)\Big) \\
\beta_4^* = \frac{1}{2}\Big(\mathcal{R}_{\rho_X}\big(q_4^* + \sqrt{q_1^* q_3^*}\big) + \mathcal{R}_{\rho_X}\big(q_4^* - \sqrt{q_1^* q_3^*}\big)\Big) \qquad \text{with } \begin{cases} Z_1(z) = (z-\beta_1^*)(z-\beta_2^*) \\ Z_2(z) = \beta_4^{*\,2} + \beta_3^*(z-\beta_1^*) \end{cases} \\
q_1^* = \frac{1}{\alpha}\frac{(z-\beta_2^*)\beta_4^{*\,2}}{Z_2(z)^2}\mathcal{G}_{\rho_Y}\big(\frac{Z_1(z)}{Z_2(z)}\big) + \frac{1}{\alpha}\frac{\beta_3^*}{Z_2(z)} + \frac{\alpha-1}{\alpha}\frac{1}{z-\beta_1^*} \\
q_2^* = \frac{z-\beta_1^*}{Z_2(z)}\mathcal{G}_{\rho_Y}\big(\frac{Z_1(z)}{Z_2(z)}\big) \\
q_3^* = \frac{1}{\alpha}\frac{(z-\beta_1^*)Z_1(z)}{Z_2(z)^2}\mathcal{G}_{\rho_Y}\big(\frac{Z_1(z)}{Z_2(z)}\big) - \frac{1}{\alpha}\frac{z-\beta_1^*}{Z_2(z)} \\
q_4^* = \frac{1}{\alpha}\frac{\beta_4^* Z_1(z)}{Z_2(z)^2}\mathcal{G}_{\rho_Y}\big(\frac{Z_1(z)}{Z_2(z)}\big) - \frac{1}{\alpha}\frac{\beta_4^*}{Z_2(z)}
\end{cases}
\tag{102}
$$

With the same procedure as (73),(74), the saddle point equations (102) can be rewritten in a simplified form:

$$
\begin{cases}
\beta_1^* = \frac{1}{\alpha}\frac{\mathcal{C}_{\mu_W}^{(\alpha)}(q_1^* q_2^*)}{q_1^*} + \frac{1}{2}\sqrt{\frac{q_3^*}{q_1^*}}\Big(\mathcal{R}_{\rho_X}\big(q_4^* + \sqrt{q_1^* q_3^*}\big) - \mathcal{R}_{\rho_X}\big(q_4^* - \sqrt{q_1^* q_3^*}\big)\Big) \\
\beta_2^* = \frac{\mathcal{C}_{\mu_W}^{(\alpha)}(q_1^* q_2^*)}{q_2^*} \\
\beta_3^* = \frac{1}{2}\sqrt{\frac{q_1^*}{q_3^*}}\Big(\mathcal{R}_{\rho_X}\big(q_4^* + \sqrt{q_1^* q_3^*}\big) - \mathcal{R}_{\rho_X}\big(q_4^* - \sqrt{q_1^* q_3^*}\big)\Big) \\
\beta_4^* = \frac{1}{2}\Big(\mathcal{R}_{\rho_X}\big(q_4^* + \sqrt{q_1^* q_3^*}\big) + \mathcal{R}_{\rho_X}\big(q_4^* - \sqrt{q_1^* q_3^*}\big)\Big) \\
q_1^* = \frac{1}{\alpha}\mathcal{G}_{\bar{\mu}_S}(z) + \big(1 - \frac{1}{\alpha}\big)\frac{1}{z} \\
q_2^* = \mathcal{G}_{\bar{\mu}_S}(z) \\
q_3^* = \frac{(z-\beta_1^*)^2}{\beta_4^{*\,2}}q_1^* - \frac{z-\beta_1^*}{\beta_4^{*\,2}} \\
q_4^* = \frac{z-\beta_1^*}{\beta_4^*}q_1^* - \frac{1}{\beta_4^*}
\end{cases}
\tag{103}
$$

Note that both sets of equations (101), (103) and (59), (76) match for $\alpha = 1$.

### F.2.3 Overlaps and optimal singular values

From (72), (101), we have:

$$
\begin{aligned}
O_Y(\gamma, \sigma_i) &\approx \frac{\alpha}{\pi\bar{\mu}_S(\gamma)}\,\mathrm{Im}\,\lim_{z\to\gamma-\mathrm{i}0^+}\frac{\beta_4^*}{Z_2(z)}\boldsymbol{y}_i^{(r)\mathsf{T}}\boldsymbol{G}_{Y^\mathsf{T}Y}\big(\frac{Z_1(z)}{Z_2(z)}\big)\boldsymbol{Y}^\mathsf{T}\boldsymbol{y}_i^{(l)} \\
&= \frac{\alpha}{\pi\bar{\mu}_S(\gamma)}\,\mathrm{Im}\,\lim_{z\to\gamma-\mathrm{i}0^+}\beta_4^*\frac{\sigma_i}{Z_1(z) - Z_2(z)\sigma_i^2}
\end{aligned}
\tag{104}
$$

Similar to (78), we can compute the optimal singular values to be:

$$
\widehat{\xi}_{y_i}^* = \frac{\alpha}{\pi\bar{\mu}_S(\gamma_i)}\,\mathrm{Im}\,\lim_{z\to\gamma_i-\mathrm{i}0^+}q_4^*
\tag{105}
$$

### F.2.4 Numerical examples

We consider the matrix $\boldsymbol{W}$ to have i.i.d. Gaussian entries with variance $1/N$, so $\mathcal{C}_{\mu_W}^{(1/\alpha)}(z) = z$. And, $\boldsymbol{X} = \boldsymbol{F} + c\boldsymbol{I}$ where $\boldsymbol{F} = \boldsymbol{F}^\mathsf{T} \in \mathbb{R}^{N\times N}$ has i.i.d. entries with variance $1/N$, and $c \neq 0$ is a real number, so $\mathcal{R}_{\rho_X}(z) = z + c$. With these choices, the solution (103) simplifies to:

$$
\begin{cases}
\beta_1^* = \frac{1}{\alpha}q_2^* + q_3^*, \quad \beta_2^* = q_1^*, \quad \beta_3^* = q_1^*, \quad \beta_4^* = q_4^* + c \\
q_1^* = \frac{1}{\alpha}\mathcal{G}_{\bar{\mu}_S}(z) + \big(1 - \frac{1}{\alpha}\big)\frac{1}{z}, \quad q_2^* = \mathcal{G}_{\bar{\mu}_S}(z) \\
q_3^* = \frac{(z-\beta_1^*)^2}{\beta_4^{*\,2}}q_1^* - \frac{z-\beta_1^*}{\beta_4^{*\,2}}, \quad q_4^* = \frac{z-\beta_1^*}{\beta_4^*}q_1^* - \frac{1}{\beta_4^*}
\end{cases}
\tag{106}
$$

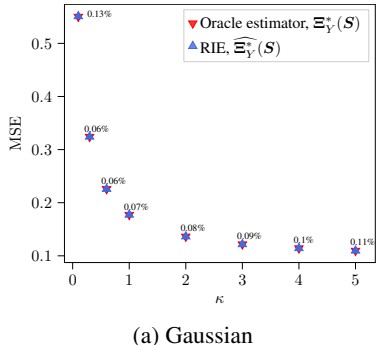

(a) Gaussian

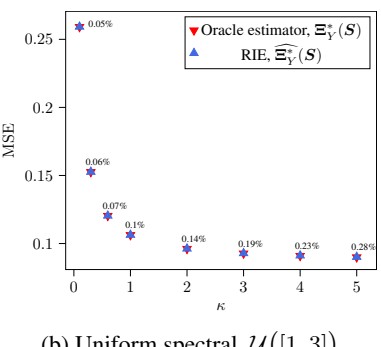

(b) Uniform spectral, $\mathcal{U}\big([1,3]\big)$

Figure 25: Estimating $\boldsymbol{Y}$. MSE is normalized by the norm of the signal, $\|\boldsymbol{Y}\|_{\mathrm{F}}^2$. $\boldsymbol{X}$ is a shifted Wigner matrix with $c = 3$, and $\boldsymbol{W}$ has i.i.d. Gaussian entries of variance $1/N$, and $N/M = 2$. The RIE is applied to $N = 2000$, $M = 1000$, and the results are averaged over 10 runs (error bars are invisible).

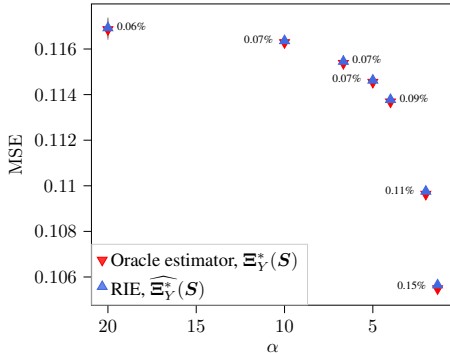

Figure 26: MSE of estimating $\boldsymbol{Y}$ as a function of aspect-ratio $\alpha > 1$, $\boldsymbol{Y}$ has Gaussain entries of variance $1/N$, and $\kappa = 5$. MSE is normalized by the norm of the signal, $\|\boldsymbol{Y}\|_{\mathrm{F}}^2$. $\boldsymbol{X}$ is a shifted Wigner matrix with $c = 3$, and $\boldsymbol{W}$ has i.i.d. Gaussian entries of variance $1/N$. The RIE is applied to $N = 2000$, $M = 1/\alpha N$, and the results are averaged over 10 runs (error bars are invisible). Average relative error between RIE $\widehat{\Xi}_Y^*(\boldsymbol{S})$ and Oracle estimator is also reported.

After a bit of algebra, we find that $q_4^*$ is the solution to the following qubic equation:

$$
\begin{aligned}
2x^3 + 3c\,x^2 &+ \left[c^2 + 2 - \left(z - \frac{1}{\alpha}\mathcal{G}_{\bar{\mu}_S}(z)\right)\left(\frac{1}{\alpha}\mathcal{G}_{\bar{\mu}_S}(z) + \frac{\alpha-1}{\alpha z}\right)\right] x \\
&- c\left[\left(z - \frac{1}{\alpha}\mathcal{G}_{\bar{\mu}_S}(z)\right)\left(\frac{1}{\alpha}\mathcal{G}_{\bar{\mu}_S}(z) + \frac{\alpha-1}{\alpha z}\right) - 1\right] = 0
\end{aligned}
\tag{107}
$$

In figure 25 the MSE of RIE and the oracle estimator is plotted for two cases of priors: $\boldsymbol{Y}$ with Gaussian entries and $\boldsymbol{Y}$ with uniform spectral density.

**Effect of aspect-ratio $\alpha$.** In Figure 26, we take $\boldsymbol{Y}$ to have Gaussian entries (with variance $\frac{1}{N}$), and the MSE is depicted for various values of the aspect-ratio $\alpha > 1$. We see that as $M$ decreases ($\alpha$ increases) the estimation error (of $\boldsymbol{Y}$) increases.

# G    Details on numerical implementations

## G.1    Numerical approximation of $\mathcal{G}_{\bar{\mu}_S}(z)$

The first step to construct the RIEs is to compute the Stieltjes transform of the observation matrix $\boldsymbol{S}$. In section 19.5 of [58], several approaches have been proposed to approximate the Stieltjes transform of the spectral density of a given matrix. In our implementations, we use the Cauchy kernel method in which for a given matrix $\boldsymbol{A}$ with $N$ singular values (or eigenvalues) $\big(\sigma_i\big)_{1 \le i \le N}$, $\mathcal{G}_{\mu_A}(z)$

is approximated as:

$$\mathcal{G}_{\mu_A}(z) \approx \frac{1}{N} \sum_{i=1}^{N} \frac{1}{z - \sigma_i - \mathrm{i}\eta_i}$$

with $\eta_i$'s the "widths" of the kernel at each singular value (more precisely the imaginary part is a sum of Lorentzians with width $\eta_i$ around peaks at $\sigma_i$). The construction of the RIEs uses the Stieltjes transform of the limiting symmetrized measure of $\boldsymbol{S}$. In the numerical experiments we approximate this quantity as:

$$\mathcal{G}_{\bar{\mu}_S}(z) \approx \frac{1}{2N} \sum_{i=1}^{N} \left( \frac{1}{z - \gamma_i - \mathrm{i}\eta} + \frac{1}{z + \gamma_i - \mathrm{i}\eta} \right) \tag{108}$$

with a fixed width $\eta = \sqrt{1/2N}$. Note that for the case of $\alpha > 1$ ($M < N$), $\boldsymbol{S}$ has $M$ non-trivially zero singular values, and in the approximation above $N$ should be replaced by $M$.

### G.2 Construction of the RIEs

In the RIEs derived in [14, 19], the final estimator for optimal singular values (eigenvalues) was rather simple and only required to compute the Stieltjes transform on the real line which can be easily and safely performed using the approximation above (see remark 2 in section 19.5.2 in [58]). However, in the RIEs of this work, we need to solve a system of equations in the limit $\epsilon \to 0$ ($z$ close to the real line). For this, to compute the optimal singular value $\widehat{\xi}^*_{y_i}$ (or optimal eigenvalues $\widehat{\xi}^*_{x_i}$), we evaluate $\mathcal{G}_{\bar{\mu}_S}(z)$ for $z = \gamma_i - \mathrm{i}\frac{\varepsilon}{\sqrt{2N}}$. In this way, the other parameters (e.g. $q_4^*$) are evaluated for $z$ very close to the real line, and the theoretical limit $\lim_{\epsilon \to 0}$ in (49), (78) can be estimated numerically. Moreover, as we considered a fixed width in our numerical approximation of Stieltjes transform (108), $\varepsilon$ should be chosen to compensate the width for the cases where the support of $\bar{\mu}_S$ is wider. For example, for fixed $N$, as we increase SNR (from 1 to 5) the support of $\bar{\mu}_S$ grows, however we still have $N$ singular values and the kernel's width in (108) is fixed, so $\varepsilon$ should be larger for higher SNRs to get a more accurate approximation of $\mathcal{G}_{\bar{\mu}_S}(z)$.

### G.3 Mismatch between RIEs and Oracle estimators

The RIEs are conjectured to have the same performance as the Oracle estimators in the limit $N \to \infty$. Therefore, we believe that the mismatch between the proposed RIEs and the Oracle estimators is a finite size effect. Moreover, this finiteness affects the accuracy of estimated parameters, since $\mathcal{G}_{\bar{\mu}_S}(z)$ is approximated numerically and we do not use random matrix theory to find its exact form.

Generically, the mismatch between the RIE and Oracle estimator is larger for the case of estimating $\boldsymbol{X}$. We expect that this is because of the extra approximation step in the derivation of the optimal eigenvalues. In the fifth line of (49), the sums are approximated by an integral which is the Stieltjes transform of $\rho_X$. This approximation does not appear in derivation of the optimal singular values for $\boldsymbol{Y}$, see (78).

All in all, the small relative error (less than $1\%$) between RIEs and Oracle estimators in our numerical results validates our optimality conjecture and demonstrates that RIEs can be successfully used in practice.

## H  Spherical integrals and matrix lemmas

### H.1  Spherical Integrals

For two symmetric matrices $\boldsymbol{A}, \boldsymbol{B} \in \mathbb{R}^{N \times N}$, the *spherical integral* is defined as:

$$\mathcal{I}_N(\boldsymbol{A}, \boldsymbol{B}) = \left\langle \exp \left\{ \frac{N}{2} \operatorname{Tr} \boldsymbol{A} \boldsymbol{U} \boldsymbol{B} \boldsymbol{U}^\intercal \right\} \right\rangle_{\boldsymbol{U}}$$

where the average is w.r.t. the *Haar* measure over the group of (real) orthogonal $N \times N$ matrices. The spherical integrals can also be defined w.r.t. the unitary or symplectic group. These integrals are often referred to as *Harish Chandra-Itzykson-Zuber* (HCIZ) integrals in mathematical physics literature. The study of these objects dates back to the work of mathematician Harish Chandra [62]

and they have since been extensively studied and developed in both physics and mathematics. In particular, [21] studied the limit of the integral in the case where one of the matrices, say $\boldsymbol{A}$, has finite rank.

**Theorem 1** (**Rank-one spherical integral**, Guionnet and Maïda [21])**.** *Let $\theta$ be the only non-zero eigenvalue of $\boldsymbol{A}$ (so it is rank one), and the empirical eigenvalue distribution of $\boldsymbol{B}$ converge weakly towards $\rho_B$. Then, for $\theta$ sufficiently small (see details in Theorem 2 in [21]), we have:*

$$\lim_{N \to \infty} \frac{1}{N} \ln \mathcal{I}_N(\boldsymbol{A}, \boldsymbol{B}) = \frac{1}{2} \int_0^\theta \mathcal{R}_{\rho_B}(t)\, dt \equiv \frac{1}{2} \mathcal{P}_{\rho_B}(\theta) \tag{109}$$

When $\boldsymbol{A}$ has higher (but finite) rank, theorem 7 in [21] states that the limit is the sum over eigenvalues of the expression on the rhs of (109).

**Non-symmetric case.** In the non-symmetric case the *rectangular spherical integral* is defined, for the matrices $\boldsymbol{A} \in \mathbb{R}^{M \times N}, \boldsymbol{B} \in \mathbb{R}^{N \times M}$, as:

$$\mathcal{J}_N(\boldsymbol{A}, \boldsymbol{B}) = \left\langle \exp\left\{ \sqrt{NM}\, \mathrm{Tr}\, \boldsymbol{A}\boldsymbol{U}\boldsymbol{B}\boldsymbol{V} \right\} \right\rangle_{\boldsymbol{U}, \boldsymbol{V}}$$

where $\boldsymbol{U} \in \mathbb{R}^{N \times N}, \boldsymbol{V} \in \mathbb{R}^{M \times M}$, and the expectation is w.r.t. the *Haar* measure over orthogonal matrices of size $N \times N$ and $M \times M$.

**Theorem 2** (**Rank-one rectangular spherical integral**, Benaych-Georges [22])**.** *Let $N/M \to \alpha \in (0, 1]$, and $\theta$ be the only non-zero singular value of $\boldsymbol{A}$, and the empirical singular value distribution of $\boldsymbol{B}$ converges weakly towards $\mu_B$. Then, for $\theta$ sufficiently small (see details in Theorem 2.2 in [22]), we have:*

$$\lim_{N \to \infty} \frac{1}{N} \ln \mathcal{J}_N(\boldsymbol{A}, \boldsymbol{B}) = \int_0^\theta \frac{\mathcal{C}_{\mu_B}^{(\alpha)}(t^2)}{t}\, dt = \frac{1}{2} \int_0^{\theta^2} \frac{\mathcal{C}_{\mu_B}^{(\alpha)}(t)}{t}\, dt \equiv \frac{1}{2} \mathcal{Q}_{\mu_B}^{(\alpha)}(\theta^2) \tag{110}$$

In our derivation, we use a generalization of this formula, namely when $\boldsymbol{A}$ has higher (but fixed) rank, the limit is the sum over singular values of the expression on the rhs of (110). Although we are not aware if this generalization has been proved, we believe that the ideas found in [63] can be applied to show it holds.

**Remark 6.** *It is known that additional terms may be present on the rhs of* (109) *and* (110) *when the parameter $\theta$ is "large". This has been rigorously proved at least in the case of symmetric $\boldsymbol{A}$ and $\boldsymbol{B}$ (see theorem 6 in [21]). In the replica calculation the order of magnitude of this parameter is determined by the solutions of the saddle point equations, but it is difficult to fully control its order of magnitude. However the numerics show very good agreement between our explicit RIEs and the Oracle estimator, which strongly suggests it is sound to use* (109) *and* (110)*.*

## H.2   Matrix analysis tools

**Proposition 3** (**Inverse of a block matrix**, Bernstein [61])**.** *For a block matrix $\boldsymbol{F} = \begin{bmatrix} \boldsymbol{A} & \boldsymbol{B} \\ \boldsymbol{C} & \boldsymbol{D} \end{bmatrix}$ with $\boldsymbol{A} \in \mathbb{R}^{N \times N}, \boldsymbol{B} \in \mathbb{R}^{N \times M}, \boldsymbol{C} \in \mathbb{R}^{M \times N}, \boldsymbol{D} \in \mathbb{R}^{M \times M}$, if $\boldsymbol{A}$ and $\boldsymbol{D} - \boldsymbol{C}\boldsymbol{A}^{-1}\boldsymbol{B}$, are non-singular, then,*

$$\boldsymbol{F}^{-1} = \begin{bmatrix} \boldsymbol{A}^{-1} + \boldsymbol{A}^{-1}\boldsymbol{B}(\boldsymbol{D} - \boldsymbol{C}\boldsymbol{A}^{-1}\boldsymbol{B})^{-1}\boldsymbol{C}\boldsymbol{A}^{-1} & -\boldsymbol{A}^{-1}\boldsymbol{B}(\boldsymbol{D} - \boldsymbol{C}\boldsymbol{A}^{-1}\boldsymbol{B})^{-1} \\ -(\boldsymbol{D} - \boldsymbol{C}\boldsymbol{A}^{-1}\boldsymbol{B})^{-1}\boldsymbol{C}\boldsymbol{A}^{-1} & (\boldsymbol{D} - \boldsymbol{C}\boldsymbol{A}^{-1}\boldsymbol{B})^{-1} \end{bmatrix}$$

**Block structure of $\boldsymbol{G}_{\mathcal{S}}(z)$**   The matrix $\boldsymbol{G}_{\mathcal{S}}(z)$ is:

$$\boldsymbol{G}_{\mathcal{S}}(z) = (z\boldsymbol{I} - \boldsymbol{\mathcal{S}})^{-1} = \begin{bmatrix} z\boldsymbol{I}_N & -\boldsymbol{S} \\ -\boldsymbol{S}^{\mathsf{T}} & z\boldsymbol{I}_M \end{bmatrix}^{-1}$$

Using Proposition 3, first we need to compute the inverse matrix $\left(z\boldsymbol{I}_M - (-\boldsymbol{S}^{\mathsf{T}})(z\boldsymbol{I}_N)^{-1}(-\boldsymbol{S})\right)^{-1}$ which simply reads:

$$\left(z\boldsymbol{I}_M - \frac{1}{z}\boldsymbol{S}^{\mathsf{T}}\boldsymbol{S}\right)^{-1} = z\left(z^2\boldsymbol{I}_M - \boldsymbol{S}^{\mathsf{T}}\boldsymbol{S}\right)^{-1} = z\boldsymbol{G}_{\boldsymbol{S}^{\mathsf{T}}\boldsymbol{S}}(z^2)$$

Consequently, we find:

$$\boldsymbol{G}_{\mathcal{S}}(z) = \begin{bmatrix} \frac{1}{z}\boldsymbol{I}_N + \frac{1}{z}\boldsymbol{S}\boldsymbol{G}_{\boldsymbol{S}^{\mathsf{T}}\boldsymbol{S}}(z^2)\boldsymbol{S}^{\mathsf{T}} & \boldsymbol{S}\boldsymbol{G}_{\boldsymbol{S}^{\mathsf{T}}\boldsymbol{S}}(z^2) \\ \boldsymbol{G}_{\boldsymbol{S}^{\mathsf{T}}\boldsymbol{S}}(z^2)\boldsymbol{S}^{\mathsf{T}} & z\boldsymbol{G}_{\boldsymbol{S}^{\mathsf{T}}\boldsymbol{S}}(z^2) \end{bmatrix} \tag{111}$$

**Inverse of $C_X^*$**    For $C_X^*$ since the blocks $B, C$ are zero, the inverse is simply:

$$
\begin{aligned}
C_X^{*\,-1} &= \begin{bmatrix} \left[(z - \zeta_1^*)I_N - \zeta_3^* X^2\right]^{-1} & 0 \\ 0 & \left[(z - \zeta_2^*)I_M\right]^{-1} \end{bmatrix} \\[6pt]
&= \begin{bmatrix} \frac{1}{\zeta_3^*}\left[\frac{z - \zeta_1^*}{\zeta_3^*}I_N - X^2\right]^{-1} & 0 \\ 0 & \frac{1}{z - \zeta_2^*}I_M \end{bmatrix} \\[6pt]
&= \begin{bmatrix} \frac{1}{\zeta_3^*}G_{X^2}\left(\frac{z - \zeta_1^*}{\zeta_3^*}\right) & 0 \\ 0 & \frac{1}{z - \zeta_2^*}I_M \end{bmatrix}
\end{aligned}
\tag{112}
$$

**Inverse of $C_Y^*$**    Let the block structure of $C_Y^*$ be as in Proposition 3, then

$$
\begin{aligned}
(D - CA^{-1}B)^{-1} &= \left((z - \beta_2^*)I_M - \beta_3^* Y^\mathsf{T}Y - \frac{\beta_4^{*2}}{z - \beta_1^*}Y^\mathsf{T}Y\right)^{-1} \\[6pt]
&= \left((z - \beta_2^*)I_M - \left(\beta_3^* + \frac{\beta_4^{*2}}{z - \beta_1^*}\right)Y^\mathsf{T}Y\right)^{-1} \\[6pt]
&= (z - \beta_1^*)\left(Z_1(z)I_M - Z_2(z)Y^\mathsf{T}Y\right)^{-1} \\[6pt]
&= \frac{z - \beta_1^*}{Z_2(z)}\left(\frac{Z_1(z)}{Z_2(z)}I_M - Y^\mathsf{T}Y\right)^{-1} \\[6pt]
&= \frac{z - \beta_1^*}{Z_2(z)}G_{Y^\mathsf{T}Y}\left(\frac{Z_1(z)}{Z_2(z)}\right)
\end{aligned}
$$

where $G_{Y^\mathsf{T}Y}$ is the resolvent of the matrix $Y^\mathsf{T}Y$. So, we have

$$
C_Y^{*\,-1} = \begin{bmatrix} (z - \beta_1^*)^{-1}I_N + \frac{\beta_4^{*2}}{(z - \beta_1^*)Z_2(z)}Y G_{Y^\mathsf{T}Y}\left(\frac{Z_1(z)}{Z_2(z)}\right)Y^\mathsf{T} & \frac{\beta_4^*}{Z_2(z)}Y G_{Y^\mathsf{T}Y}\left(\frac{Z_1(z)}{Z_2(z)}\right) \\[6pt] \frac{\beta_4^*}{Z_2(z)}G_{Y^\mathsf{T}Y}\left(\frac{Z_1(z)}{Z_2(z)}\right)Y^\mathsf{T} & \frac{z - \beta_1^*}{Z_2(z)}G_{Y^\mathsf{T}Y}\left(\frac{Z_1(z)}{Z_2(z)}\right) \end{bmatrix}
$$

**Lemma 3.** *Consider two vectors $x, y \in \mathbb{R}^N$. The symmetric matrix $xy^\mathsf{T} + yx^\mathsf{T}$ has rank at most two with non-zero eigenvalues $x^\mathsf{T}y \pm \|x\|\|y\|$.*

*Proof.* Construct the matrices $A \in \mathbb{R}^{2\times N}, B \in \mathbb{R}^{N\times 2}$ as follows:

$$
A = \begin{bmatrix} x^\mathsf{T} \\ y^\mathsf{T} \end{bmatrix}, \quad B = \begin{bmatrix} y & x \end{bmatrix}
$$

Then, we have that $xy^\mathsf{T} + yx^\mathsf{T} = BA$. Using the lemma 4, we have that:

$$
z^2 \det\left(zI_N - BA\right) = z^N \det\left(zI_2 - AB\right)
$$

So, the characteristic polynomial of $xy^\mathsf{T} + yx^\mathsf{T}$ is $z^{N-2} \det\left(zI_2 - AB\right)$, which implies that the $xy^\mathsf{T} + yx^\mathsf{T}$ has eigenvalue 0 with multiplicity $N - 2$, plus the eigenvalues of the $2 \times 2$ matrix $AB$. The matrix $AB$ is:

$$
AB = \begin{bmatrix} x^\mathsf{T}y & \|x\|^2 \\ \|y\|^2 & x^\mathsf{T}y \end{bmatrix}
$$

which has two eigenvalues $x^\mathsf{T}y \pm \|x\|\|y\|$. $\qquad\square$

**Lemma 4.** *For matrices $A \in \mathbb{R}^{M\times N}, B \in \mathbb{R}^{N\times M}$, we have:*

$$
z^M \det\left(zI_N - BA\right) = z^N \det\left(zI_M - AB\right)
$$

*Proof.* Construct the matrices $C, D \in \mathbb{R}^{(M+N)\times(M+N)}$ as follows:

$$
C = \begin{bmatrix} zI_M & A \\ B & I_N \end{bmatrix}, \quad D = \begin{bmatrix} I_M & 0_{M\times N} \\ -B & zI_N \end{bmatrix}
$$

We have:

$$
\det CD = z^N \det\left(zI_M - AB\right), \quad \det DC = z^M \det\left(zI_N - BA\right)
$$

The result follows from the fact that $\det CD = \det DC$. $\qquad\square$

