# OpenReview forum: "Bayesian Extensive-Rank Matrix Factorization with Rotational Invariant Priors"
_NeurIPS.cc/2023/Conference — NeurIPS 2023 spotlight_

### Official Review · Reviewer_PN68 · 2023-07-05

**Soundness:** 3 good
**Presentation:** 4 excellent
**Contribution:** 4 excellent
**Rating:** 7
**Confidence:** 3

**Summary:**

The authors consider the problem of matrix factorization of a noisy measurement in the setting of all matrices having an rotationally invariant prior. They provide a non-rigorous but comprehensive theoretical derivation of their results. They also provide a number of experiments validating there theoretical claims.

**Strengths:**

- Explicit formulas for the reconstruction of the matrix factors
- Strong experimental support for the theoretical claims

**Weaknesses:**

- The analysis is limited to the rotationally invariant setting

**Questions:**

- It would be nice to have a short summary and outlook at the end of the paper
- In line 67 what do you mean by proper distribution?
- Can you briefly elaborate on the relationship between your work and the works 34-36 mentioned in the introduction?

**Limitations:**

The limitations have been adequately addressed.

---

> ### Author Rebuttal · Authors · 2023-08-07
>
> We thank the reviewer for the review. We address the questions below:
>
> 1. We will surely include a conclusion section in the final version, also taking into account comments made by the referees.
>
> 2. Depending on the nature of the problem, one can consider a general class of priors for the factors, and tries to find the parameters maximizing the posterior. For example, in ref [37] authors consider Gaussian priors and try to estimate the matrices. We will rephrase the wording "proper distribution" in the final version.
>
> 3. Ref[34] is a general reference about variational inference. In [35,36], matrix factorization problem is solved when some of the data points (entries of the observation) are missing. In our setting, we have the full observation of data, therefore the problem setting are different.  Solving MF with full observed data based on variational inference is studied in ref[37]. In ref[37], authors showed that under Gaussian prior (which is rot. Inv.) the optimal estimate using variational inference approach is a reweighting of the SVD of the observation, which can be seen as a RIE.

---

> > ### Comment · Reviewer_PN68 · 2023-08-17
> >
> > I would like to thank the authors for the clarifications. I have slightly increased my evaluation.

---

### Official Review · Reviewer_AAM7 · 2023-07-05

**Soundness:** 3 good
**Presentation:** 3 good
**Contribution:** 3 good
**Rating:** 7
**Confidence:** 3

**Summary:**


This paper explores the Matrix Factorization problem, which involves estimating the matrices $X \in \mathbb{R}^{N \times N}$ and $Y \in \mathbb{R}^{N \times M}$ given the noisy matrix $S = \sqrt{\kappa} X Y + W$.

The focus is on the high-dimensional regime, where $N/M \to \alpha$, and the investigation includes bi-rotationally invariant $Y$ and $W$, as well as symmetric rotationally invariant $X$.

The authors examine rotationally invariant estimators, which are estimators that share the same singular vectors as the noisy matrix $S$.
The paper derives rotationally invariant estimators based on oracle knowledge of the target matrices and demonstrates that they are also Bayes optimal. By assuming concentration and utilizing replica methods, the paper derives explicitly computable estimators from the oracle estimators. The empirical performance of these derived estimators is investigated and shown to closely match that of the oracle estimators. This suggests that the estimators derived using non-rigorous methods from statistical physics are indeed optimal.

**Strengths:**

The low-rank matrix factorization problem with finite-rank matrices is now a well-studied topic. Similarly, the low-rank matrix denoising problem with extensive (diverging) ranks has garnered recent interest. However, results on matrix factorization with extensive ranks have been relatively scarce. This paper aims to fill this gap in the literature by providing results for the matrix factorization problem with diverging ranks and under general rotationally-invariant priors.

The paper is well-written overall, and Section 5 provides a concise and easy-to-follow overview of the derivation of the results, which are otherwise quite complex.

**Weaknesses:**

The main results of the paper, which are the explicitly computed Rotationally Invariant Estimators, rely on non-rigorous methods from statistical physics. While the empirical results are compelling, it would be valuable in the future to establish a more solid theoretical foundation for these estimators.

Moreover, the assumptions made on the matrices $X$ and $Y$ may be considered somewhat unnatural. It would be beneficial for the authors to provide additional motivation as to why the findings of this paper could be of interest to the NeurIPS community beyond the specific problem examined here. This could help clarify the broader significance and potential applications of the research.

**Questions:**

Can the authors expand on the relevance of the methodology developed in the paper for the analysis of the weight matrices of neural networks?

**Limitations:**

Theoretical paper with no immediate negative societal impact.

---

> ### Author Rebuttal · Authors · 2023-08-07
>
> We thank the reviewer for the comments.
>
> We agree that making the derivations mathematically rigorous is an interesting research direction. We want to bring your attention to the papers:
>  "Optimal cleaning for singular values of cross-covariance matrices" (arXiv:1901.05543) , "A short proof of Ledoit-Péché's RIE formula for covariance matrices ( arXiv:2201.05690),  that rigorously derive the optimal RIE for covariance estimation with Gaussian priors. The method developed in these works involves Gaussian integration by parts and Gaussian concentration. We believe that, using this technique we can establish the optimality of the proposed estimator at least for Gaussian priors, and this analysis would not involve replica method and spherical integrals. However such a rigorous mathematical analysis is at the moment beyond the scope of this paper.
>
> We agree that the assumptions are strong and far from practice, however the settings considered in the manuscript is one of the first setting that the matrix factorization can be solved optimally in the high-rank regimes, and we believe that this work may open up a way to study this problem in more general settings. Moreover, some of the assumptions are required to show the optimality of the estimator, but the estimator can be used in practice under milder conditions to get a spectral estimate that can be refined.
>
> Concerning the specific question on neural networks:
>
> We believe that our method can be applied to study the weight matrices in neural networks, at least when there is no non-linearity in the system. Non-linearities are prevalent in neural networks models which does not bode very well with rotation invariance, but it might be possible to consider "linearizations" such as an NTK approximation to circumvent this issue. These is an interesting open problem.

---

### Official Review · Reviewer_LQDn · 2023-07-07

**Soundness:** 3 good
**Presentation:** 3 good
**Contribution:** 2 fair
**Rating:** 7
**Confidence:** 3

**Summary:**

This paper considers a matrix factorization problem in a setting where the rank of the factor matrices grow linearly with the ambient dimensions. They assume that the factors follow a prior distribution such that: (1) One of the matrix factor is symmetric, (2) Both factors and the noise are drawn from rotationally invariant distributions. (3) The priors are known to the statistician.

They propose to study a class of rotationally invariant estimators. They derive a closed form expression for the oracle estimator in this class, and show that it is Bayes optimal. They propose an estimator and conjecture that its performance matches the oracle. They provide evidence of their conjecture through experiments.

**Strengths:**

This paper seems to be the first one that explores MF in this challenging setting, and opens up a new research direction that might be of interest. They derive a closed form neat expression for the Bayes optimal estimator, although this cannot be implemented in practice. As an alternative, they propose an estimator that can be implemented in practice. Although their result is non-rigorous, simulation suggests that this is the correct thing to do. A sub-optimal estimator for one factor that does not require prior knowledge of $\mu_X$ is also proposed. Their presentation is nice and clean.

**Weaknesses:**

Although this paper presents nice technical contributions, it is not clear why this prior structure should be considered in practice. It would be nice to give a few practical examples which past results can not cover but this work do.

**Questions:**

1. I am wondering how sensitive is the proposed estimators to misspecification of prior.
2. If the prior is not presented, is there a way to estimate them? I feel the assumption that prior is known for both the factors and noise is a bit strong. Perhaps the authors should comment a little bit. For example, explain when it is reasonable to assume this information is given.
3. Several literatures that might be useful to include:

Information-theoretic limits of MF:
[1] Bayes-optimal limits in structured PCA, and how to reach them (arXiv:2210.01237)
[2] Fundamental Limits of Low-Rank Matrix Estimation with Diverging Aspect Ratios (arXiv:2211.00488)

AMP for rotationally invariant matrices:
[1] Approximate Message Passing algorithms for rotationally invariant matrices (arXiv:2008.11892)


**Limitations:**

The limitations are clearly reflected in the model assumption, and societal impact not applicable.

---

> ### Author Rebuttal · Authors · 2023-08-07
>
> We thank the referee for the review and additional suggested references. We will add those (with possibly other references) in the final version.
>
>  We agree that the required assumptions are restrictive and do not necessarily hold in practice, however as mentioned in the manuscript the proposed estimators can be used without the assumptions to get a sub-optimal estimate that can be processed further. Moreover, as  pointed out in the comments, this work is the first one to consider the problem in the high rank regime, and we believe it may open up a new way to study the problem in this regime.
>
> We address his/her questions below:
>
> 1. Analyzing the sensitivity of the estimators to the mismatched priors is an interesting object of study that requires independent investigation, and is beyond the scope of this manuscript. There exists a rich variety of scenarios that one can consider. Here, we give a brief speculation on the performance of RIE with mismatched priors.
>
>     $\bullet$ Mismatched prior on the signal. The RIE for estimating $\mathbf{Y}$ is independent of the prior of $\mathbf{Y}$, and it is optimal in the case of mismatched prior on $\mathbf{Y}$.
>
>     For the case of estimating $\mathbf{X}$ with mismatched prior, the estimator $\sqrt{\widehat{\mathbf{\Xi}_{X^2}} (\mathbf{S})}$ is applicable and is independent of the prior, although sub-optimal. However, the RIE for $\mathbf{X}$ depends on the prior, and misspecification of the prior leads to the poor performance of RIE (comparing to the Oracle). But, we believe that the RIE can get a non-trivial estimate of $\mathbf{X}$ (which is indeed sub-optimal), as the parameters $\zeta_1, \zeta_3$ in eq. (8,9) are correctly evaluated. We provide a numerical check of the performance of RIE in this case in the file uploaded as official comment above.
>
>     $\bullet$ Mismatched prior on other factors. Misspecifying the priors for the other two factors (than the signal), leads to the incorrect evaluation of the parameters for RIE that can change the performance significantly. For example, consider estimating the matrix $\mathbf{X}$, with $\mathbf{Y}$ and $\mathbf{W}$ both Gaussian matrices but with variances unknown to the statistician. Assuming mismatched variances (different than the real ones) for  $\mathbf{Y}$ and  $\mathbf{W}$, leads to rectangular R-transforms which are equal to the true ones times some constant. This results in a non-trivial mismatch in the parameters $\zeta_1, \zeta_3$ (eq. 8,9) used in the estimator, which will change the estimated optimal eigenvalue. We provide a numerical check of the performance of RIE in this case in the file uploaded as official comment above.
>
>
> 2. We agree that the assumptions are strong, however note that these assumptions are required to show the optimality of the estimators.
> In general, for $N$ large enough estimating the prior on one factor is possible if the prior of the other two factors are known. For example, knowing the prior of $\mathbf{X}$ and the noise, we can estimate the spectral distribution of $\mathbf{Y}$, using the spectral distribution of the observation. For this, one needs to go through the free additive/multiplicative convolutions: From spectral measure of the observation and the noise, one can find the rectangular R-transform of spectral distribution of the product $\mathbf{XY}$, then again using the knowledge of prior of $\mathbf{X}$ one can find an estimate of the spectral distribution of $\mathbf{Y}$.
>
>      Moreover, even if we consider specific classes of priors for the factors we do not necessarily find unique estimates. For example, in the simpler setting of additive denoising $\mathbf{X} + \mathbf{Z}$, under the assumption that both $\mathbf{X}$, $\mathbf{Z}$ have i.i.d. Gaussian entries with unknown variances, it is not possible to estimate the variances uniquely for both $\mathbf{X}$,$\mathbf{Z}$.
>
>      Rotational invariance for the priors might not be very natural in practice, but our estimators could be used as an initialization for other algorithms (for example iterative ones - although for high rank these remain to be investigated). Additionally, assuming other priors than Gaussian does not lead to neat expressions for the estimators and the systems of equations must be solved numerically, which is impractical. On the other hand considering Gaussian priors, especially for the noise, is a common assumption in practice.

---

> > ### Comment · Reviewer_LQDn · 2023-08-17
> >
> > I want to thank the authors for the detailed clarification. I have increased my rating.

---

### Official Review · Reviewer_LgE7 · 2023-07-17

**Soundness:** 4 excellent
**Presentation:** 4 excellent
**Contribution:** 4 excellent
**Rating:** 9
**Confidence:** 4

**Summary:**

For a matrix factorization model S = \kappa XY + W, this paper proposes a method for estimating X and Y from S under the assumption that priors of X, Y, and W satisfy certain rotation invariance properties and their distributions of eigen/singular values are known. The proposed method is rather simple. First, using the singular value decomposition, we assess the left and right singular bases of S, which are eigen/singular bases of X and Y. Next, keeping the bases fixed, the eigen/singular values of X and Y are adjusted for minimizing the average mean squared errors, which can be analytically performed with the knowledge of limiting distributions of eigen/singular values of X, Y, and W. The Bayesian optimality is shown for the proposed estimators using the replica method from statistical mechanics and random matrix theory.

**Strengths:**

As long as I know, this is the first paper that shows a concrete practical method for constructing the Bayes optimal estimator for O(N) rank matrix factorization problem.

**Weaknesses:**

The shown optimality holds under rather many assumptions (rotation invariance, knowledge of limiting eigen/singular value distributions) are necessary.

**Questions:**

I am curious about what happens when c is set to zero for the shifted Wigner. Does it cause any singularity for the estimator? Or, does the estimator of X continuously converge to zero matrix as c -> 0?

**Limitations:**

Yes.

---

> ### Author Rebuttal · Authors · 2023-08-07
>
> We thank the reviewer for positive comments. We address his/her question below:
>
> $\bullet$ For $c=0$, the prior on spectral of $\mathbf{X}$ is symmetric, $\rho_X(x)=\rho_X(-x)$, and using the definition of Stieltjes transform one can see that indeed from eq. (7) the estimator is 0 for all eigenvalues. We conjecture that the estimator is continuous at $c=0$.
> Analyzing rigorously the continuity of the estimator as a function of $c$ is non-trivial, as one needs to consider the effect of $c$ on the limiting spectral measure of the observation $\mu_S$ which enters into the parameters $\zeta_1, \zeta_3$. However, ignoring this technicality, the function $G(z) + G(-z)$ in the estimator converges continuously to 0 as c goes to 0. Moreover, our numerical checks support the continuity of the estimator.

---

> > ### Comment · Reviewer_LgE7 · 2023-08-18
> >
> > Thank you for the reply. I am satisfied with it.

---

### Author Rebuttal · Authors · 2023-08-08

Numerical results on sensitivity of RIE to mismatched priors ( response to first question of Reviewer LQDn)

In figure 1, the spectral distribution of $\mathbf{X}$ is uniform on $[0,4]$, and both $\mathbf{Y}, \mathbf{W}$ are Gaussian matrices. We applied the RIE, assuming two different misspecified priors for $\mathbf{X}$:Shifted Wigner with $c=2$, Wishart with aspect-ratio $1$. Note that, the estimator $\sqrt{\widehat{{\mathbf{\Xi}_{X^2}^*}}(\mathbf{S})}$ does not require the knowledge of prior of $\mathbf{X}$, and we get the same performance for both cases, although sub-optimal. The RIE $\widehat{{\mathbf{\Xi}_X^*}}(\mathbf{S})$ performs worse than the Oracle estimator, but still it provides us with a non-trivial estimate.

In figures 2, we consider estimating $\mathbf{X}$ with Wishart prior. $\mathbf{Y}$ has uniform spectral distribution on $[1,3]$, and the noise matrix $\mathbf{W}$ has Gaussian entries. The RIE is applied with the assumption that $\mathbf{Y}$ is Gaussian so that its spectral distribution is the square-root of Marchenko-Pastur law (whose support is $[1,3]$). As discussed in the comment, we see that the RIE performs poorly, comparing to the Oracle estimator. However, note that the normalized MSE (normalized by the norm of the signal) is below $1$, and we get a non-trivial estimate of the signal matrix $\mathbf{X}$.

---

### Decision · Program_Chairs · 2023-09-21

**Decision:**

Accept (spotlight)

**Comment:**

The reviewers have good expertise and are extremely positive about the result. The manuscript considers an unusual matrix factorisation setting, one which seems to the AC to be better suited for a standard matrix analysis venue due in large part to limited attempt at motivating this question as being machine learning other than the second sentence of the introduction.  The authors and reviewers have had a useful discussion about the issues raised.